# In vivo screening characterizes chromatin factor functions during normal and malignant hematopoiesis

David Lara-Astiaso[1,2,6] ✉, Ainhoa Goñi-Salaverri[3,6], Julen Mendieta-Esteban [3,6], Nisha Narayan [1,2], Cynthia Del Valle[3], Torsten Gross[4], George Giotopoulos [1,2], Tumas Beinortas[1,2], Mar Navarro-Alonso[3], Laura Pilar Aguado-Alvaro[3], Jon Zazpe[3], Francesco Marchese[3], Natalia Torrea[3], Isabel A. Calvo [3], Cecile K. Lopez[1,2], Diego Alignani[3], Aitziber Lopez[3], Borja Saez[3], Jake P. Taylor-King [4], Felipe Prosper [3], Nikolaus Fortelny [5,7] ✉ & Brian J. P. Huntly [1,2,7] ✉

Cellular differentiation requires extensive alterations in chromatin structure and function, which is elicited by the coordinated action of chromatin and transcription factors. By contrast with transcription factors, the roles of chromatin factors in differentiation have not been systematically characterized. Here, we combine bulk ex vivo and single-cell in vivo CRISPR screens to characterize the role of chromatin factor families in hematopoiesis. We uncover marked lineage specificities for 142 chromatin factors, revealing functional diversity among related chromatin factors (i.e. barrier-to-autointegration factor subcomplexes) as well as shared roles for unrelated repressive complexes that restrain excessive myeloid differentiation. Using epigenetic profiling, we identify functional interactions between lineage-determining transcription factors and several chromatin factors that explain their lineage dependencies. Studying chromatin factor functions in leukemia, we show that leukemia cells engage homeostatic chromatin factor functions to block differentiation, generating specific chromatin factor–transcription factor interactions that might be therapeutically targeted. Together, our work elucidates the lineage-determining properties of chromatin factors across normal and malignant hematopoiesis.

Cell fate decisions are governed by the coordinated activities of transcription factors and chromatin factors, which together form gene regulatory complexes (GRCs), to orchestrate tissue-specific gene expression and cellular phenotypes[1]. The widescale description of transcription factor binding and its relationship to chromatin accessibility and gene expression, obtained from epigenomic and transcriptomic analyses across multiple developmental processes, have provided us with a highly developed understanding of the instructional

[1]Department of Haematology, University of Cambridge, Cambridge, UK. [2]Wellcome Trust-Medical Research Council Cambridge Stem Cell Institute, Cambridge, UK. [3]Centre for Applied Medical Research, University of Navarra, Pamplona, Spain. [4]Relation Therapeutics, London, UK. [5]Department of Biosciences & Medical Biology, University of Salzburg, Salzburg, Austria. [6]These authors contributed equally: David Lara-Astiaso, Ainhoa Goñi-Salaverri, Julen Mendieta-Esteban. [7]These authors jointly supervised this work: Nikolaus Fortelny, Brian J. P. Huntly. ✉e-mail: dl627@cam.ac.uk; nikolaus.fortelny@plus.ac.at; bjph2@cam.ac.uk

role for transcription factors in governing cell fates[2,3]. Conversely, although the role of individual chromatin factors, particularly those mutated across malignancies[4], are being elucidated, we still lack a global understanding of chromatin factor functions in cellular differentiation. Specifically, whether chromatin factors have specific or redundant roles during lineage differentiation, the identity of specific transcription factor–chromatin factor interactions and the molecular mechanisms that govern these interactions are unresolved questions.

We have chosen to address these fundamental questions in the exemplar process of hematopoiesis, where multiple mature cells with diverse, specific functions derive from a single self-renewing cell type, the hematopoietic stem cell (HSC)[5]. The study of hematopoiesis benefits from a comprehensive cellular blueprint[3], which describes normal differentiation and malignant transformation, and a well-annotated molecular blueprint[2–4], including detailed single-cell transcriptional landscapes and comprehensive maps of transcription factor activity derived from epigenomic and in vivo loss-of-function (LOF) studies across hematopoietic lineages[6–11]. In addition, the importance of both chromatin factors and transcription factors in hematopoietic differentiation has been further emphasized by recent studies, which have documented mutations that alter the function of transcription factors and chromatin factors to be highly recurrent and almost uniform in hematological malignancies such as acute myeloid leukemia (AML)[12].

Given our increasing ability to manipulate the dynamic epigenome[13], understanding chromatin regulation in normal and malignant hematopoiesis is key for harnessing regenerative therapeutics to restore damaged bone marrow function and in the treatment of hematological malignancies. However, many unanswered fundamental questions remain, including: What are the key interacting components (chromatin factors and transcription factors) of the GRCs that orchestrate lineage differentiation? Does their composition change during differentiation? Do chromatin factors have specificity for lineage determination and, if so, is this dependent on transcription factor recruitment to target loci? What cross talk occurs within the components of individual GRCs or between different GRCs? Which of these mechanisms are corrupted in leukemia and does this drive the induction or maintenance of the disease? In this study, we sought to answer many of these fundamental questions, combining comprehensive CRISPR-mediated knockout of multiple chromatin factors ex vivo and in vivo, with functional, epigenetic and transcriptional studies at both bulk and single-cell resolution.

## Results

### Functional screens of chromatin factors in hematopoiesis

To interrogate the roles of chromatin factors in regulating normal hematopoiesis, we developed four screening platforms coupling cytokine-instructed differentiation of primary hematopoietic progenitors collected from mice, with fluorescence-activated cell sorting (FACS) readouts to study key lineage transitions (Fig. 1a,b): (1) self-renewal versus differentiation; (2) branching between myeloid and mega-erythroid lineages; (3) myeloid differentiation of multipotent progenitors; and (4) myeloid differentiation of myeloid-primed granulocyte-macrophage progenitors (GMPs). Cross-referencing the

single-cell RNA sequencing (scRNA-seq)-derived expression profiles of each readout population to existing hematopoietic expression, we demonstrated fidelity with their in vivo counterparts (Extended Data Fig. 1a,b). Next, we generated a CRISPR library targeting 680 genes, including the vast majority of chromatin factors expressed by myeloid and mega-erythroid lineages (Supplementary Tables 1 and 2). We then delivered our library to both Cas9 (green fluorescent protein (GFP)⁺) and non-Cas9 (GFP⁻) progenitors ex vivo, cocultured them throughout our in vitro differentiation conditions, sorted 'readout' populations based on surface markers and quantified their single-guide RNA (sgRNA) distributions. Next, we calculated a lineage score for each chromatin factor by analyzing the differences in sgRNA content between populations, using the non-Cas9 distributions as a background. This analysis identified 142 chromatin factors with significant lineage scores in any of the four lineage transitions (Fig. 1c,d and Extended Data Fig. 1e). Finally, replicate screens for the strongest 200 chromatin factors demonstrated high correlation between replicates, indicating reproducible methodology (Extended Data Fig. 1d).

Examination of the lineage scores revealed a high degree of phenocopy among factors belonging to the same complex, but also antagonistic behavior for specific chromatin factor families (Fig. 1c,d and Extended Data Fig. 1e). For instance, several members of the cohesin and mediator complexes (*Stag2*, *Med20*) were required to elicit differentiation toward myeloid or mega-erythroid fates, confirming dynamic chromatin looping as a general requirement for differentiation[14,15]. However, genes associated with the RNA elongation machinery (*Phf5a*, *Ash1l*) operated to preserve progenitor multipotency. As reported in other systems, H3K4 methyltransferases and chromatin remodelers showed high functional diversity[16]. Myeloid/lymphoid or mixed-lineage leukemia protein 4 (MLL4) complex genes (*Kmt2d*, *Kdm6a*) regulated progenitor identities and early myeloid priming, while histone-lysine N-methyltransferase Set1-like (SET1) complex components were required for differentiation and erythroid priming. BRG1- or BRM-associated factors (BAF) members[16] behaved predominantly as pro-myeloid regulators, but nucleosome remodeling deacetylase (NuRD) and imitation switch (ISWI) factors[17–19] facilitated mega-erythroid fates. Finally, and in contrast, certain repressive complexes demonstrated functional homogeneity, where heterochromatin (*Setdb1*, *Cbx3*), histone deacetylases (*Hdac1* and *Hdac3*) and coREST[20,21] members all functioned as myeloid repressors. Validating these results, the screen-based phenotypes of ten individual chromatin factor knockouts were confirmed by analyzing their ex vivo differentiation patterns (Fig. 1e and Extended Data Fig. 2a–d).

Collectively, these findings highlight substantial functional diversity within chromatin complexes, suggesting that specific chromatin factor subcomplexes work at different branches and stages along the differentiation trajectories.

### Chromatin factor roles during in vivo hematopoiesis

Next, we used Perturb-seq to explore the functional diversity of 80 factors in their proper physiological context, in vivo hematopoiesis, at single-cell resolution. This list comprises 60 chromatin factors with strong dependencies combined with 20 lineage-specific transcription factors, with known functional effects, as control perturbations.

---

**Fig. 1 | Functional screens identify lineage specificities for chromatin factors in hematopoiesis. a**, Schema of the experimental approach used, describing the murine progenitor populations used, their isolation and transduction with the CRISPR library, subsequent differentiation, flow sorting of specific differentiation readouts and read-count based analysis of function. CF, chromatin factor; TF, transcription factor. **b**, Differentiation systems and FACS-based readouts: (1) self-renewal versus differentiation; (2) lineage priming: mega-erythroid versus myeloid; (3) myeloid differentiation: mature myeloid versus non-myeloid; and (4) terminal myeloid maturation versus immature. **c,** Averaged lineage scores of 'differentiation versus self-renewal' (*y* axis) versus

'lineage priming' (*x* axis) for 554 genes. Significant hits (*n* = 93 genes with 50% or more significant guide RNAs (gRNAs) in either comparison) are shown with a red cross. NTC sgRNAs are shown with a blue cross. Data for non-Cas9 cells are shown in the background using a yellow-blue density. **d**, Lineage scores for chromatin factors grouped on the basis of complex membership. The dot color represents the lineage score, the dot size the percentage of significant guides (Supplementary Table 3). HDAC, histone deacetylase; PRMT, protein arginine methyltransferase. **e**, Exemplar immunophenotypic validations for chromatin factors in the lineage priming (top) and myeloid differentiation (bottom) systems. SSC-A, side scatter area.

Experimentally, multipotent progenitors (Lin⁻Sca1⁺c-Kit⁺ (LSKs)) were transduced with targeted libraries before transplantation into sublethally irradiated mice. Thereafter, 2 weeks after transplant, we isolated their progeny (Lin⁻ and Lin⁺c-Kit⁺ cells) and used Perturb-seq

to jointly measure their transcriptomes and chromatin factor perturbations (Fig. 2a and Supplementary Fig. 5). This approach reconstituted the main hematopoietic lineages: progenitor (HSC) myeloid (GMP) (granulocyte progenitor, granulocytes and monocytes),

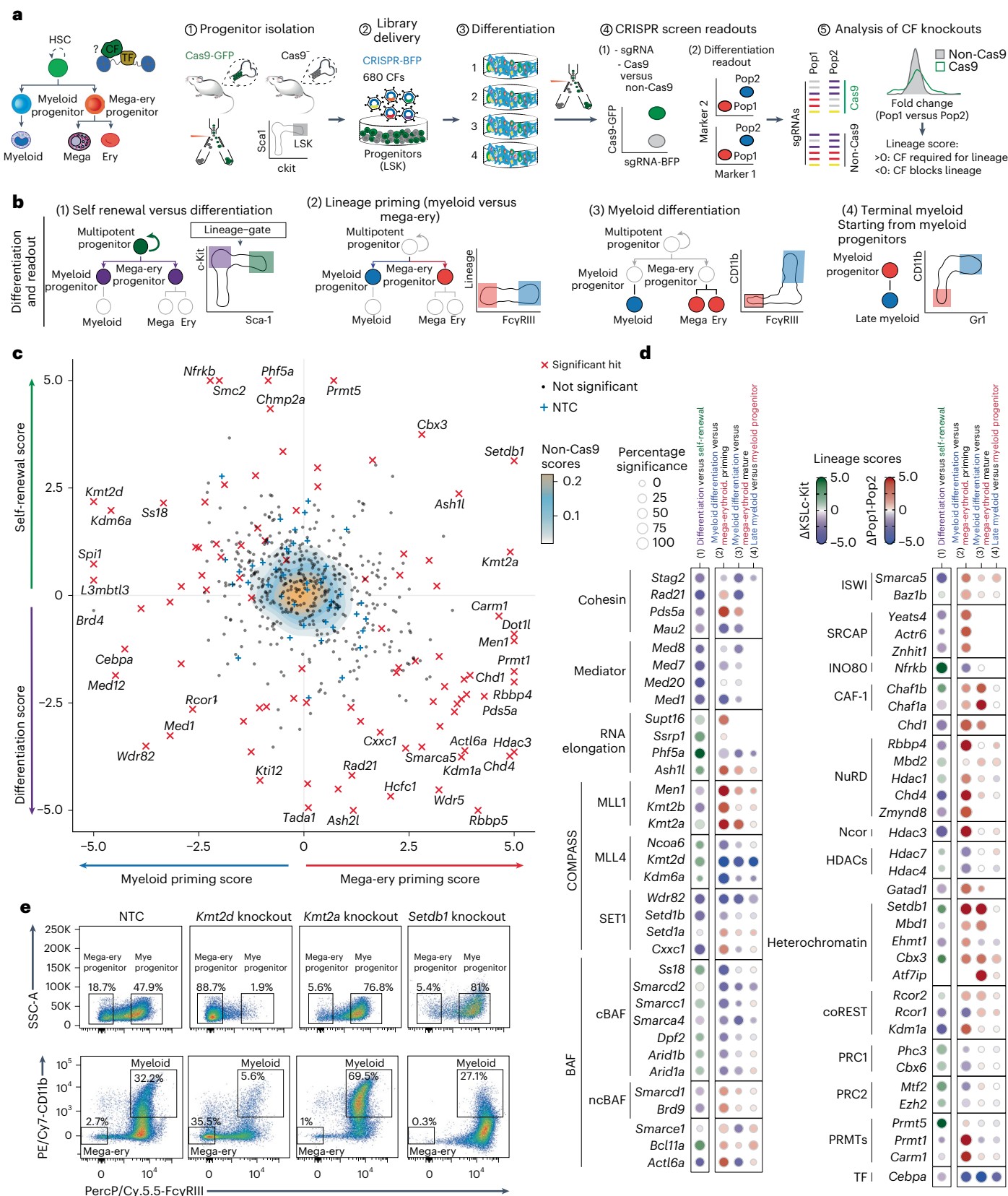

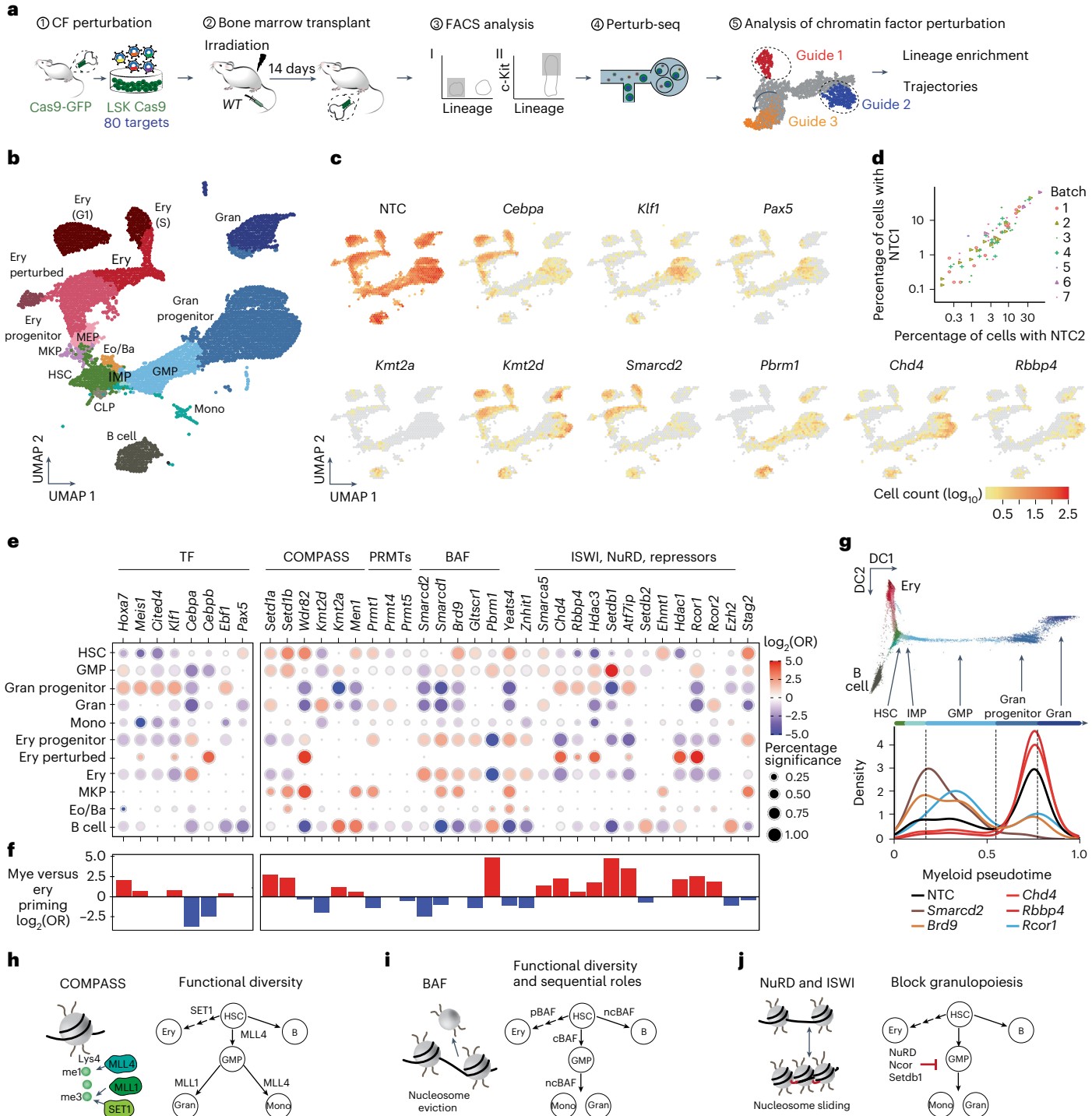

**Fig. 2 | Perturb-seq highlights disparate lineage dependencies for chromatin complexes during hematopoiesis. a,** Schematic drawing of the in vivo Perturb-seq: LSK progenitors were sorted from Cas9-GFP mice infected with the chromatin factor knockout library and transplanted into irradiated recipient mice; bone marrow was collected 14 days after transplant, sorted for an immature phenotype (lineage− and Lin+c-Kit+) and the Perturb-seq and downstream analyses were performed. **b,** Uniform manifold approximation and projection (UMAP) of the single-cell transcriptomes. Clusters are annotated using external reference maps[56]. The analysis integrates seven different biological replicates. CLP, common lymphoid progenitor; Eo/Ba, esosinophil-basophil progenitor; Ery, erythroblast; G1, G1 phase; Gran, granulocyte; IMP, immature myeloid progenitor; MEP, mega-erythroid progenitor; MKP, megakaryocyte progenitor; Mono, monocyte; S, S phase. **c,** UMAP showing the distribution of unperturbed

cells (NTC sgRNAs) and specific perturbations. **d,** Scatterplot showing comparisons between experimental batches. Each dot represents the abundance of two NTC sgRNAs in a given population. Pearson correlation between NTC and sgRNAs per experimental batch = 0.962 with $P = 1.20 \times 10^{-67}$. **e,** Enrichment and depletion of chromatin factor knockouts across hematopoietic populations (Supplementary Table 4). Dot color and size relate to the $\log_2$ odds ratio (OR) and the percentage of significant enrichments. **f,** Effect of specific chromatin factor knockouts on myeloid versus erythroid priming. Positive values (red) show enhanced myeloid priming. Negative values (blue) indicate reduced myeloid priming. **g,** Trajectory analysis of specific chromatin factor knockouts along myeloid differentiation. Cells are ordered from HSCs to mature granulocytes using pseudotime. DC1, diffusion component 1; DC2, diffusion component 2. **h–j,** Graphic representation of the roles of key chromatin regulatory complexes.

mega-erythroid, basophil and lymphoid (B cell), with most cells spanning the myeloid/erythroid branches (Fig. 2b and Extended Data Fig. 3a–c).

To rule out knockout-independent confounding patterns arising from neutral selection and amplification of individual LSK clones, we compared the distribution of 14 different nontargeting control (NTC) guides across hematopoietic lineages in seven separate experiments. Different NTCs demonstrated a homogeneous distribution with high correlation (Pearson correlation = 0.96) confirming a robust approach (Fig. 2c,d). Conversely, sgRNAs targeting the lineage-determining transcription factors *Cebpa* (myeloid), *Klf1* (erythroid) and *Pax5* (B cell) were absent from their cognate lineages, as expected (Fig. 2d). In line with the ex vivo bulk results, LOF of chromatin factors demonstrated strong lineage-specific patterns and marked functional diversity (Fig. 2e,f and Extended Data Fig. 3e). To characterize their roles, we used three metrics: (1) assessment of chromatin factor knockout enrichment and depletion across the different hematopoietic lineages; (2) analysis of the effect of chromatin factor knockouts at key lineage branching points: myeloid versus erythroid and monocyte versus granulocyte; and (3) analysis of the progression of chromatin factor knockouts along myeloid and erythroid differentiation trajectories using pseudotime analysis (Fig. 2f–h and Extended Data Figs. 4a–d and 5a).

Like ex vivo, we found that disruption of cohesin (*Stag2* knockout) blocked differentiation, causing accumulation of progenitors and myeloid deficiency (Fig. 2e). In line with previous studies[22], perturbation of the COMPASS H3K4 methyltransferases revealed marked functional diversity (Fig. 2e–h and Extended Data Fig. 4a–d). As reported[23] and like our ex vivo screen, SET1 catalytic subunits (*Setd1a*, *Setd1b*) were required for erythropoiesis and their perturbation eradicated erythroid differentiation, leading to myeloid and megakaryocytic progenitor accumulation. Perturbation of the SET1 structural subunit *Wdr82* partially phenocopied *Setd1a* and *Setd1B* knockouts, generating a marked accumulation of megakaryocyte progenitors. However, knockout also defined additional roles for *Wdr82* as a B cell and granulocyte regulator, suggesting that Wdr82 cooperates with different chromatin complexes during hematopoiesis. LOF of MLL1 (*Kmt2a* and *Men1* knockout) also blocked terminal granulocytic differentiation but further enhanced B cell priming. Finally, the H3K4 mono-methyltransferase MLL4 (*Kmt2d* knockout) functioned as a pleiotropic regulator of HSCs self-renewal, early myeloid branching and monocyte versus granulocyte specification. Collectively, these results demonstrate a lack of redundancy between H3K4 methyl writers during hematopoiesis.

Analysis of BAF perturbation patterns showed similar phenotypic diversity (Fig. 2e–g,i). Disruption of the canonical BAF (cBAF) member *Smarcd2* confirmed the strong myeloid dependency found ex vivo, inducing an early erythroid skewing and accelerated erythropoiesis, a similar pattern to the *Smarcd2* knockout mouse[24]. Alternatively, disruption of the noncanonical (ncBAF) complex, defined by *Brd9*, caused a major blockade of B cell development. Moreover, despite a mild myeloid priming defect for *Brd9* knockout, these cells did not undergo terminal myeloid differentiation but accumulated at the progenitor stages. In stark contrast, disruption of the polybromo-associated BAF (pBAF)-defining subunit *Pbrm1* prevented erythropoiesis, augmenting myeloid and B cell outputs. These results reveal very different roles for BAF subcomplexes during hematopoietic differentiation.

As predicted in the bulk screens, disruption of complexes with repressive functions, including NuRD (*Chd4* and *Rbbp4*), ISWI (*Smarca5*) and heterochromatin repressors (*Atf7ip*, *Setdb1*), produced a similar pattern characterized by accelerated granulocytic versus erythroid and B cell trajectories (Fig. 2e–g,h,j and Extended Data Fig. 5a). This suggests that most epigenetic repressors act to safeguard the diversity of progenitor identities[25,26] and, by limiting extensive myelopoiesis, ensure balanced lineage output. In addition, coRest (*Rcor1*, *Hdac1*) and NuRD (*Chd4*) complex repressive activity proved crucial for terminal erythropoiesis and its depletion induced the accumulation of aberrant

erythroid cells, which expressed high levels of both mature and progenitor markers and were rarely found in the unperturbed scenario (Extended Data Fig. 5b–f). Finally, disruption of *Kdm6a*, *Hmgbx4* and *Ash1l* led to other aberrant populations not present in the unperturbed state (Extended Data Fig. 5b–e).

Transcriptional analysis of chromatin factor perturbations provided a molecular basis for the lineage dependencies observed for BAF and SET/MLL members, where *Pbrm1* knockout and *Setd1a* knockout caused downregulation of erythroid regulators but *Smarcd2* knockout strongly reduced myeloid markers and transcription factors (Extended Data Fig. 6a). In addition, perturbations that blocked differentiation trajectories—*Rcor1*, *Wdr82*, *Stag2* and *Brd9* knockout—upregulated stem cell transcription factors (*Hoxa7*, *Meis1*) and surface markers (*Kit*, *Cd34*), highlighting that these alterations may facilitate leukemic transformation. By contrast, perturbations of chromatin repressors that show a clear myeloid bias did not significantly alter the balance between erythroid and granulocytic programs, suggesting that subtler and cumulative changes drive these phenotypes. Indeed, gene set enrichment analysis of these perturbations revealed a marked upregulation of inflammatory pathways (tumor necrosis-α or JAK/STAT) and Jun/Fos targets, mediators known to enhance myeloid lineage outputs under inflammatory stimuli[27] (Extended Data Fig. 6b).

Together, these results demonstrate functional diversity of chromatin factors in hematopoiesis similar to that of transcription factors. Interestingly and unlike transcription factors, most chromatin factor dependencies cannot be simply explained by their gene expression levels and patterns (Extended Data Fig. 5g), suggesting that other mechanisms like posttranscriptional regulation, protein complex assembly or differential recruitment explain their functional diversity.

## Chromatin factors regulate lineage-determining transcription factor accessibility

Intrigued by these findings, we sought to elucidate the interactions between chromatin factors and transcription factors that may explain the observed lineage-specific dependencies. To this end, we CRISPR-engineered the knockout of ten strong lineage-dependent chromatin factors in multipotent progenitors and induced both lineage priming and myeloid differentiation. Thereafter, we used an assay for transposase-accessible chromatin with sequencing (ATAC-seq) to perform transcription factor footprinting analysis[28] and define synergistic or antagonistic connections between chromatin factors and lineage-determining transcription factors that explain chromatin factor lineage dependencies (Fig. 3a–d and Extended Data Fig. 7a). In line with their in vivo effects, disruption of repressive factors (*Rbbp4*, *Hdac3* and *Setdb1*) resulted in increased accessibility of transcription factors like Cebps, Hlf[29] and AP-1 (ref. 30) that drive myelopoiesis on inflammation. This suggests that these repressive complexes attenuate a myeloid transcription factor-mediated program triggered by inflammatory cytokines. Conversely, myeloid-dependent chromatin factors (MLL4/Kmt2d, Smarcd2/cBAF and Wdr82) regulate chromatin accessibility around the binding sites of key myeloid-determining transcription factors (Cebp, Pu.1, Spib). Disruption of ncBAF (*Brd9* knockout) demonstrated milder effects on myeloid transcription factor binding site accessibility, in line with its subtler in vivo effects. Interestingly, we detected different degrees of specificity of chromatin factor–transcription factor connections, where MLL4 (*Kmt2d* knockout) regulated the accessibility of a more restricted set of myeloid transcription factors (Cebp, AP-1 and Hlf) but Wdr82 mediates the accessibility of a larger transcription factor repertoire. Finally, we interrogated the dynamics of these effects using a time series analysis in *Wdr82* and *Kmt2d* knockouts (Extended Data Fig. 7b–d). *Wdr82* knockout induced reduction in myeloid transcription factor accessibility at early time points (day 3), followed by increased mega-erythroid accessibility at later time points (days 5–7), suggesting that Wdr82 directly interacts with myeloid transcription factors and that its depletion indirectly enhances

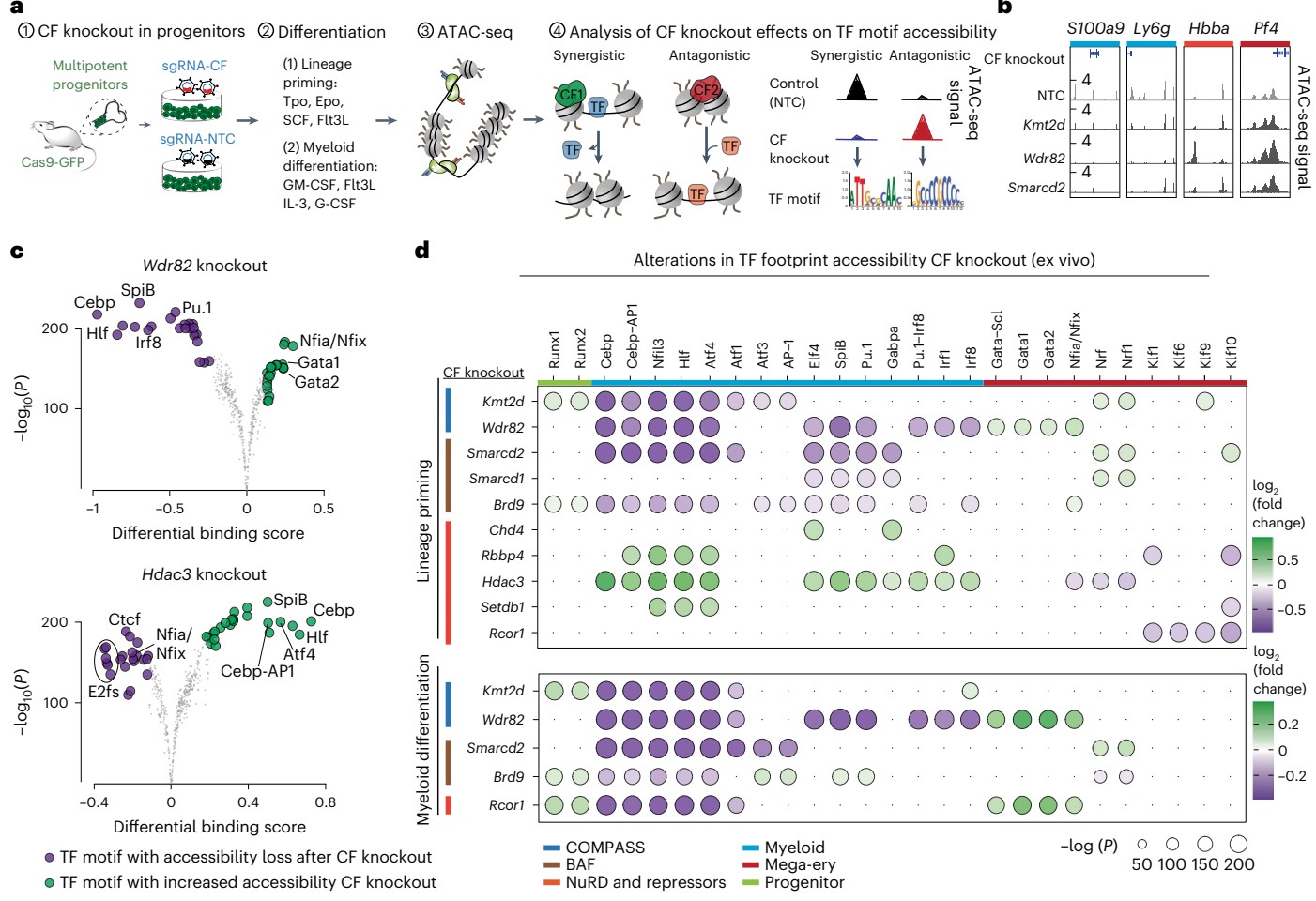

**Fig. 3 | Chromatin factors regulate the accessibility of lineage-determining transcription factors. a**, Schema of the experiment; LSK progenitors were collected from Cas9-GFP mice, transduced with a single chromatin factor or NTC control gRNA, placed into differentiation medium for 7 days and ATAC-seq was performed and analyzed to infer the interacting synergistic and antagonistic transcription factors. **b**, Chromatin accessibility for representative myeloid and mega-erythroid loci after disruption of specific chromatin factors. The accessibility profiles correspond to two merged replicate experiments. The *S100a9*, *Ly6g*, *Hbba* and *Pf4* loci coordinates are chr3:90685583–90703276, chr15:75134355–75144437, chr7:103869996–103874051 and chr5:90771031–

90773155, respectively. The *y* axis ranges for all knockouts in the same regions are 0.02–1.5, 0.02–1, 0.02–2 and 0.02–4, respectively. **c**, Volcano plot showing differentially bound transcription factor motifs (estimated by TOBIAS) in *Wdr82* and *Hdac3* knockouts under lineage priming conditions. Transcription factor motifs demonstrating gained and lost accessibility in each chromatin factor knockouts (compared to NTCs) are shown in green and purple, respectively. *n* = 2 biologically independent experiments. **d**, Heatmap summarizing the effect of ten chromatin factor knockouts on transcription factor motif footprints estimated by TOBIAS. *n* = 2 biologically independent experiments. Dot color and size relate to the log$_2$ fold change and the −log$_{10}$($P_{adj}$) value, respectively[28].

mega-erythroid transcription factors activity as a secondary effect. In general, we could demonstrate that the effects on TF-motif accessibility are not related to a decreased TF expression following CF-KO (Supplementary Fig. 6), suggesting that specific interactions between chromatin factors and lineage-determining TFs coordinately drive lineage differentiation.

## ncBAF is required for terminal myeloid differentiation

Ex vivo chromatin footprinting of BAF perturbation could not explain the sequential requirement of cBAF versus ncBAF complexes in, respectively, early myeloid priming and myeloid maturation. We speculated that this may reflect incomplete recapitulation of myeloid progression in our ex vivo system; therefore, we used an in vivo system to further dissect the roles of BAF subcomplexes in myelopoiesis (Fig. 4a). First, using Perturb-seq at a later time point (28 days), we detected a massive expansion of *Brd9* knockout clones that mapped predominantly to the multipotent (HSC) and myeloid progenitor (GMP) compartments (Fig. 4b,c). This confirmed that *Brd9* is required for full myeloid

maturation and revealed that *Brd9* disruption conferred a competitive growth advantage to progenitors.

Chromatin accessibility analysis of *Brd9* knockout myeloid progenitors (Fig. 4d) demonstrated an aberrant pattern, with a marked loss of accessibility at the myeloid maturation loci (*Mpo*, *S100a7*, *S100a8*, *S100a9* and *Ctsg*) and at the motifs of Cebp and AP-1, transcription factors mediating terminal myeloid maturation (Fig. 4e,f). By contrast, progenitor loci (*Hoxa7*, *Hoxa9*, *Hoxa10*, *Gata2*, *Meis1*) and progenitor-associated transcription factor motifs, including GATA2, demonstrated increased accessibility (Fig. 4e,f). ChIP–seq of Cebpa and Cebpe in freshly sorted GMPs validated the motif analysis derived from ATAC-seq (Fig. 4g–i), confirming that *Brd9* knockout leads to reduced accessibility at the binding sites of these two progranulocytic transcription factors.

Finally, corroborating these results, ChIP–seq analysis of Brd9 in myeloid progenitors (GMPs) and mature myeloid cells (monocytes) showed a strong enrichment for the Cebp and AP-1 (ATF4) motifs in mature myeloid cells and highlighted a switch in the transcription

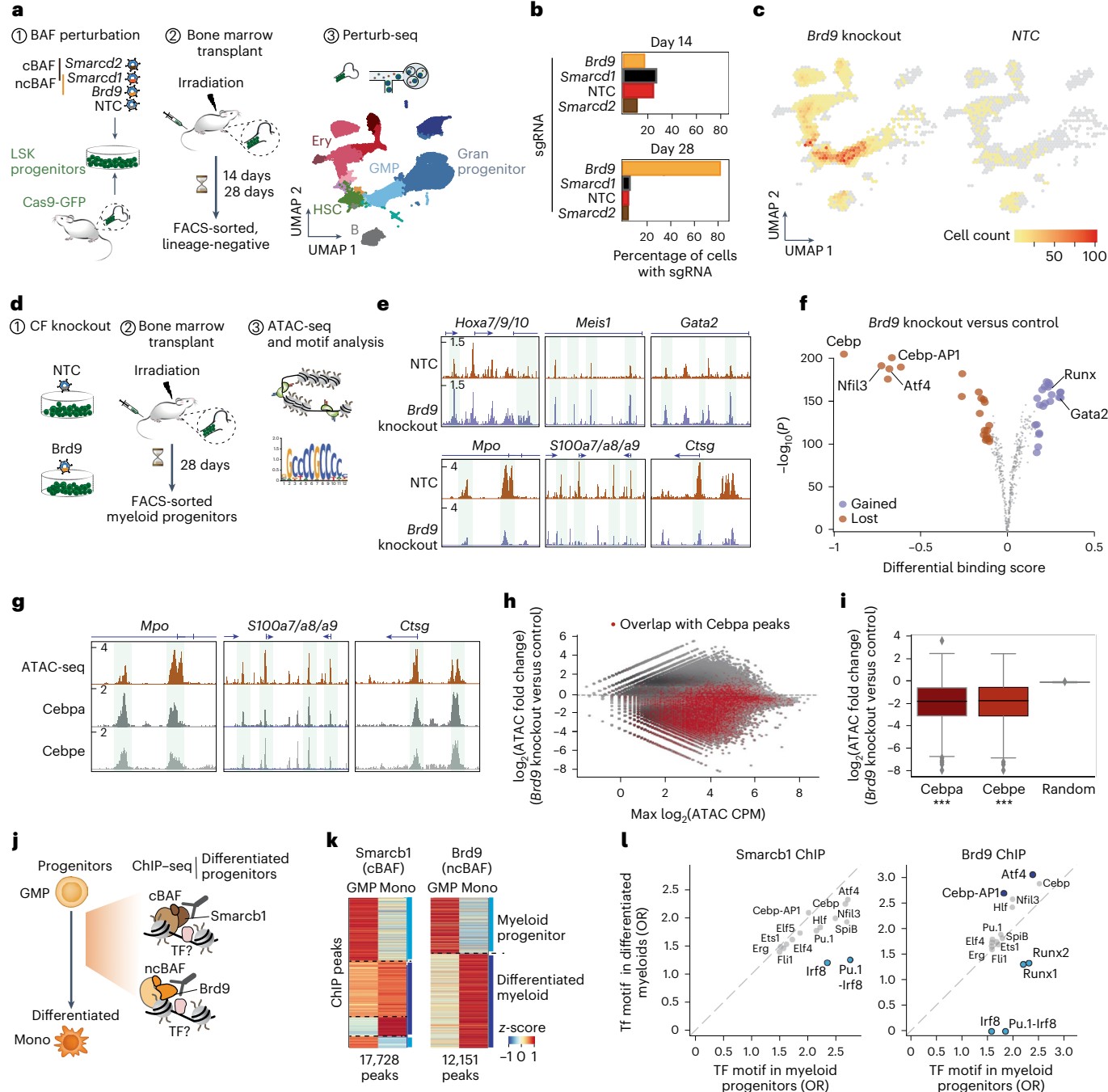

**Fig. 4 | Disruption of ncBAF leads to a pre-leukemic accumulation of myeloid progenitors with diminished Cebp-AP-1 activity. a**, Schema of the in vivo perturbation of cBAF (*Smarcd2* sgRNA) and ncBAF (*Brd9* sgRNA); in vivo transplantation of LSK cells transduced with a library of *Smarcd1*, *Smarcd2*, *Brd9* and NTC guides, into irradiated recipient mice and lineage-negative cells were collected and sorted at either 14 or 28 days for Perturb-seq as in Fig. 2. **b**, Proportions of cells with a specific sgRNA at 14 and 28 days after transplant. **c**, UMAP showing the distribution of *Brd9* knockout and control (NTC sgRNA) cells at 28 days after transplant. **d**, Schema of the in vivo ATAC-seq experiment; *Brd9* knockout LSK were generated and transplanted as in **a** and collected at day 28 for the ATAC-seq analysis. **e**, The ATAC-seq signal at myeloid progenitor (top) and differentiated (bottom) loci. Coordinates: *Hoxa7, Hoxa9, Hoxa10*: chr6:52214971–52236669, chr11:18912448–19036437; *Meis1*: chr6:88186822–8820729; *Gata2*: chr11:87788022–87796472; *Mpo*: chr3:90651905–90703284; *S100a7, S100a8, S100a9*: chr14:56098019–56107286; *Ctsg*: chr14:56098019–56107286. **f**, Volcano plot showing differentially bound transcription factor motifs (estimated by TOBIAS) between control (NTC) and *Brd9* knockout GMPs. *n* = 2 biologically independent experiments. **g**, Genome browser tracks

showing ATAC-seq, Cebpa ChIP–seq and Cebpe ChIP–seq in wild-type (WT) myeloid progenitors (GMPs). Loci coordinates are the same as in Fig. 4e. **h**,**i**, Quantification of accessibility changes between *Brd9* knockout and control (NTC) GMPs. **h**, MA plot showing loci overlapping with Cebpa binding (red). **i**, Box plots showing accessibility loss (statistically tested using a two-sided Kolmogorov–Smirnov test, *n* = 2) at Cebpa (*n* = 11,316, statistic = 0.86, *P* = 1 × 10⁻³²³) and Cebpe sites (*n* = 10,409, statistic = 0.85, *P* = 1 × 10⁻³²³). The box plot displays the median as the center line of the box, with the box representing the distribution's 25th (minima) and 75th (maxima) percentiles. The whiskers extend up to 1.5 times the interquartile range (IQR) (Q3–Q1) from the minima and maxima. **j**, Schema of the ChIP–seq analysis. **k**, Heatmaps showing specific binding of Smarcb1 (cBAF) and Brd9 (ncBAF) in myeloid progenitors (GMPs) and bone marrow monocytes. Two merged independent ChIP–seq experiments were used. **l**, Transcription factor motif enrichment measured by HOMER at the cBAF and ncBAF sites between progenitor (GMP) and mature (monocytes) myeloid cells. The axes represent the transcription factor motif odds-ratio (OR). All colored transcription factor motifs have *P*ₐ𝒹ⱼ < 0.001. The analysis was performed with two independent experiments per population.

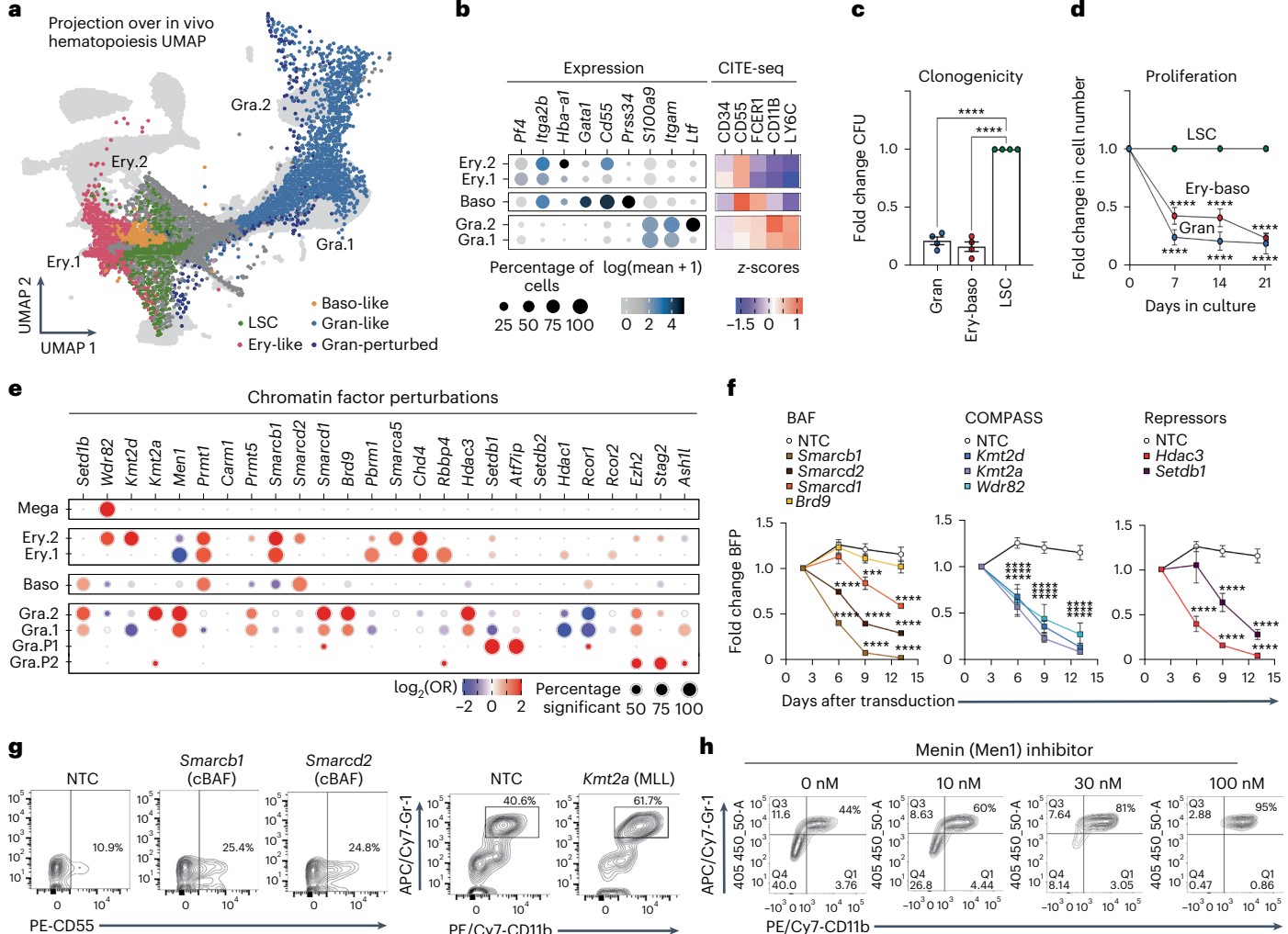

**Fig. 5 | *Npm1c* and *Flt3-ITD* leukemia abrogates normal chromatin factor function to maintain leukemic fitness through enforcing differentiation blockade. a**, UMAP projection of single-cell transcriptomes from *Npm1c* and *Flt3-ITD* primary leukemia. The color-coded clusters correspond to cells with specific signatures: leukemic stem cells (LSC, green), granulocyte-like (Gra.1 and Gra.2, Gran-perturbed, blue), erythroid-like (Ery.1 and Ery.2, red) and basophil-like (Baso-like, yellow). The analysis integrates datasets from six different Perturb-seq experiments. **b**, mRNA-derived and CITE-seq-derived expression of lineage makers in the differentiated leukemia subpopulations. **c**,**d**, Clonogenic (**c**) and proliferation (**d**) assays for differentiated leukemia subpopulations, isolated according to the strategy in Extended Data Fig. 9e. Colonies were counted after 7 days of culture in methylcellulose. Proliferation and clonogenic values were obtained from *n* = 4 biologically independent experiments. ****P < 0.0001 (two-way analysis of variance (ANOVA)). The error bars are the s.e.m., the midpoints show the mean. **e**, Enrichment analyses of specific chromatin factor knockouts across differentiated leukemia subpopulations. Dot color and size relate to the log₂(OR) and the percentage of significant enrichments versus NTCs, respectively. The analysis is based on measurements for two sgRNAs per chromatin factor target. All values are shown in Supplementary Table 6. Gra.P1 and Gra.P2, granulocyte-like. **f**, Perturbed growth curves for leukemic chromatin factor knockouts. The assay measures the change in the proportion of blue fluorescent protein (BFP) BFP sgRNA-expressing cells over time, *n* = 4 biologically independent experiments. All ***P < 0.001, except for Smarcd1 versus NTC (day 9) where ***P = 0.0009 (two-way ANOVA). The error bars are the s.e.m., the midpoints show the mean. **g**, FACS analysis of mega-erythroid (CD55) and myeloid (CD11b, Gr1) surface differentiation markers in leukemia cells depleted for cBAF (*Smarcb1* and *Smarcd2* knockout) and MLL (*Kmt2a* knockout) components. **h**, FACS analysis of myeloid surface differentiation markers (CD11b, Gr1) in leukemia cells treated with increasing doses of Men1 inhibitor (revumenib). Raw data can be found in Supplementary Data 1.

factor partnership of Brd9, from a broader spectrum in GMPs to a Cebp-AP-1-centric association in mature myeloid cells (Fig. 4j–l). We expanded this approach to other chromatin factor complexes and hematopoietic lineages to highlight lineage-specific transcription factor–chromatin factor interactions with regulatory potential; for example, a strong connection between Brd9 binding and the Ebf1 motif may explain the strong B cell dependency of ncBAF complex members (Extended Data Fig. 8a–e).

Together, these results demonstrate that specific transcription factor–chromatin factor interactions mediate lineage specification in vivo. In particular, myeloid maturation is governed by a transcription factor–chromatin factor switch, where ncBAF and Brd9 initially

complex with a broad range of transcription factor partners before specifically interacting with Cebp factors for terminal differentiation. Furthermore, *Brd9* loss induces a preleukemia-like phenotype of differentiation block and retention of a leukemia-associated transcriptional program, related, at least in part, to the failure to transition from 'progenitor transcription factor programs' to later differentiation programs.

## Chromatin factors enforce differentiation blockade in leukemia

Having extensively dissected chromatin factor function in normal hematopoiesis, we decided to explore their roles in the aberrantly

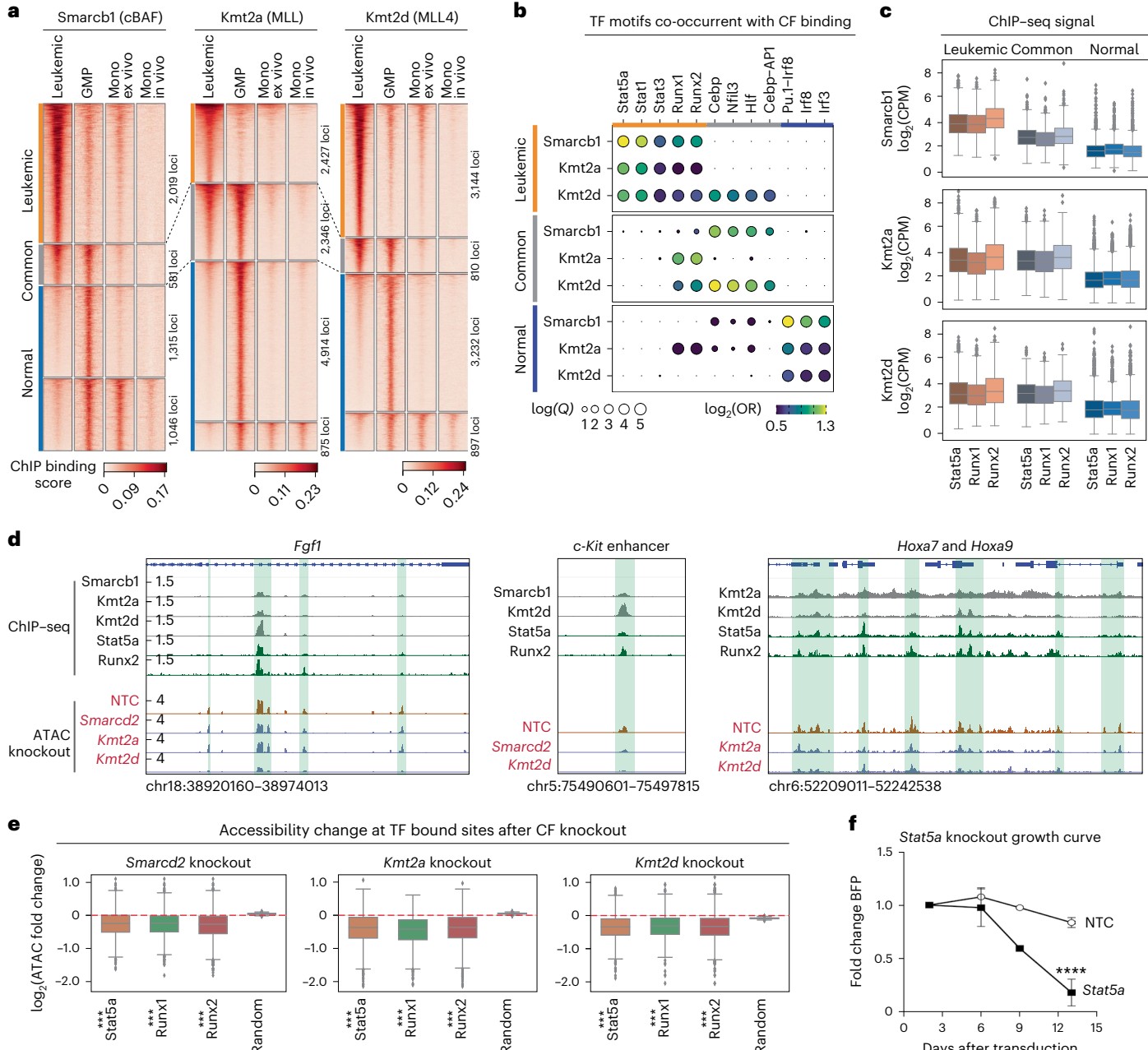

**Fig. 6 | Chromatin factors enforce differentiation blockade in AML through corrupted transcription factor interactions. a**, Heatmaps showing ChIP–seq signal for cBAF (Smarcb1), MLL (Kmt2a) and MLL4 (Kmt2d) in leukemia (*Npm1c* and *Flt3-ITD*), in vivo myeloid progenitors (GMPs), in vivo monocytes and ex vivo-derived primary monocytes. *n* = 2 independent ChIP–seq experiments per factor. **b**, Motif enrichment analysis of cBAF, MLL and MLL4 binding patterns specific for leukemia (leukemic), common between leukemia and myeloid progenitors (common), and specific for normal myeloid cells (normal). **c**, Box plot showing the Stat5a, Runx1 and Runx2 binding signal (ChIP–seq) at the leukemic, common and normal loci defined in Fig. 6a (*n* = 2). The number of loci comprising the leukemic, common and normal categories are 2,019, 581 and 2,361 for Smarcb1; 2,427, 2,346 and 5,789 for Kmt2a; and 3,144, 810 and 4,129 for Kmt2d. **d**, Genome browser tracks showing the ChIP–seq signal for Smarcb1, Kmt2a, Kmt2d, Stat5a and Runx2 in leukemia, and ATAC–seq for control and chromatin factor-depleted leukemia cells. The chosen loci are leukemic-specific. *n* = 2 independent

experiments. The green highlighted regions shown identify chromatin factor–transcription factor binding and altered accessibility on chromatin factor knockout. **e**, Box plots showing changes in chromatin accessibility at leukemic loci bound by Stat5a, Runx1 and Runx2 on depleting specific chromatin factors. cBAF, *n* = 2 independent experiments. *Smarcd2* knockout: *n* = 1,385, 1,050, 1,711; statistic = 0.73, 0.75, 0.76; *P* = 5 × 10⁻¹⁵, 0, 0. *Kmt2a* knockout: *n* = 1,288, 695, 1,427; statistic = 0.78, 0.86, 0.78; *P* = 0, 9 × 10⁻¹⁶, 0. *Kmt2d* knockout: *n* = 1,482, 990, 1,913; statistic = 0.70, 0.70, 0.69; *P* = 2 × 10⁻¹⁵, 0, 0. The decay in accessibility was tested statistically using a two-sided Kolmogorov–Smirnov test. **f**, Growth curves for Cas9-leukemic cells expressing NTC and anti-Stat5a sgRNAs, *n* = 3 independent experiments. ***P* < 0.001 (two-way ANOVA). The error bars are the s.e.m., the midpoints show the mean. The box plots in **c** and **e** display the median and the distribution's 25th (minima) and 75th (maxima) percentiles. The whiskers extend up to 1.5 times the IQR (Q3–Q1) from the minima and maxima.

blocked differentiation states typical of leukemia. To this end, we chose an aggressive *Npm1c* and *Flt3-ITD* model, driven by the two most common co-occurring mutations in AML that synergize to generate a highly

corrupted chromatin landscape, which recapitulates many aspects of human AML with the same genotype[31]. We isolated primary leukemia cells from *Npm1c*, *Flt3-ITD* Cas9 mice, cultured them and used cellular

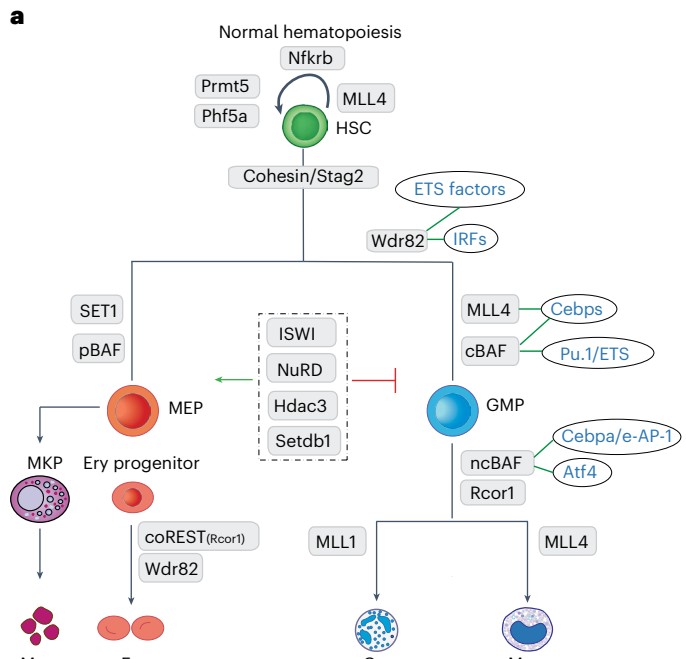

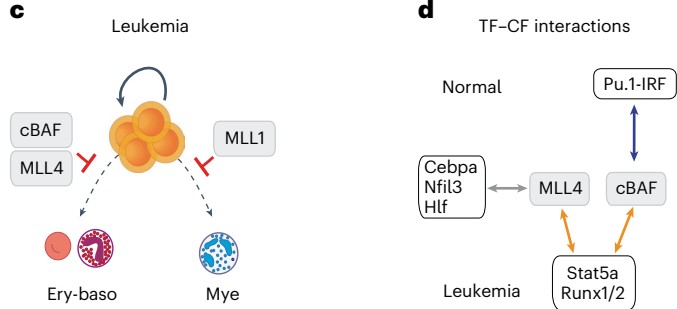

**Fig. 7 | Summary model of chromatin factor function and chromatin factor–transcription factor interactions in normal and malignant hematopoiesis.** **a**, Roadmap of chromatin factor requirements for major hematopoietic cell fate decisions, identifying individual chromatin factors required for specific lineages and the transcription factor families they interact with to orchestrate these decisions. **b**, Table explaining the roles of specific chromatin factor– transcription factor complexes. **c**, Examples of how chromatin factor function is hijacked in leukemia, where cBAF-, MLL4- and MLL1-containing complexes block rather than facilitate hematopoietic differentiation. **d**, Examples of 'transcription factor switches' that mechanistically underpin the different functions of chromatin factors in normal and malignant hematopoiesis.

indexing of transcriptomes and epitopes sequencing (CITE-seq) to interrogate their differentiation status (Fig. 5a,b). This identified a core of leukemia stem cell-like cells with high levels of stem cell transcripts (*Bcat1*, *Sox4*) and markers (Cd34) and also subpopulations with more differentiated transcriptomes resembling granulocytic, erythroid, basophilic and megakaryocytic states, which demonstrated decreased fitness when isolated and grown in liquid cultures and clonogenic assays (Fig. 5c,d and Extended Data Fig. 9a–e).

Importantly, Perturb-seq analysis of 50 chromatin factor knockouts uncovered latent trajectories toward such differentiated endpoints (Fig. 5e and Extended Data Fig. 9f). Specifically, the depletion of MLL1-COMPASS (*Kmt2a* and *Men1* knockouts), NCoR (*Hdac3* knockout), ncBAF (*Brd9* and *Smarcd1*-knockouts), *Prmt5* and the heterochromatin regulators *Setdb1* and *Atf7ip* induced transition toward several granulocytic states. In contrast, LOF of cBAF (*Smarcd2* and *Smarcb1* knockouts), MLL4-COMPASS (*Kmt2d* knockout) and *Prmt1* pushed leukemia toward basophil and erythroid fates. Interestingly, unlike their roles in normal hematopoiesis, disruption of the repressive factors *Smarca5*, *Chd4* and *Rbbp4* induced the same erythroid trajectory. Finally, like the normal setting, LOF of *Wdr82* generated a megakaryocytic state, which was absent from the unperturbed scenario (Extended Data Fig. 9g).

Analysis of the growth dynamics of several individual chromatin factor knockouts revealed a marked defect in cell growth, indicating that the differentiation caused by their depletion leads to leukemic exhaustion (Fig. 5e,f). This was especially pronounced for cBAF and COMPASS members, revealing *Npm1c* and *Flt3-ITD* leukemia to be highly dependent on the epigenetic activities regulated by these complexes. However, of interest and unlike previous reports for MLL-driven leukemias[32,33], our *Npm1c* and *Flt3-ITD* model did not show vulnerability to *Brd9* (ncBAF) disruption, highlighting that different leukemia mutations produce specific chromatin states that are variably dependent on individual chromatin factors.

Of note, our analysis found potential therapeutic targets in *Npm1c* and *Flt3ITD* leukemia, including *Prmt1* (Fig. 5e and Extended Data Fig. 9h), which may be amenable to therapeutic exploitation. As a proof of principle for the therapeutic implications of our approach, treatment of the cells with the clinical grade menin inhibitor (revumenib, previously known as SNDX-5613), which is currently producing promising results in a clinical trial in *KMT2A*-mutated and *NPM1*-mutated AML (AUGMENT-01; ClinicalTrials.gov registration: NCT04065399) (ref. 34), recapitulated the *Men1* and *Kmt2a* knockout single-cell phenotype to induce a dose-dependent granulocytic differentiation and decrease in proliferation (Fig. 5g,h and Supplementary Fig. 7).

These collective findings demonstrate how leukemias hijack chromatin factors involved in homeostatic differentiation to aberrantly block latent differentiation pathways and how this can be therapeutically exploited to facilitate leukemia exhaustion.

## Chromatin factors engage in corrupted transcription factor interactions in leukemia

Finally, to interrogate the molecular mechanisms that underpin the requirement for the cBAF and COMPASS–MLL complexes in *Npm1c* and *Flt3-ITD* AML, and how these differ from normal hematopoiesis, we compared the genome-wide binding patterns of Smarcb1 (an exemplar of cBAF), Kmt2a (COMPASS-MLL1) and Kmt2d (COMPASS-MLL4) using ChIP–seq across leukemia, normal myeloid progenitors (GMP) and mature myeloid subsets (Fig. 6a). Of note, these analyses demonstrated marked redistribution of the cBAF and COMPASS–MLL1/MLL4 complexes on leukemia induction (Fig. 6a and Extended Data Fig. 10b–d), identifying three major binding patterns: (1) leukemic-specific, enriched in molecular functions such as tyrosine kinase signaling related to the *Flt3-ITD* mutation; (2) common to leukemia and myeloid progenitors; and (3) normal-specific.

Motif analysis across the three binding patterns revealed subverted leukemic transcription factor–chromatin factor interactions (Fig. 6b), specifically between Stat and Runx transcription factors, and MLL1 and MLL4 and cBAF complexes. In contrast, Pu.1 and IRF factors were associated with cBAF specifically in normal cells. ChIP–seq profiling of Stat5a, Runx1 and Runx2 confirmed the motif analysis, demonstrating cobinding of these transcription factors and BAF or MLL1 and MLL4 complexes at key pro-leukemia genes including *Hoxa7* and *Hoxa9* (Fig. 6c,d and Extended Data Fig. 10e). Moreover, chromatin accessibility profiling of leukemia cells disrupted for cBAF (*Smarcd2* knockout), MLL1 (*Kmt2a* knockout) or MLL4 (*Kmt2d* knockout) showed that these chromatin factors are required to maintain optimal accessibility at the Stat5a and Runx2 binding loci (Fig. 6d,e and Extended Data Fig. 10f). Finally, LOF of *Stat5a* in *Npm1c* and *Flt3-ITD* cells significantly reduced their proliferative fitness, confirming the importance of these transcription factor–chromatin factor switches for leukemic maintenance (Fig. 6f). Taken together, these findings demonstrate how individual chromatin factors required for normal myeloid lineage determination engage in corrupted interactions with alternative transcription factor partners to promote differentiation blockade, thereby maintaining cellular fitness in AML.

## Discussion

In this study, we generated a detailed lineage dependency map for chromatin factors during hematopoiesis (Fig. 7a,b). We uncovered a remarkable phenotypic diversity that, for some chromatin factors, phenocopies key lineage-determining transcription factor functions (i.e. Smarcd2/Cebpa or Pbrm1/Klf1). Demonstrating the complex nature of chromatin factor regulation, we showed highly divergent roles for complexes that mediate the same, or very similar, epigenetic activities, including the different COMPASS H3K4 methyltransferases or the various BAF subcomplexes. The lack of redundancy among COMPASS complexes has been described in other systems[35], suggesting lineage-specific requirements for H3K4 methylation deposition by particular COMPASS members or regulation via catalytic-independent roles[36,37]. In addition, reshaping of BAF complexes regulates cellular fates in pancreatic B cells[38], and here we show evidence for another switch, from cBAF to ncBAF, which regulates myeloid differentiation, ensuring full lineage progression. Of note, *Brd9* and ncBAF perturbation led to the accumulation of myeloid progenitors with a preleukemic gene expression program, mimicking the aberrant splicing of *BRD9* that results in its degradation, a process mechanistically implicated in the AML precursor lesion, myelodysplastic syndrome[39]. Lastly, in stark contrast to functional diversity for some chromatin factors, we observed a common function for different chromatin repressors as attenuators of excessive granulopoiesis, suggesting repressive chromatin factors as a key buffering mechanism in the interplay between inflammatory signaling and chromatin state.

What then underlies such chromatin factor specificity? Inspired by previous studies[40], we demonstrated that specific transcription factor–chromatin factor interactions mediate lineage diversification via the regulation of local accessibility and thus the binding site specificity of key lineage-determining transcription factors. However, as the chromatin factors investigated include a large number of proteins with diverse functions (remodelers, epigenetic readers, epigenetic writers, epigenetic erasers), we think it unlikely that a simple 'one-size-fits-all' mechanism governs chromatin factor–transcription factor interactions. We believe it more probable that the interactions are usually directed by transcription factors, which physically recruit specific chromatin factors to induce lineage-specific chromatin configurations[41–45]. However, an alternative and non-mutually exclusive explanation could be a sequential model, where specific chromatin factors, already deployed through multivalent interactions with epigenetic modifications, regulate subsequent transcription factor activity by modulating the chromatin state at the transcription factor binding sites. Specifically, we propose this as a possible mechanism whereby chromatin repressors attenuate myeloid pro-inflammatory transcription factor responses.

Regardless of the specific detail, central to the chromatin factor–transcription factor collaboration is the chromatin factor-mediated regulation and maintenance of locus accessibility and thus transcription factor binding[46]. As evidenced in our dynamic ATAC-seq studies, early alterations in accessibility are observed even after knockout of methyltransferase complex components (*Kmt2d* and *Wdr82*) that lack chromatin remodeling activity. Therefore, these data also inform interactions among chromatin factors with regulatory potential; the changes in accessibility suggest that the MLL4 and SET1 complexes recruit chromatin remodelers[47], probably BAF members, an observation reinforced by the strong knockout phenocopy observed between *Kmt2d* and several cBAF subunits. However, whether this recruitment involves direct protein–protein interactions, or via an indirect mechanism mediated by a local pattern of histone methylation that, in turn, recruits remodelers via their own reader modules[48], will require further investigation.

Finally, by studying chromatin factor requirement in AML, we highlight corrupted roles for certain chromatin factors in malignant hematopoiesis. Here, they alter their normal lineage regulatory role to conversely block differentiation in leukemia, reiterating that this blockade is an active process required for leukemic fitness (Fig. 7c). In characterizing these patterns, we also identified leukemic dependencies that may be amenable to inhibition or targeting, including *Prmt1*, *Hdac3*, *Setdb1* and *Kmt2d*. Furthermore, we identified that these altered roles relate to leukemia-specific transcription factor–chromatin factor interactions, including COMPASS and BAF factors that rewire their transcription factor networks toward Runx and Stat transcription factors (Fig. 7d). These observations also have clinical implications; transcription factors and chromatin factors have pleiotropic effects across multiple tissues; however, targeting leukemia-specific transcription factor–chromatin factor interactions, important only for leukemia cells, will likely have much lower toxicity and higher specificity. This can be achieved chemically or through synthetic approaches[49,50] and requires not only a detailed structural understanding of specific interactions, but also knowledge of the mechanisms governing individual transcription factor–chromatin factor associations[51]. Our study provides a blueprint to expand such approaches to leukemia.

Our approach combining large-scale CRISPR screening with downstream single-cell analysis in vivo could be readily deployed to assess the role of other classes of proteins in hematopoiesis or adapted to other organ and tumor systems. However, several limitations must be considered. First, our currently limited ex vivo differentiation readout could be supplemented by the use of other cytokine cocktails that permit interrogation of lymphoid lineages[52]. In addition, our in vivo approach does not completely reflect steady-state hematopoiesis, but more regenerative hematopoiesis in the after transplant setting. Thus, some of the roles described for individual chromatin factors, including the blockade of granulopoiesis by several repressors, may differ under steady-state conditions. Screening in homeostasis could be achieved by combining inducible Cas9 systems[53–55] that permit inducible LOF of specific factors in steady-state conditions, after transplantation full reconstitution and a return to homeostatic hematopoiesis.

Taken together, the results of this study show that chromatin factors constitute a specific regulatory layer that should be accorded equal weighting with transcription factors when studying cell fate decisions. It lays the basis for additional, in-depth interrogation of specific chromatin factor–transcription factor interactions and functions, using multidisciplinary approaches ranging from in vivo functional approaches to protein–protein interactions, which we feel are warranted to further elucidate chromatin factor–transcription factor functions.

## Online content

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

## Methods

### Mouse models

C57BL/6J (strain 000664, The Jackson Laboratory) and B6J.129(Cg)-$Gt(ROSA)26Sor^{tm1.1(CAG-cas9*/-EGFP)Fezh}$/J (strain no. 026179, The Jackson Laboratory) were used for all experimental procedures. The $Npm1c$/$Flt3$-$ITD$/Cas9 model has been extensively described previously[31,57,58]. The maximal tumor size allowed by the Home Office license for this project and authorized by the Animal Welfare Ethical Review Body at the University of Cambridge is 1.2 cm in diameter; however, as the animals developed a liquid not a solid tumor, in none of the experiments was this tumor size exceeded. Housing conditions were: 12h–12h dark–light cycle, at a temperature of 21 ± 1 °C and 40% humidity.

### Bulk CRISPR screens

**CRISPR library construction.** sgRNA-CRISPR libraries (Supplementary Table 2) were ordered from Integrated DNA Technologies and cloned using a Gibson assembly mastermix (New England Biolabs) in the CRISPR sequencing (CRISP-seq) backbone (catalog no. 85707, Addgene).

**Cloning of individual sgRNAs.** Individual oligonucleotides were cloned in the CRISP-seq backbone (Supplementary Table 2) using a Golden-Gate reaction with a 100-ng vector backbone, 1 µl annealed sgRNA oligonucleotides, 1 µl Esp3I (New England Biolabs) and 1 µl T4 DNA Ligase (New England Biolabs) using the following program: 10× (5 min at 37 °C, 10 min at 22 °C), 30 min at 37 °C and 15 min at 75 °C. Individual colonies were picked and grown in lysogeny broth and ampicillin overnight. Plasmids were isolated with the ZymoPURE MiniPrep Kit (Zymo Research) and sequenced using a U6 forward primer (Supplementary Table 8).

**Lentiviral production.** HEK 293T (catalog no. 12022001-DNA-5UG, Sigma-Aldrich) cells were transfected with the CRISP-seq vectors, pMD2-G (plasmid no. 12259, Addgene) and psPAX2 (plasmid no. 12260, Addgene) using Lipofectamine 3000 according to the manufacturer's protocol. After 10 h, media was replaced with Opti-MEM (Thermo Fisher Scientific) plus 1% penicillin-streptomycin (Thermo Fisher Scientific). The viral supernatant was collected 48 h after transfection, filtered using 0.45-µM filters and concentrated with a 100 Kda Centricon at 3,000$g$ at 4 °C.

**Isolation of murine hematopoietic progenitors (LSKs) from bone marrow.** Femur, tibia, ileum, humerus, sternum and scapula were collected from 12–14-week-old C57BL/6J and $ROSA$-Cas9 mice (equal ratio of males and females), crushed in cold autoMACS Running Buffer and filtered through a 70-µM strainer. Erythrocytes were lysed and c-Kit$^+$ cells were enriched using mouse CD117 magnetic beads (Miltenyi Biotec). The c-Kit-enriched fraction was stained with anti-lineage (B220, CD3, CD11b, Gr1, Ter-119), anti-CD117 (c-Kit) and anti-Sca1 (Supplementary Table 9). LSK cells were FACS-sorted in 1 ml DMEM/F-12 (Thermo Fisher Scientific) + 1× penicillin-streptomycin, centrifuged at 350$g$ for 10 min and processed for ex vivo screens (see below). Dilution for all antibodies was 1:100, except 1:50 for CD34. An exemplar gating strategy can be found in Supplementary Fig. 1.

**Ex vivo CRISPR screens cultures.** After FACS sorting, multipotent (LSK) or myeloid (GMP) progenitors were resuspended at 250 cells per µl in DMEM/F-12 plus 1% penicillin-streptomycin-glutamine (Gibco), polyvinyl alcohol (PVA) 87% hydrolyzed (P8136, 363081 or 363146), 1× insulin-transferrin-selenium-ethanolamine (Gibco), 1× HEPES (Gibco), 100 ng ml$^{-1}$ mouse TPO (PeproTech) and 10 ng ml$^{-1}$ mouse SCF (PeproTech) and plated in 96-well plates with 25,000 cells (100 µl) per well. Immediately after plating, cells were transduced with the CRISPR libraries to reach a multiplicity of infection of approximately 20%. After 12 h, 2.5 volumes of fresh medium were added. Then, 48 h

after infection, cells were transferred (1,000 cells per ml) to the 'screen media' and cultured.

**Stem cell versus differentiation.** Growth time was 7 days. The medium was complete DMEM/F-12, 1% penicillin-streptomycin-glutamine (Gibco), PVA 87% hydrolyzed (P8136, 363081 or 363146), 1× insulin-transferrin-selenium-ethanolamine (Gibco), 1× HEPES (Gibco), 100 ng ml$^{-1}$ mouse TPO (PeproTech) and 10 ng ml$^{-1}$ mouse SCF (PeproTech)[59].

**Lineage priming.** Growth time was 5 days. The medium was complete DMEM/F-12, 1% penicillin-streptomycin-glutamine (Gibco), PVA 87% hydrolyzed (P8136, 363081 or 363146), 1× insulin-transferrin-selenium-ethanolamine (Gibco), 1× HEPES (Gibco), 100 ng ml$^{-1}$ mouse TPO (PeproTech) and 10 ng ml$^{-1}$ mouse SCF (PeproTech) + 1 ng ml$^{-1}$ mouse Flt3L (PeproTech) and 1 U ml$^{-1}$ Epo (R&D Systems).

**Myeloid differentiation.** Growth time was 4 days. The medium was IMDM (Thermo Fisher Scientific), 20% FCS (Thermo Fisher Scientific), 1% penicillin-streptomycin-glutamine (Gibco), 10 ng ml$^{-1}$ mouse GM-CSF (PeproTech), 10 ng ml$^{-1}$ mouse SCF (PeproTech), 5 ng ml$^{-1}$ mouse G-CSF (PeproTech), 5 ng ml$^{-1}$ mouse interleukin-3 (IL-3) (PeproTech), 5 ng ml$^{-1}$ mouse interleukin-6 (IL-6) (PeproTech), 5 ng ml$^{-1}$ mouse interleukin-5 (IL-5) (PeproTech), 5 ng ml$^{-1}$ mouse Flt3L (PeproTech), 2 ng ml$^{-1}$ mouse TPO (PeproTech) and 2 U ml$^{-1}$ Epo (R&D Systems).

**Terminal myeloid differentiation.** Growth time was 2 days. The medium was IMDM, 20% FCS, 1% penicillin-streptomycin-glutamine (Gibco), 10 ng ml$^{-1}$ mouse GM-CSF (PeproTech), 10 ng ml$^{-1}$ mouse SCF (PeproTech), 5 ng ml$^{-1}$ mouse G-CSF (PeproTech), 5 ng ml$^{-1}$ mouse IL-3 (PeproTech), 5 ng ml$^{-1}$ mouse IL-6 (PeproTech), 5 ng ml$^{-1}$ mouse IL-5 (PeproTech), 5 ng ml$^{-1}$ mouse Flt3L (PeproTech).

**CRISPR FACS readouts.** Cultures were collected and stained with the readout-specific antibody cocktails (below and Supplementary Table 9) plus a viability marker (TOPRO or propidium iodide). Cas9 (GFP$^+$) and non-Cas9 (GFP$^-$) populations were sorted in 1.5 ml PBS + 0.1% BSA (Thermo Fisher Scientific).

**Stem cell differentiation.** Multipotent progenitor (Lin$^-$c-Kit$^+$Sca-1$^+$). Differentiated (Lin$^-$c-Kit$^+$Sca-1$^-$).

**Lineage priming.** Myeloid progenitor (Lin$^-$c-Kit$^+$Sca-1$^+$FcγRIII$^+$). Mega-erythroid progenitor (Lin$^-$c-Kit$^+$Sca-1$^+$FcγRIII$^+$).

**Myeloid differentiation.** Mature myeloid (FcγRIII$^+$CD11b$^+$). Non-myeloid (FcγRIII$^-$CD11b$^-$).

**Terminal myeloid differentiation.** Mature myeloid (CD11b$^+$Gr1$^+$). Immature myeloid (CD11b$^-$Gr1$^-$). A dilution of 1:100 was used for all antibodies. Exemplar gating strategies can be found in Supplementary Figs. 2 and 3.

**Bulk CRISPR library preparation, sequencing and preprocessing.** Sorted cells were lysed in 40 µl of 0.2% SDS and 2 µl of proteinase K (New England Biolabs) at 42 °C for 30 min. Then, genomic DNA (gDNA) was isolated with a 2× solid-phase reversible immobilization (SPRI) cleanup and NGS libraries were prepared from purified gDNA with a two-step PCR protocol using 2× KAPA HiFi Master Mix (Roche): first PCR: 10 µM Read1-U6 and Read2 scaffold primer mix (Supplementary Table 2); 3 min at 98 °C; 20× (10 s at 98 °C, 10 s at 62 °C, 25 s at 72 °C); 2 min at 72 °C. Second PCR: 10 µM P5 and P7 index mix: 3 min at 98 °C; 10× (10 s at 98 °C, 10 s at 62 °C, 25 s at 72 °C); 2 min at 72 °C.

Libraries were purified with 1× SPRI cleanup and sequenced at 10 M reads per sample (paired-end 50 bp in a NextSeq 1000 system). Raw data were processed with bcl2fastq (v.2.20) into FASTQ files and then processed using a custom script (see 00_NR_CRISPR_extract.pi in the analysis code)[60] to isolate the 20-mer protospacers; then, they were mapped using Bowtie2 (v.2.3.4.2) using an index file containing the sgRNA sequences (Supplementary Table 2). Raw counts can be found in Supplementary Data 2.

**Computational analysis of FACS-based CRISPR screens.** Analyses were performed in R (v.4.0.2). Biological replicates were merged by summing the counts. Aggregated counts were normalized by calculating normalizing factors using the function calcNormFactors from edgeR (v.3.32.1) (ref. [61]) on nontargeting guide counts. Counts were transformed to counts per million (CPM) and $\log_2$-normalized using limma (v.3.46.0) (ref. [62]). A raw lineage score comparing pairs of populations (A and B) was then calculated by subtracting the $\log_2$ CPMs of population A from population B, for each library of guides. To assess significance, we next calculated the probability of observing a given score in the Cas9 data given the non-Cas9 data, where no effective knockout occurs. For each comparison and each library, we centered and scaled the Cas9 data based on the mean and standard deviation calculated from the non-Cas9 data. The resulting normalized scores were used to calculate the probabilities of observing values as extreme (two-sided) using the function pnorm. The resulting probabilities represent the probability of an observed value given a background distribution but with the important difference to $P$ values that in our analyses the background distribution was not based on replicates but on the non-Cas9 data. We next corrected these probabilities for multiple testing using the function p.adjust with the Benjamini–Hochberg method and selected values smaller than 0.05 as significant.

**Validation of single candidates with flow cytometry.** Cas9 progenitor cells (LSKs) were transduced with CRISPR sequencing lentiviral vectors, cultured and stained using the conditions described above, and analyzed with a FACSAria. FACS data were analyzed with FlowJo v.10.8.0 (FlowJo LLC).

**Perturb-seq**
**Perturb-seq libraries.** For each target, we cloned the top two performing sgRNAs in the lenti-Perturb-seq-BFP vector, which we built by modifying the original lenti-CRISPR-BFP vector by replacing the original sgRNA scaffold for a sgRNA scaffold containing the 10× capture-sequenced CR1Cs1 (ref. [63]). Lentiviral particles were prepared as specified for bulk screens.

**In vivo Perturb-seq.** We performed seven experiments with 10–15 factors and two nontargeting control sgRNAs per batch. In each batch, 300,000 LSKs were isolated from 12–14-week-old *ROSA26*-Cas9 mice (equal ratio of males and females), and transduced with the Perturb-seq library to reach 10% infection (>1,000× coverage). After transduction, cells were left to recover for 36 h in stem cell medium: DMEM/F-12 plus 1% penicillin-streptomycin-glutamine (Gibco), PVA 87% hydrolyzed (P8136, 363081 or 363146), 1× insulin-transferrin-selenium-ethanolamine (Gibco), 1× HEPES (Gibco), 100 ng ml⁻¹ mouse TPO (PeproTech) and 10 ng ml⁻¹ mouse SCF (PeproTech)[59]. Then, cell number and viability were assessed with the Cellometer K2 Image Cytometer (Nexcelom Bioscience) and 50,000 viable cells were transplanted to each irradiated (902 cGy, 1 min) 12-week-old adult B6.SJL-*Ptprc^a Pepc^b*/BoyJ (CD45.1) mice (strain no. 002014, The Jackson Laboratory) via tail injection.

After 2 weeks, mice were euthanized, c-Kit⁺ cells were isolated and stained with TOPRO (viability), anti-lineage (CD3, CD19, Ter119, CD11b, Gr1) and anti-CD117 (c-Kit) antibodies (1:100 dilution). Then, we gated GFP⁺ (Cas9) and BFP⁺ (sgRNA) cells, and FACS-sorted lineage⁻ and lineage⁺c-Kit⁺ fractions. Cells from each of these gates were processed

in the Chromium Controller to reach 500 cells per sgRNA. An exemplar gating strategy can be found in Supplementary Fig. 4.

**Perturb-seq in leukemia.** A total of $0.25–0.5 \times 10^6$ double-mutant (DM) Cas9 cells were transduced with Perturb-seq libraries using retronectin-mediated infection (Takara Bio) and maintained in culture for 6 days. Transduced, BFP⁺ and 7-AAD⁻ (BD Biosciences) live cells were FACS-sorted (BD Influx, BD Biosciences). Finally, 16,000 live cells (cell number and viability assessed with the Cellometer K2 Image Cytometer) were processed in a 10× scRNA-seq partition aiming at a final coverage of 500 single cells per sgRNA.

**Perturb-seq library preparation.** Single-cell libraries were generated using the Chromium Next GEM Single Cell 3′ Reagent Kits v.3.1 (Dual Index) using the manufacturer's recommended protocol. The resulting libraries were sequenced in a NovaSeq system to a final coverage of 50,000 reads per cell for 3′ Gene Expression libraries and 5,000 reads per cell for CRISPR Feature Barcode libraries.

**CITE-seq**
CITE-seq was performed on $2 \times 10^6$ DM Cas9 murine leukemic cells, stained with TotalSeq-B antibodies (BioLegend) for CD11b, Ly6C, CD115, CD14, CD150, CD48, CD34, CD117, CD55, CD41, CD326 and FcγRI (Supplementary Table 9) according to the manufacturer's protocol. Stained cells were FACS-sorted for 7-AAD⁻ (BD Biosciences) live cells (BD Influx; BD Biosciences).

scRNA-seq libraries were prepared at the Cancer Research UK Cambridge Institute Genomics Core Facility using the Chromium Single Cell 3′ Library & Gel Bead Kit v.3.1, Chromium Chip G Kit and Chromium Single Cell 3′ Reagent Kits v.3.1 User Guide (part no. CG000317 for CITE-seq). Libraries were sequenced in a NovaSeq system with a final coverage of 50,000 reads per cell for 3′ gene expression libraries and 5,000 reads per cell for antibody Feature Barcode libraries.

**Perturb-seq and CITE-seq analysis**
Analyses were performed in R (v.4.0.2) unless otherwise stated.

**Basic processing and alignment.** Raw reads were processed and aligned to the GRCm38/mm10 reference genome assembly (GENCODE vM23/Ensembl 98) using cellranger count (v.6.1.1).

**Quality control and integration.** Starting with the 'filtered' data matrix from cellranger, additional quality control and processing was performed. First, low-quality cells were filtered based on the number of detected genes, unique molecular identifiers (UMIs) and the percentage of mitochondrial reads using Seurat (v.4.0.0) (ref. [64]). For each sample, the 90th percentile of cells was calculated based on the number of detected UMIs and the number of detected genes. Cells with less than 20% of the 90th percentile (and less than 500 genes and 1,000 UMIs as minimum cutoffs) were removed. Cells with more than 10% of mitochondrial reads were also removed. Second, cell cycle phases were inferred using the function CellCycleScoring from Seurat. Third, gRNAs were assigned to cells using the gRNA matrix provided by cellranger. In the case of multiple detected gRNAs per cell, guides matching 75% or more of the reads per cell were used. If no guide matched 75% or more of reads in a cell, this cell was left unassigned. Fourth, data were aligned across samples and cell cycle effects were removed using the function align_cds from Monocle 3 (v.0.2.3.0) (ref. [65]). Finally, UMAP projection and clustering was performed using the functions reduce_dimension and cluster_cells from Monocle 3.

**Cell type assignment (in vivo and ex vivo).** Cell types were predicted using the package singleR (v.1.4.1), based on a dataset from Izzo and colleagues[56] and a dataset from the packages CytoTRACE (v.0.3.3) (ref. [66]). SingleR[67] was run using the Wilcoxon method for differential analysis.

Cells in clusters with more than 80% of cells predicted as granulocytes, granulocyte progenitors or immature B cells in the bone marrow dataset from CytoTRACE were assigned based on this dataset. All other clusters were assigned based on the predictions by Ninkovic et al.[45]. Eosinophils and basophils were combined in one label. Cells predicted as erythrocytes were further split into MEPs, erythroid progenitors or erythrocytes based on a comparison of gene signatures with external datasets[68] and on the expression of key marker genes: (1) low *Gata2* and high *Gata1, Epor* and *Klf1* marked the transition from MEPs to erythroid progenitors; (2) induction of *Hba-a1* and *Hbb-b1*, increased *Tfrc* expression, enrichment of S and G1 cell cycle signatures mark the transition from erythroid progenitors to erythrocytes. Finally, cells coexpressing high levels of marker genes from the two distinct mature lineages (erythroid and myeloid) were removed as probable doublets or cells with contamination of ambient RNA.

Next, cells constituting less than 10% of a cluster were reassigned to the majority in each cluster. MEPs with *Gata2* expression greater than *Gata1* expression were labeled as early MEPs. MEP clusters with strong cell cycle phase signatures were labeled accordingly. A cluster of MEPs harboring predominantly *Rcor1* knockouts was labeled as 'erythroid perturbed'.

**Cell type enrichment analyses.** To test differences in the distributions of knockouts and NTCs, we tested the enrichment of knockouts compared to NTCs within each cell type. Clusters with fewer than five NTCs or less than 25% NTCs were removed from this analysis. A Fisher's exact test was used with the function fisher.test. Enrichment was tested against each NTC separately. *P* values were adjusted using the function p.adjust with the Benjamini–Hochberg method.

**Viability analysis.** Counts of cells harboring knockouts at day 14 were transformed to CPMs and normalized to the number of cells harboring NTCs. The normalized cell counts were compared to the equally normalized read counts in the gRNA pool before cell infection, resulting in a log fold change that represents viability. Thus, negative or positive values represent an enrichment or loss, respectively of knockout-harboring cells at day 14 (after cell infection) relative to NTCs.

**Cross-projection of ex vivo and leukemia samples to in vivo data.** To project ex vivo and leukemia samples onto the in vivo data, we adapted the ProjecTILs algorithm (v.2.0.2) (ref. [69]) predicting UMAP coordinates for each cell from the ex vivo and leukemia samples based on the UMAP coordinates of in vivo cells using a *k*-nearest neighbor approach with *k* = 20 neighbors.

**Differential expression analysis.** We performed differential expression comparing cells with chromatin factor knockouts to NTCs using nebula (v.1.1.8). We removed clusters with fewer than 31 cells and genes with fewer than 21 reads. We ran nebula[70] with default parameters, testing differences of knockouts to NTCs with fixed effects (parameter 'pred') and adding sample information as random effects (parameter 'id'). For genes, where the algorithm did not converge, we reran nebula with the 'negative binomial lognormal mixed model' model. *P* values were corrected for multiple testing using the function p.adjust with the Benjamini–Hochberg method. Gene set enrichment was performed using the function fgsea from the fgsea package[71].

**Pseudotime trajectory analysis.** Pseudotime for the different lineage trajectories was identified using diffusion maps[72] applied to the NTC population, using the SCANPY (v.1.9.1) (ref. [73]) functions tl.diffmap and tl.dpt. Perturbed cells were then mapped to their nearest *k* = 15 nontargeting cells in the principal component analysis (PCA) space, considering the first *n* = 8 principal components, and then assigned the mean pseudotime value across these cells. PCA cutoffs were found via elbow plots by assessing the variance accounted for in the first *n* principal

components. Each branch was extracted for separate analysis using the aforementioned cell labels (Fig. 2a) and pseudotime was scaled to the unit interval. We plotted perturbations with crucial biological significance, which incidentally showed visually striking distribution differences compared to the nontargeting cell population.

The CITE-seq read counts obtained from cellranger were normalized to log CPMs and then scaled. Perturb-seq data in leukemia were processed in the same way as the in vivo data, as described above. Enrichment analyses were performed on clusters instead of cell types.

## Chromatin accessibility analysis of chromatin factor knockouts

**Isolation of progenitors, CRISPR LOF and ex vivo differentiation.** A total of 20,000 Cas9-LSK cells were transduced with the lenti-CRISPR-BFP virus, expressing the top performing sgRNA against each chromatin factor and cultured for 48 h under multipotent conditions (detailed above). Then, cells were stimulated with cytokine cocktails for lineage priming or myeloid differentiation for 5 days. For the time-course experiment, cells were perturbed and immediately grown for 3, 5 and 7 days under lineage priming or myeloid conditions. Finally, the CRISPR edited progeny was FACS-sorted (BFP⁺GFP⁺) into 1× PBS + 0.5% BSA and collected by centrifugation for ATAC-seq.

ATAC-seq was performed according to the Fast-ATAC protocol described in Corces et al.[74]. Briefly, 50,000 sorted cells were centrifuged at 500*g* for 7 min and resuspended into 25 µl Tagmentation Mix: 1× TD buffer (FC-121-1030, Illumina), 0.01% digitonin (Sigma-Aldrich), 0.1% Tween-20 (Sigma-Aldrich) and 0.1% NP-40 (Thermo Fisher Scientific). Tagmentation was performed at 37 °C for 30 min with agitation at 1,000 rpm. After the tagmentation reaction, 2 µl proteinase K, NaCl (150 mM final concentration) and SDS (0.3% final concentration) were added and the samples were incubated at 50 °C for 30 min. Then, gDNA was purified with SPRI Beads (Beckman Coulter) added at a 2× ratio; tagmented genomic regions were amplified using PCR with the KAPA Master Mix (Roche) and 5 µM P5 and P7 Nextera Indexing Primers (Supplementary Table 8) using the following program: 5 min at 72 °C, 2 min at 98 °C, 8× (98 °C for 20 s, 60 °C for 30 s, 72 °C for 1 min) and 5 min at 72 °C. The ATAC-seq libraries were sequenced at 50 million reads (paired-end 50 bp) on a NextSeq 1000 system.

**Data processing and analysis.** Based on the ATAC-seq nf-core pipeline, we ran Trim Galore (v.0.6.6) (ref. [75]) with Cutadapt (v.3.4) (ref. [76]) using the default parameters to trim low-quality and adapter sequences. We then aligned these reads to the GRCm38/mm10 reference genome assembly with decoy sequences using Bowtie2 (v.2.3.4.2) (ref. [77]) with the following parameters: -X 1000 --no-discordant --no-mixed --very-sensitive. Then, we removed duplicated regions with Picard (v.2.25.4) (Broad Institute, https://broadinstitute.github.io/picard/), noninteresting chromosomes (for example, chrM, chrUn) and blacklisted regions included in the ENCODE blacklist (v.2.0) (ref. [65]). Finally, we removed the Tn5 adapters with alignmentSieve (v.3.5.1) (ref. [78]) (--ATACshift parameter) and indexed the final BAM files with SAMtools (v.1.3.1) (ref. [68]). These BAM files were then processed to CPM-scaled BigWig files with bamCoverage (v.3.5.1) (ref. [79]).

To identify the ATAC peaks, we pooled replicates and converted the paired BAM files to single-read BED format using the function bamToBed from BEDTools (v.2.27.1) (refs. [80,81]). Then, we used MACS (v.2.2.7.1) (ref. [82]) with the parameters --broad -f BED --keep-dup all --nomodel --shift -75 --extsize 150 to call peaks. To compare peak strength between conditions, we generated a unified peak set for all experiments, ending with, respectively, 376,658 and 207,724 peaks in LSKs and Npm1c or Flt3-ITD murine leukemic cells (DM cells), respectively. We then annotated these consensus peaks with the function annotatePeaks from HOMER (v.4.10) (ref. [83]), counted the reads on them with featureCounts (v.2.0.1) (ref. [84]) and calculated adjusted CPM values with the edgeR (v.3.34.1) trimmed mean of *M*-values method[85].

Additionally, we used DESeq2 (v.1.32.0) (ref. 73) to measure the fold change between conditions, defining the peaks with absolute $\log_2$(fold change) values greater than 0.75 and $P_{adj}$ values lower than 0.01 as differentially enriched. Finally, we filtered out peaks with fewer than two CPMs or ten reads in the compared conditions (knockout versus WT).

**Motif analysis.** To look for differential transcription factor motif enrichment between knockout and WT, we used TOBIAS (v.0.13.2) (ref. 28). Following the program guidelines, we generated a consensus set of peaks with all the peaks called previously in both the control and compared knockout sample. We then renamed and formatted HOMER's list of vertebrate known motifs to make it suitable for TOBIAS. After generating the transcription factor footprint BigWig files (using the function ATACorrect with the parameters --read_shift 0 0 and the function ScoreBigwig with default parameters), we computed the differentially bound motifs with the function BINDetect.

Additionally, we used the output transcription factor binding coordinates of each of the motifs to measure the gain or loss of transcription factor union to chromatin for each chromatin factor knockout.

## ChIP–seq analysis of normal and leukemic populations
**Isolation of in vivo hematopoietic cells from bone marrow.** Bone marrow cells were collected from 12–14-week-old C57BL6 mice as described above and stained for the isolation of the following cells: GMP: lineage⁻ (CD3, CD19, CD11b, Gr1, Ter119, B220), c-Kit⁺, Sca-1⁻, FcγRIII⁺, CD34⁺; MEP: lineage⁻ (CD3, CD19, CD11b, Gr1, Ter119, B220), c-Kit⁺, Sca-1⁻, FcγRIII⁻, CD34⁻; monocytes: CD3⁻, CD19⁻, Ter119⁻, CD11b⁺; B cell: CD3⁻, CD19⁺, Ter119⁻, CD11b⁻; erythroid cells were FACS-sorted from the spleens of 12-week-old C57BL6 mice as CD3⁻, CD19⁻, Ter119⁺, CD11b⁻, Gr1⁻. Dilution was 1:100 for all antibodies, except 1:50 for CD34 cells. Cells were sorted in PBS + 0.1% BSA and cross-linked immediately after sorting. The FACS antibodies are shown in Supplementary Table 9.

**Isolation of leukemic cells.** *Npm1c/Flt3-ITD*/Cas9 DM cells were generated from lineage-depleted bone marrow cells of primary transgenic mice after leukemia onset (female, 12 weeks old) as described previously[57,58]. Cells were maintained in XVIVO-20 medium (Lonza) supplemented with 5% FCS, 1% penicillin-streptomycin-glutamine, mouse SCF 50 ng ml⁻¹ (PeproTech), mouse IL-3 10 ng ml⁻¹ (PeproTech) and mouse IL-6 10 ng ml⁻¹ (R&D Systems), in a 37 °C and 5% CO₂ atmospheric environment. *Npm1c/Flt3-ITD*/Cas9 DM cells were passaged every 2 days and cultured for a short time (passages 3–5) to maintain the original leukemic properties.

**Cross-linking.** Freshly sorted normal cells or early passage (3–5) leukemic cells were cross-linked at room temperature with 3 mM ethylene glycol bis(succinimidyl succinate), disuccinimidyl glutarate and dimethyl adipimidate (Thermo Fisher Scientific) for 20 min followed by 1% formaldehyde (Thermo Fisher Scientific) for another 5 min. Then, glycine was added to 125 mM and incubated for 5 min to quench the cross-linkers. Finally, cells were pelleted at 750g for 7 min, washed twice with cold 0.5% BSA/PBS containing 1× cOmplete Protease Inhibitors (Roche) and flash frozen at −80 °C.

**Chromatin immunoprecipitation.** Cross-linked cells were thawed and resuspended in 1.5 ml ice-cold cell lysis buffer (10 mM HEPES, pH 7.5, 10 mM NaCl, 0.2% NP-40 (Thermo Fisher Scientific)) plus cOmplete Protease Inhibitors for 10 min on ice. Then, nuclei were pelleted at 5,000g for 7 min, resuspended in sonication buffer (0.5% SDS, 5 mM EDTA) and pelleted again at 8,000g, then resuspended in 50–100 μl sonication buffer and sonicated for five cycles (30 s ON, 30 s OFF) in a Bioruptor Nano (Diagenode). Then, chromatin extracts were diluted in four volumes of ChIP dilution buffer (25 mM HEPES, 185 mM NaCl, 1.25% Triton X-100 plus cOmplete Protease Inhibitors) and incubated with the relevant antibodies (Supplementary Table 10) at 4 °C for 10–12 h.

The following day, 25 μl Magna ChIP Protein A + G (Merck Millipore) were added and incubated for 3 h at 4 °C. Bead-bound chromatin was washed twice with radioimmunoprecipitation assay (RIPA) buffer (10 mM Tris-Cl, pH 8, 150 mM NaCl, 0.1% SDS, 1% Triton X-100, 1 mM EDTA), twice with RIPA-500 buffer (10 mM Tris-Cl, pH 8, 500 mM NaCl, 0.1% SDS, 1% Triton X-100, 1 mM EDTA), twice with LiCl buffer (10 mM Tris-Cl, pH 8, 550 mM LiCl, 0.5% sodium deoxycholate, 0.5% NP-40, 1 mM EDTA) and once with TE buffer. ChIPped DNA was reverse-cross-linked by 30 min incubation with 2 μl proteinase K in 50 μl ChIP elution buffer (10 mM Tris-Cl, pH 8, 300 mM NaCl, 0.2 mM EDTA, 0.4% SDS) at 55 °C followed by 1-h incubation at 68 °C. Finally, the ChIPped DNA was purified with a 2.2× SPRI cleanup and quantified using the Qubit dsDNA HS Assay Kit (Thermo Fisher Scientific).

Every ChIP–seq experiment was performed in replicate except for Kmt2d and Kmt2a in early myeloid (GMP) and erythroid (MEP) progenitors. All attempts at replication were successful except for a failed Smarcb1 ChIP–seq experiment in MEPs, which was removed from the analysis and substituted by a third ChIP–seq experiment to reach $n = 2$.

**Preparation of ChIP–seq libraries.** ChIP–seq libraries were prepared from 0.5–10 ng of ChIPped DNA using the Next Ultra II kit (New England Biolabs) following the manufacturer's instructions. ChIP–seq libraries were sequenced to 100 million reads per sample (paired-end 50 bp) in a NextSeq 1000 system and demultiplexed using bcl2Fastq (v.2.20).

**ChIP–seq data processing and analysis.** Based on the ChIP–seq nf-core pipeline[86], we first processed the FASTQ files to BAM files as described for ATAC-seq, skipping the Tn5 adapter removal. (The statistics for each ChIP–seq experiment are detailed in Supplementary Table 11). Next, to identify the peaks for each sample, we pooled replicates and used MACS with the parameters -f BAMPE --keep-dup all. To compare peak strength between cell types, we generated a unified peak set per chromatin factor (Brd9, Kmt2a, Kmt2d and Smarcb1). We then followed the steps explained in the ATAC-seq data processing and analysis section to annotate the peaks and calculate the CPM reads on them. Finally, we measured the fold change between cell types using DESeq2 (ref. 85) to get peaks with an absolute $\log_2$(fold change) greater than 0.75 and a $P_{adj}$ lower than 0.01. Normalized ChIP–seq peak counts can be found in the Supplementary Data 3–6.

**Motif analysis in ChIP–seq peaks.** We first generated a list of cell type-specific peak coordinates for each of the analyzed chromatin factors. To do so, we selected all peaks that were significantly enriched or depleted in pairwise comparisons of GMP, myeloid, MEP, erythroid and B cell on each of the chromatin factors. Then, we clustered and manually curated these coordinates to get a list of cell type-specific peaks per chromatin factor. Enrichment of transcription factors in cell type-specific peak coordinates was then analyzed using the function findMotifsGenome from HOMER (v.4.10). For each chromatin factor, cell type-specific peaks were compared to all peaks found across all five cell types and all four chromatin factors as background. Motif enrichment analyses were centered on the 100 bp surrounding the peak summit.

**Comparison of normal and leukemic patterns.** To identify transcription factor switches in leukemia, we defined subsets of peaks that were gained in DM AML cells (leukemic), shared in DM and GMP cells (common) and not present in DM cells but present in GMPs and monocytes (normal). Gained peaks were defined by $\log_2$(fold change) values greater than 1, lost peaks by $\log_2$(fold change) values smaller than −1, and shared peaks by absolute $\log_2$(fold change) values lower than 0.5. The subsetting was done per chromatin factor in a consensus peak dataset of Smarcb1, Kmt2a and Kmt2d. Similar to motif analysis in ChIP–seq peaks section, we looked for enrichment of transcription factors in each of the subsets using the function findMotifsGenome

from HOMER; all peaks from the consensus (across all subsets) were used as the background set.

To measure the Stat5a binding signal over the chromatin factor-bound sites, we collected Stat5a ChIP–seq sequencing data at the same consensus coordinates stated above and computed the CPM values as described in the ATAC-seq data processing and analysis section.

### Functional assays in leukemia
**Cell transduction and sorting of differentiated populations.** *Npm1*/*Flt3-ITD*/Cas9 DM murine cells were transduced using a retronectin-transduction protocol (Takara Bio) with lenti-Perturb-seq-BFP vectors targeting *Smarcb1*, *Smarcd2*, *Brd9* and *Smarcd1*, *Kmt2a*, *Kmt2d*, *Wdr82*, *Hdac3*, *Setdb1*, *Stat5a* or NTC. Then, growth was monitored by flow cytometry (LSRFortessa II; BD Biosciences) of BFP+ cells in culture at 2, 6, 9 and 13 days after transduction. Fold change in BFP+ cells at each time point relative to the proportion of BFP+ cells at day 2 was calculated. Immunophenotypic analysis was performed by flow cytometry at 6 days by staining with anti-CD11b, anti-Ly6G/Ly6C (Gr1) and anti-CD55 (Supplementary Table 9). The proportions of granulocyte-like (CD11b^high Gr-1+) and erythroid/basophil-like (CD55^high) were quantified with FlowJo (v.10.8.1). *P* values were calculated using a two-way ANOVA or ratio-paired *t*-test (Prism v.9.1, GraphPad Software).

**Clonogenic and cell proliferation assays of DM AML cells.** A total of $2 \times 10^6$ DM murine leukemic cells were stained for CD11b, Gr-1, CD55, CD41 and CD34 (Supplementary Table 9). Granulocyte-like (CD11b^high Gr-1+), erythroid/basophil-like (CD55^high CD41−) and CD34+ fractions were subsequently FACS-sorted (BD Influx; BD Biosciences). One thousand cells of each sorted population were seeded in 1 ml methylcellulose medium (M3434, STEMCELL Technologies) supplemented with recombinant mouse SCF (PeproTech) and mouse IL-3 (PeproTech) with recombinant IL-6 and Epo (R&D Systems) in duplicate. Methylcellulose cultures were maintained at 37 °C and 5% $CO_2$; total colony forming units (CFUs) were enumerated 7 days later. Photographs of colonies were obtained with the STEMvision instrument and software (STEMCELL Technologies). Data are presented as a fold change of average CFUs per 1,000 cells seeded, relative to the CD34+ fraction. Ten thousand granulocyte-like, erythroid/basophil-like or CD34+ DM cells were also maintained in standard culture conditions for 21 days, and the number of cells was counted every 7 days. The total cell number is presented as a fold to the corresponding CD34+ counterpart for each time point. *P* values were calculated using a two-way ANOVA (Prism v.9.1, GraphPad Software).

### Statistics and reproducibility
No statistical method was used to predetermine sample size. The lineage scores (bulk CRISPR screens) of the top 200 hits were validated in replicate screens. Likewise, Perturb-seq was performed in replicate for the top 40 chromatin factors. Epigenetic profiling (ATAC-seq and ChIP–seq) was performed in two biologically independent replicates (except for Kmt2d and Kmt2a ChIP–seq in MEPs) following a common practice in the field. Growth curves of chromatin factor knockouts were calculated from three to four independent experimental batches. Immunophenotypic patterns derived from chromatin factor perturbation were replicated at least in two biologically independent experiments.

**Data exclusion.** No data points were excluded from any of the analyses except for one ChIP–seq experiment (Smarcb1 in MEPs), which was excluded from the analysis due to a low signal-to-noise ratio. This data point was substituted by another ChIP–seq experiment to reach $n = 2$.

### Randomization
Cell-based assays (screens and validations) and mouse allocation were randomized; proper batch designs were ensured to avoid confounding effects. In the Perturb-seq analysis, we used several NTCs across all experimental batches. The investigators were not blinded to allocation during experiments and outcome assessment.

### Ethical compliance
Murine ethical compliance was fulfilled under the Guidelines of the Care and Use of Laboratory Animals and were approved by the Institutional Animal Care and Use Committees at the University of Navarra, Spain, and the Animal Welfare Ethical Review Body at the University of Cambridge, UK. Research in the UK was conducted under Home Office license PP3042348.

### Reporting summary
Further information on research design is available in the Nature Portfolio Reporting Summary linked to this article.

### Data availability
Bulk expression patterns of hematopoietic populations: Gene Expression Omnibus (GEO) accession no. GSE60103. Single-cell expression patterns of hematopoiesis: GEO accession no. GSE124822. Perturb-seq datasets (in vivo, ex vivo and leukemic): GEO accession no. GSE213511. Chromatin accessibility of CF-knockouts: GEO accession no. GSE213506. ChIP–seq datasets of chromatin factors (in vivo, ex vivo and leukemic): GEO accession no. GSE213507. Source data are provided with this paper.

### Code availability
Analysis code is available in a dedicated GitHub repository at https://github.com/csbg/tfcf (ref. 87).

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

## Acknowledgements

We thank M. Wiederstein, University of Salzburg, for HPC support, K. Kania from the CRUK Genomics Core for assistance with single-cell methods and N. Cano and M. Dawes for superb administrative support. We thank D. Kent and A. Sebe-Pedrós for their constructive advice and discussions regarding the manuscript. The CRISPR sequencing-BFP backbone (plasmid no. 85707, Addgene) was produced by I. Amit, Weizmann Institute of Science. pMD2.G (plasmid no. 12259, Addgene) and psPAX2 (plasmid no. 12260, Addgene) was produced by D. Trono, École Polytechnique Fédérale de Lausanne. This project was funded by the 'La Caixa' Foundation (agreement no. LCF/PR/HR20/52400016, B.J.P.H., F.P.), Marie Skłodowska-Curie International Fellow (no. 886474, D.L.-A.), a European Hematology Association Junior Research Grant (D.L.-A.), Cancer Research UK Programme Grant (no. DRCRPG-Nov22/100014, B.J.P.H.), the European Research Council (no. 647685, B.J.P.H.), the Cancer Research UK Cambridge Major Centre (no. C49940/A25117, B.J.P.H.), a Kay Kendall Leukaemia Fund Junior Fellowship (no. KKL1440, N.N.), the National Institute for Health and Care Research (NIHR) Cambridge Biomedical Research Centre (no. BRC-1215-20014, B.J.P.H.), the UK Research and Innovation Medical Research Council (MRC) (no. MC_PC_17230, B.J.P.H.) supporting the Wellcome–MRC Cambridge Stem Cell Institute, Wellcome Trust (no. 203151/Z/16/Z, B.J.P.H.). The funders had no role in study design, data collection and analysis, decision to publish or preparation of the manuscript. The views expressed are those of the authors and not necessarily those of the NIHR or the Department of Health and Social Care. For the purpose of open access, the author has applied a CC BY public copyright license to any author-accepted manuscript version arising from this submission.

## Author contributions

D.L.-A. and B.J.P.H. conceptualized the study. D.L.-A., A.G.-S., N.N., C.D.V., G.G., M.N-A, T.B., J.Z., L.P.A.-A., F.M., N.T., I.A.C., C.K.L., D.A., A.L. and B.S. devised the methodology and data collection. D.L.-A., J.M.-E., T.G., J.P.T.-K., N.F. and B.J.P.H. carried out the data analysis. D.L.-A., J.M.-E., N.F. and B.J.P.H. visualized the data. D.L.-A., F.P. and B.J.P.H. acquired the funding. B.J.P.H. managed the project. D.L.-A., N.F. and B.J.P.H. supervised the project. D.L.-A. and B.J.P.H. wrote the original draft. D.L.-A., J.M.-E., N.F. and B.J.P.H. reviewed and edited the manuscript draft.

## Competing interests

T.G. and J.P.T.-K. are employees receiving compensation from Relation Therapeutics. J.P.T.-K. is a founder of Relation Therapeutics. D.L.-A. is a consultant of Relation Therapeutics. The other authors declare no competing interests.

## Additional information

**Extended data** is available for this paper at https://doi.org/10.1038/s41588-023-01471-2.

**Correspondence and requests for materials** should be addressed to David Lara-Astiaso, Nikolaus Fortelny or Brian J. P. Huntly.

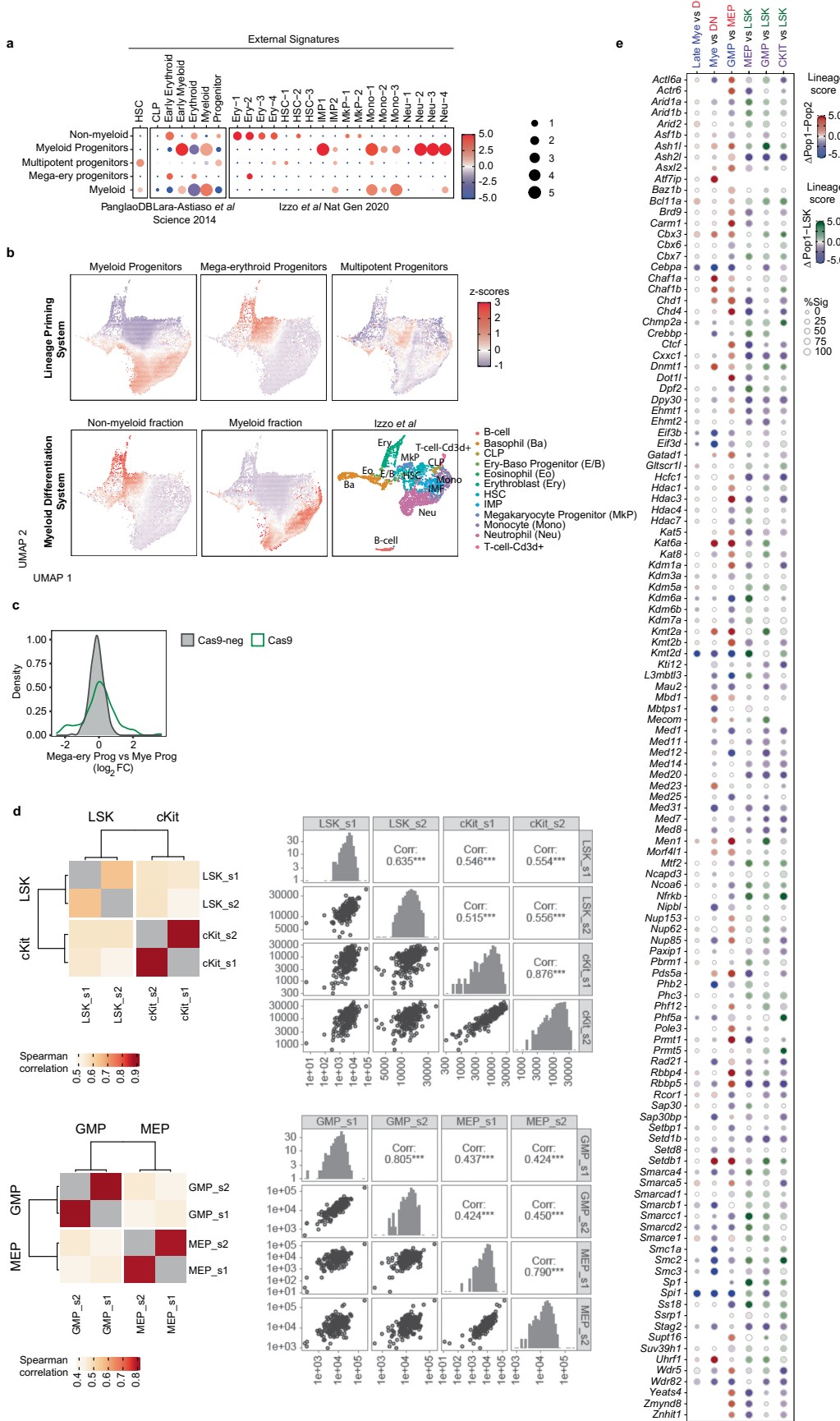

**Extended Data Fig. 1 | See next page for caption.**

**Extended Data Fig. 1 | Characterization of CRISPR Screen systems.**
(**a**) Comparison of expression profiles of the different readout populations from our screens to lineage specific signatures from 3 different studies, named along the bottom of the graph. Comparisons are based on enrichment analyses between the screen signatures and the reference signatures. Dot colour and size relate to log2 odds ratio and -log10 adjusted p-value, respectively. CLP – Common Lymphoid Progenitor, MkP – Megakaryocyte Progenitor, IMP – Immature Myeloid Progenitor, Ery1-4 – Erythroblasts, Neu1-4 – Neutrophils, Mono1-3 – Monocytes. P-values were calculated using the Fisher's exact test. (**b**) Comparison between the expression profiles of FACS-sorted populations from our *ex vivo* systems and a single-cell map of normal haematopoiesis[56].

Bulk transcriptomic signatures derived from FACS-sorted populations were projected on the single-cell map from Izzo and colleagues. (**c**) Example distribution of the CF lineage scores calculated from the Cas9 (green border) and Non-Cas9 (grey) populations. (**d**) Replicate analysis for 200 CFs screened in a second experiment under Self-renewal (top) and Lineage Priming conditions (bottom). (left) heatmaps showing correlation (Spearman) between two replicates, (right) scatter plots showing correlation (Spearman) between replicates. P-values are based on the algorithm AS 89 using the function cor. test in R. (**e**) Lineage scores for all hits. The color of each dot represents the aggregated lineage score. The size represents the number of significant guides, as per key to the right. All values are shown in Supplementary Table 3.

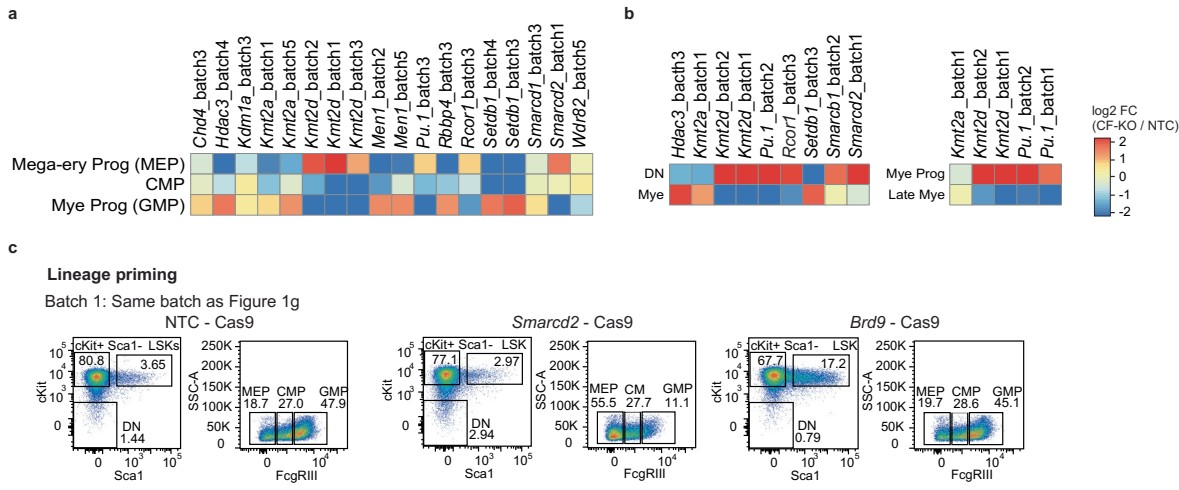

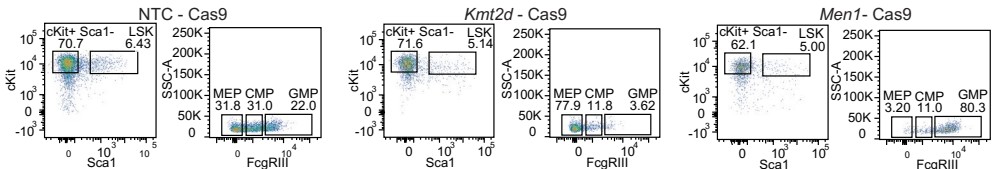

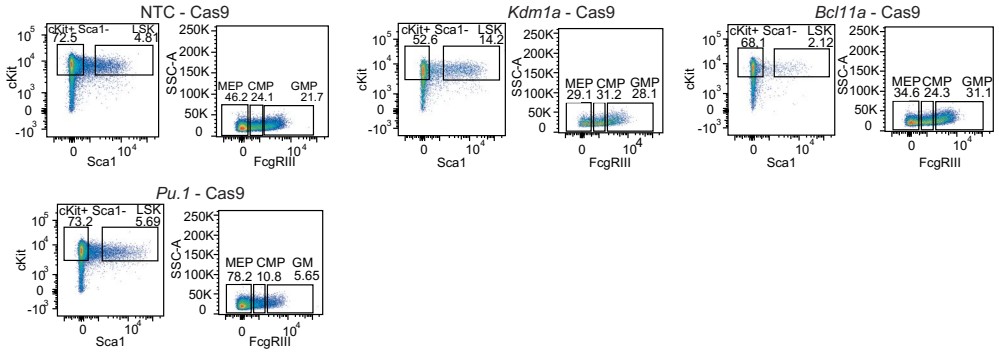

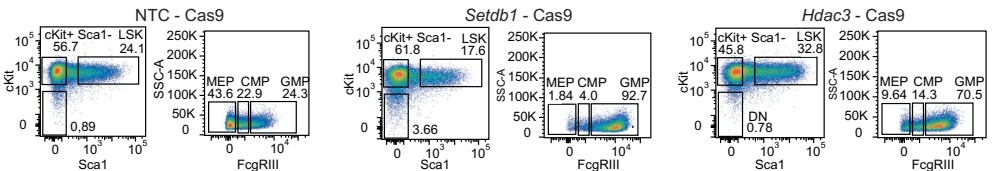

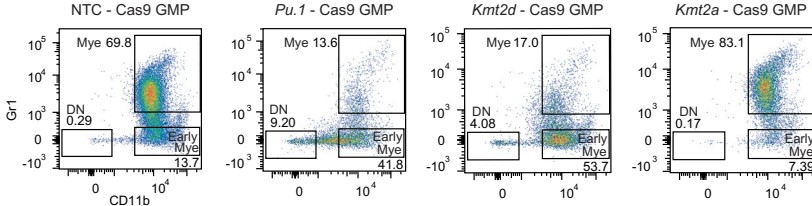

**Extended Data Fig. 2 | See next page for caption.**

**Extended Data Fig. 2 | Validation of the effects of individual CF-KOs.**
**(a-b)** Heatmap showing changes in representative populations for each CF-KO compared to a Non-Targeting Control (Fold-change in population abundances versus NTC) under lineage-priming **(a)** and Myeloid differentiation and terminal Myeloid maturation **(b)**. The Myeloid master regulator *Pu.1* (*Spi1*) was included as a positive control. Gates and values for the selected populations are derived from Supplementary Fig. 2c, d. **(c-d)** Exemplar FACS plots showing validation results for individual CF-KOs under lineage priming conditions **(c)** and Terminal Myeloid Differentiation **(d)**. These validations were performed in different batches. Each batch included a Non-Targeting Control condition. All results were compared with the NTC included in each batch.

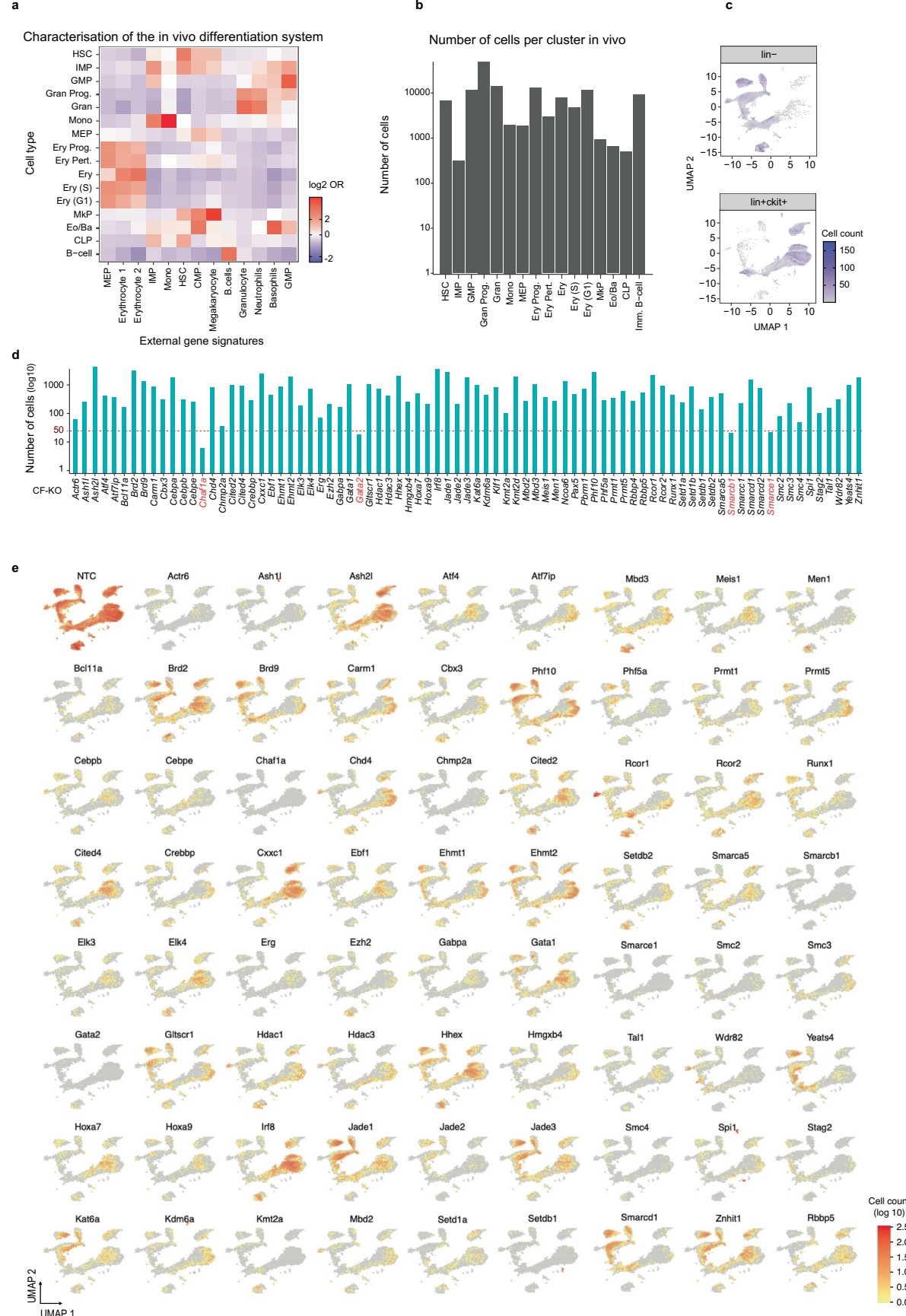

**Extended Data Fig. 3 | See next page for caption.**

**Extended Data Fig. 3 | Characterization of the *in vivo* Perturb-seq system.**
**(a)** Comparison of expression profiles from our *in vivo* single-cell clusters and external cell-type signatures from Izzo et al, 2020[56]. **(b)** Number of cells per cell type. **(c)** UMAP projection of the Lineage- and Lineage+ ckit+ fractions. Color scale represents the number of cells in each area. **(d)** Number of cells with a sgRNAs targeting specific CFs. CF-KOs for which less than 50 cells (in red) were detected were removed from subsequent analysis. **(e)** Visualization of TF- and CF-KO patterns derived from *in vivo* Perturb-seq of Chromatin Regulatory Complexes during lineage specification. The distribution of NTCs is shown as background in grey in all plots. Cells are aggregated and the color of each area represents the density of cells in each area.

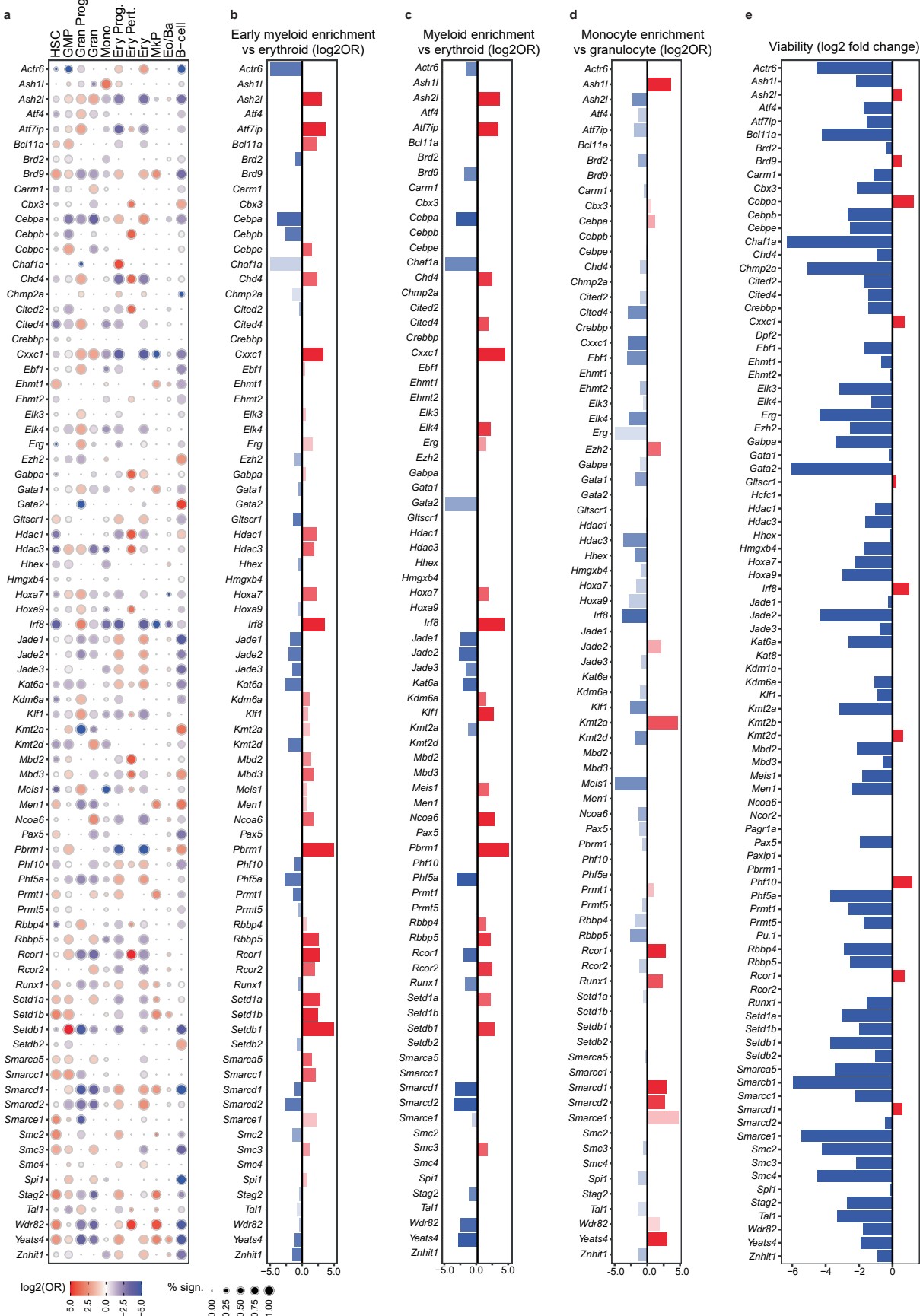

**Extended Data Fig. 4 | See next page for caption.**

**Extended Data Fig. 4 | Extended *in vivo* Perturb-seq analysis of Chromatin Regulatory Complexes during lineage specification. (a)** Enrichment analyses of CF-KOs across 11 cellular states spanning the main hematopoietic lineages, all values are shown in Supplementary Table 4. Dot color and size relate to the log2 odds ratio and the percent of significant enrichments versus NTCs, respectively. The analysis is based on measurements of two aggregated sgRNAs per CF target. **(b)** CF-KO effects on early Myeloid versus Erythroid lineage branching, positive values (red) indicate CF-KOs leading to increased Myeloid outputs. **(c)** CF-KO effects on Myeloid versus Erythroid total outputs, positive values (red) indicate CF-KOs leading to increased Myeloid outputs. **(d)** CF-KO effects on Granulocyte versus Monocyte total outputs, positive values (red) indicate CF-KOs leading to increased monocytic outputs. **(e)** CF-KO effects on viability/survival of CF-KOs after 14 days post-transplant. Negative values (blue) indicate that cells with specific CF-KOs have growth/engraftment disadvantages a when compared to the Control (NTC harboring) cells. Positive values (red) indicate that cells with CF-KOs have growth advantages when compared to the Control (NTC harboring) cells.

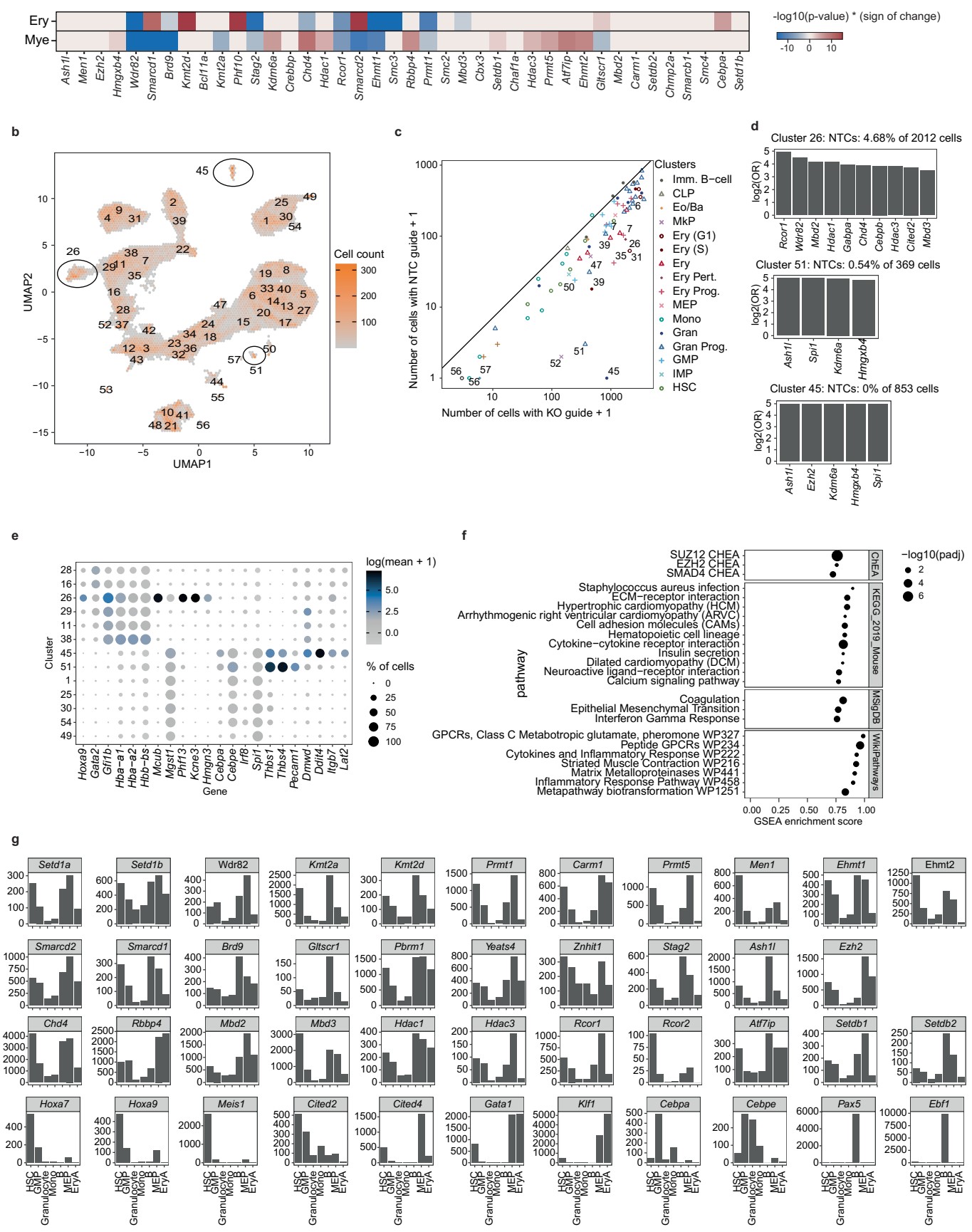

**Extended Data Fig. 5 | See next page for caption.**

**Extended Data Fig. 5 | Extended *in vivo* Perturb-seq analysis of Chromatin Regulatory Complexes during lineage specification. (a)** Heatmap summarizing trajectory analysis for CF-KOs along Myeloid and Erythroid branches ordered from HSCs to mature lineage using pseudotime. Colors are given by signed negative log10 p-values (for p<0.01) generated by a *t*-test between targeting and non-targeting control populations such that negative values correspond to reduced differentiation capability and positive values correspond to increased differentiation capability. **(b-e)** Analysis of aberrant cellular states generated after specific CF-KOs. **(b)** UMAP showing localization of 53 subclusters across the hematopoietic landscape. **(c)** Plot showing the abundance of CF-sgRNAs with respect to Control-sgRNAs across the 53 hematopoietic subclusters. Clusters deviating from the diagonal are rare or absent in the unperturbed scenario. **(d)** Enrichment of specific CF-KO cells in three representative aberrant subclusters: Erythroid-perturbed (cluster 26) and Granulocytic-Perturbed (clusters 45 and 51). **(e)** Marker genes of aberrant clusters. **(f)** Functions specific of the Erythroid-perturbed cluster (26). P-values were calculated by random sampling as implemented in the fgsea R package. **(g)** Barplots showing the expression levels of selected Chromatin and Transcription factors. The bars represent the normalized read counts taken from an RNA-seq dataset[68].

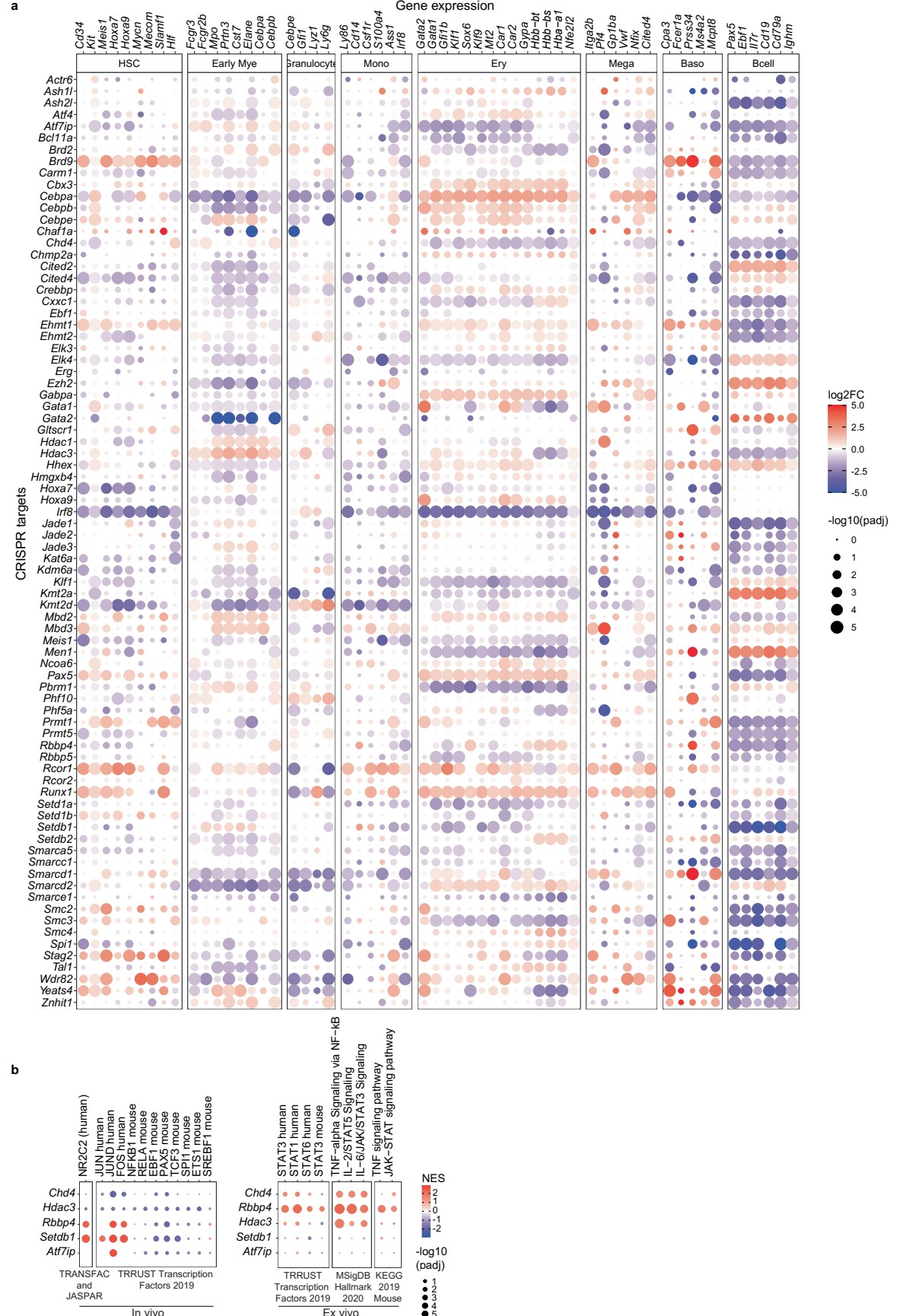

**Extended Data Fig. 6 | See next page for caption.**

**Extended Data Fig. 6 | Extended analysis of transcriptomic effects of CF-KOs.**
**(a)** Analysis of the effect of Chromatin factor disruption (CF-KOs) on lineage specific expression patterns, comprising markers and transcription factors specific for progenitor, Myeloid, Erythroid, Megakaryocytic, Basophil and B-cell lineages. The color of each dot represents the log2 fold change (compared to NTCs), the size represents the −log10 adjusted p-value, as per key to the bottom right. P-values were calculated using negative binomial mixed models from the nebula R package. All values are shown in Supplementary Table 5. **(b)** Gene set enrichment analysis (GSEA) of differentially expressed genes in knockouts of factors belonging to repressive complexes. The color of each dot represents normalized enrichment score, the size represents the −log10 adjusted p-value. P-values were calculated by random sampling as implemented in the fgsea R package.

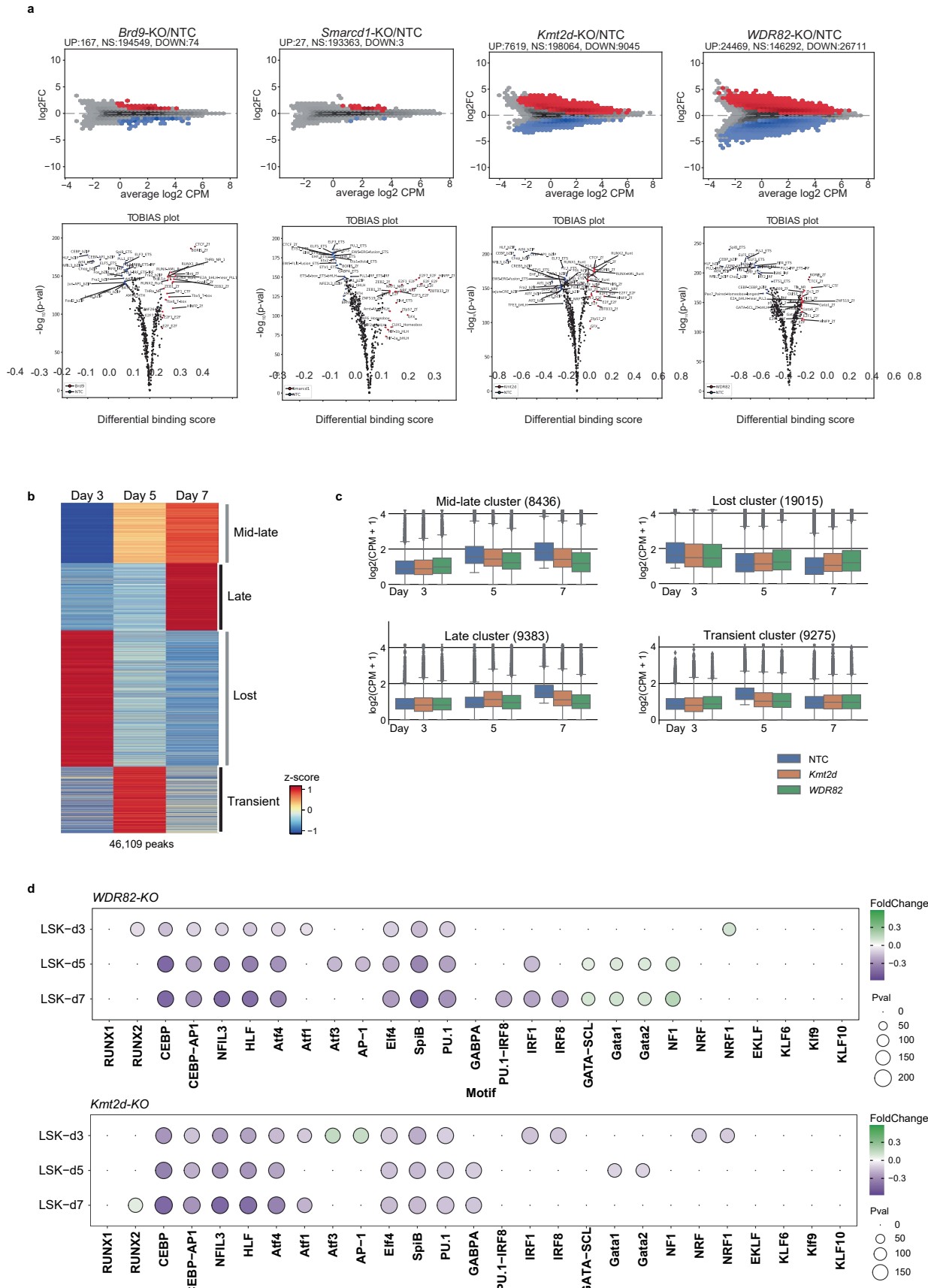

**Extended Data Fig. 7 | See next page for caption.**

**Extended Data Fig. 7 | Extended analysis of effects of CF-KOs on chromatin accessibility and TF footprints. (a)** MA plots demonstrating differential accessibility analysis between selected CF-KOs and Control (NTC). Up- and down-regulated genomic loci are indicated in red and blue, respectively. (lower panel) Volcano Plots showing the differentially bound TF motifs (estimated by TOBIAS) between the same CF-KOs and Control (NTC). Gained and lost footprints are indicated in red and blue, respectively, n = 2 independent experiments. **(b)** Time-series analysis of chromatin accessibility dynamics under ex vivo priming conditions at day 3, 5 and 7, n=2 independent experiments.

**(c)** Effect of *Kmt2d-* and *Wdr82-*KOs on the differential accessible patterns derived from the time-series analysis. n=8436, 19015, 9383, and 9275 for all conditions and days in the mid-late, lost, late, and transient clusters, respectively (n=2). Boxplots display the median and the distribution's 25th (minima) and 75th (maxima) percentiles. The whiskers extend up to 1.5 times the interquartile range (Q3-Q1) from the minima and maxima. **(d)** Time-series analysis of differentially bound TF motifs (estimated by TOBIAS) under lineage priming conditions for *Kmt2d-* and *Wdr82-*KOs.

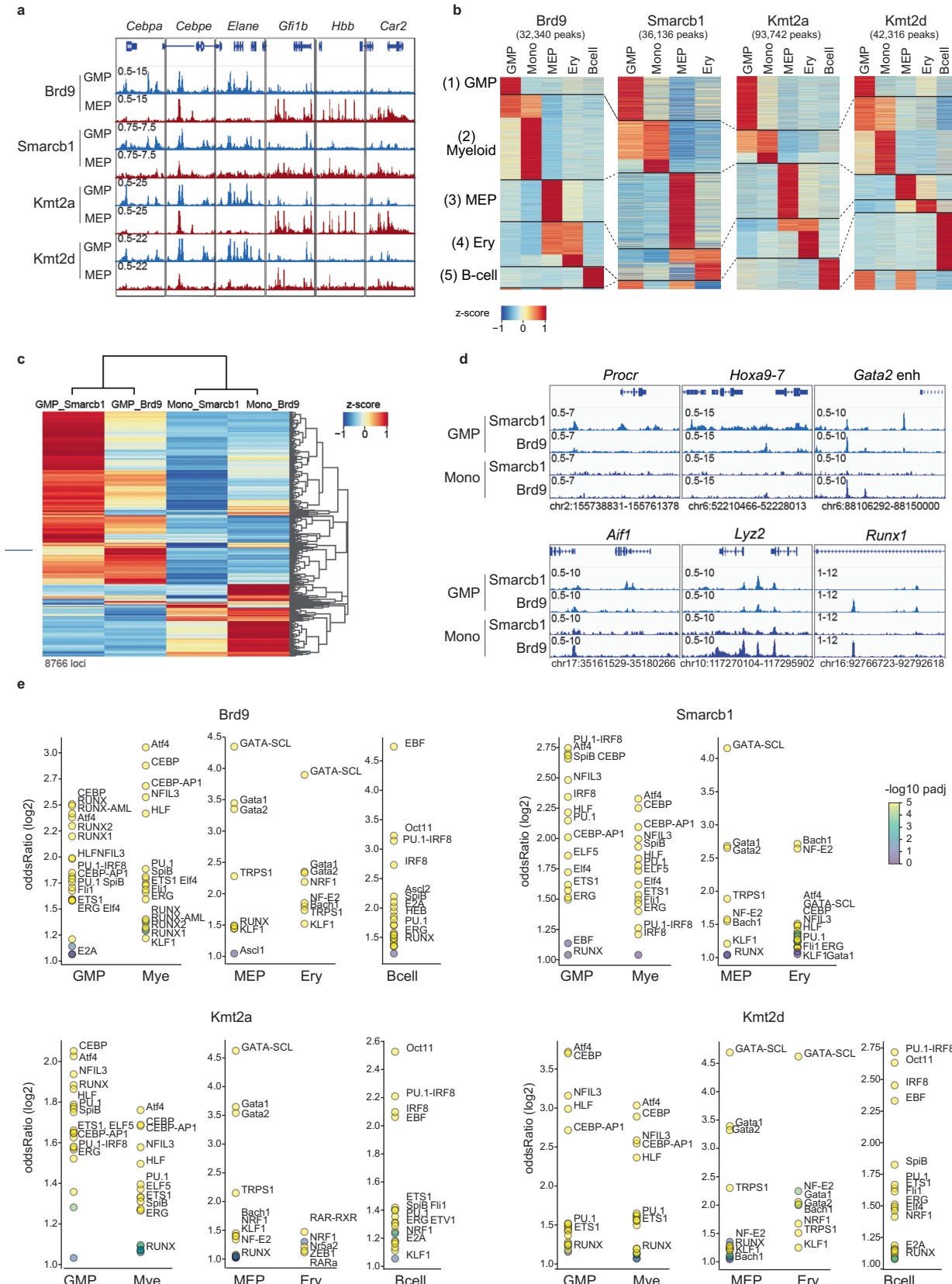

**Extended Data Fig. 8 | See next page for caption.**

**Extended Data Fig. 8 | *In vivo* binding patterns of BAF and COMPASS complexes. (a)** CF binding at representative loci in Myeloid (GMP) and Erythroid (MEP) progenitors. Genomic Coordinates: *Cebpa* chr7:35,114,878-35,131,210; *Cebpe* chr14:54,702,383-54,717,520; *Elane* chr10:79,871,207-79,893,610 ; *Gfi1b* chr2:28,585,038-28,624,000; *Hbb* chr7:103,845,151-103,886,745; *Car2* chr3:14,855,264-14,912,573. (**b**) Heatmaps showing lineage-specific binding patterns for each CF. (**c**) Heatmap showing joint analysis of Smarcb1/cBAF and Brd9/ncBAF binding in Myeloid progenitors (GMPs) and in mature Myeloid cells (monocytes). This analysis shows that Smarcb1/cBAF has widespread binding at early Myeloid stages while Brd9/ncBAF exhibits more presence in mature

Myeloid cells (Monocytes). Strong overlap between cBAF and ncBAF complexes seems limited to few regions. (**d**) Representative binding tracks of Smarcb1 and Brd9 binding in GMP and Monocytes at progenitor loci (upper panel) and mature Myeloid loci (lower panel). (**e**) TF motif co-occurrence in lineage specific binding patterns of Smarcb1 (cBAF), Brd9 (ncBAF), Kmt2a (MLL) and Kmt2d (MLL4). TF motifs (discovered with HOMER) are sorted by their odds ratios (y-axis) in Kmt2a- Kmt2d- Brd9- and Smarcb1- lineage specific peaks: GMP, Mye (GMP & Monocytes), MEP, Ery (MEP & Erythrocytes) and B-cells. The color scale reflects the −log10 p-adjusted values for each TF motif.

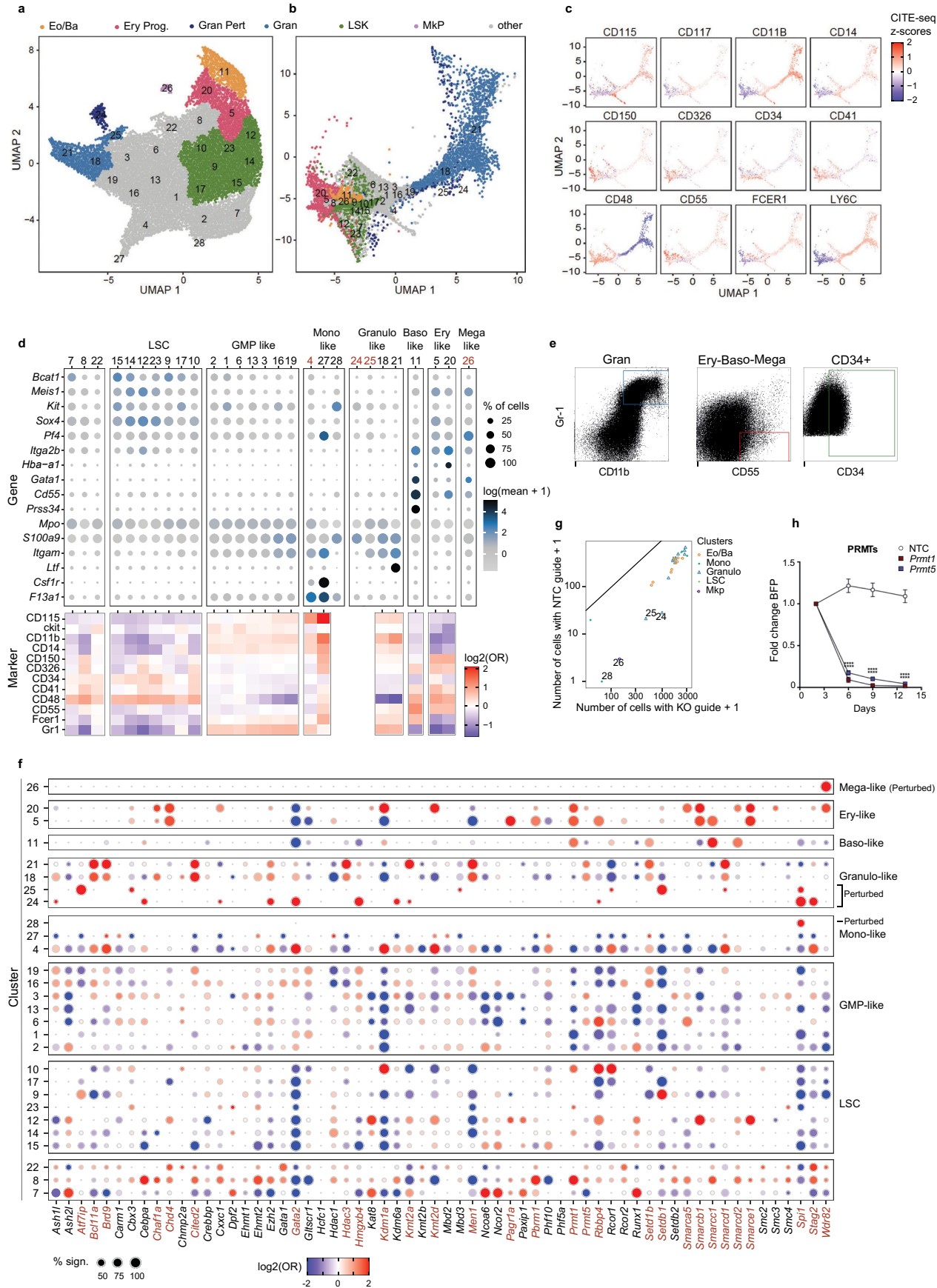

**Extended Data Fig. 9 | See next page for caption.**

**Extended Data Fig. 9 | Extended analysis of Chromatin Factor roles in *Npm1c/Flt3-ITD* Leukemia. (a)** UMAP showing original projection of *Npm1c/Flt3-ITD* single-cell transcriptomes. **(b)** UMAP projection of *Npm1c/Flt3-ITD* single-cell transcriptomes projected over the Hematopoietic in vivo map derived from bone marrow at 14-day post-transplant. **(c)** Scaled CITE-seq signal for 9 surface markers in leukemic cells. **(d)** Expression analysis of markers over the different leukemic clusters. According to their mRNA and Surface marker patterns these are classified into: Leukemic Stem Cells (LSC), GMP-like, Monocyte-like, Granulocyte-like, Basophil-like, Megakaryocyte-like and Erythroid-like. Clusters in red are absent in the unperturbed (NTC) cells. **(e)** Exemplar sorting strategy of leukemic subpopulations showing traits of differentiation into Granulocyte (Gran) or mixed Erythroid-Basophil populations. **(f)** Enrichment analyses of all CF-KOs across leukemia subpopulations. Disruption of factors highlighted in red induce differentiation pathways in leukemia. All values are shown in Supplementary Tables 6 and 7. **(g).** Plot showing the abundance of specific CF-sgRNAs with respect to Control-sgRNAs across the leukemic subclusters. Subclusters deviating from the diagonal are rare or absent in the unperturbed scenario. **(h)** Growth curves of *Prmt1-* and *Prmt5*-KO cells, n=3 biologically independent experiments. The cells expressing each sgRNA harbor a BFP reporter and, the assay measures the change in the proportion of BFP expressing cells over time. ***P<0.001 (Two Way ANOVA). Error bars are SEM.

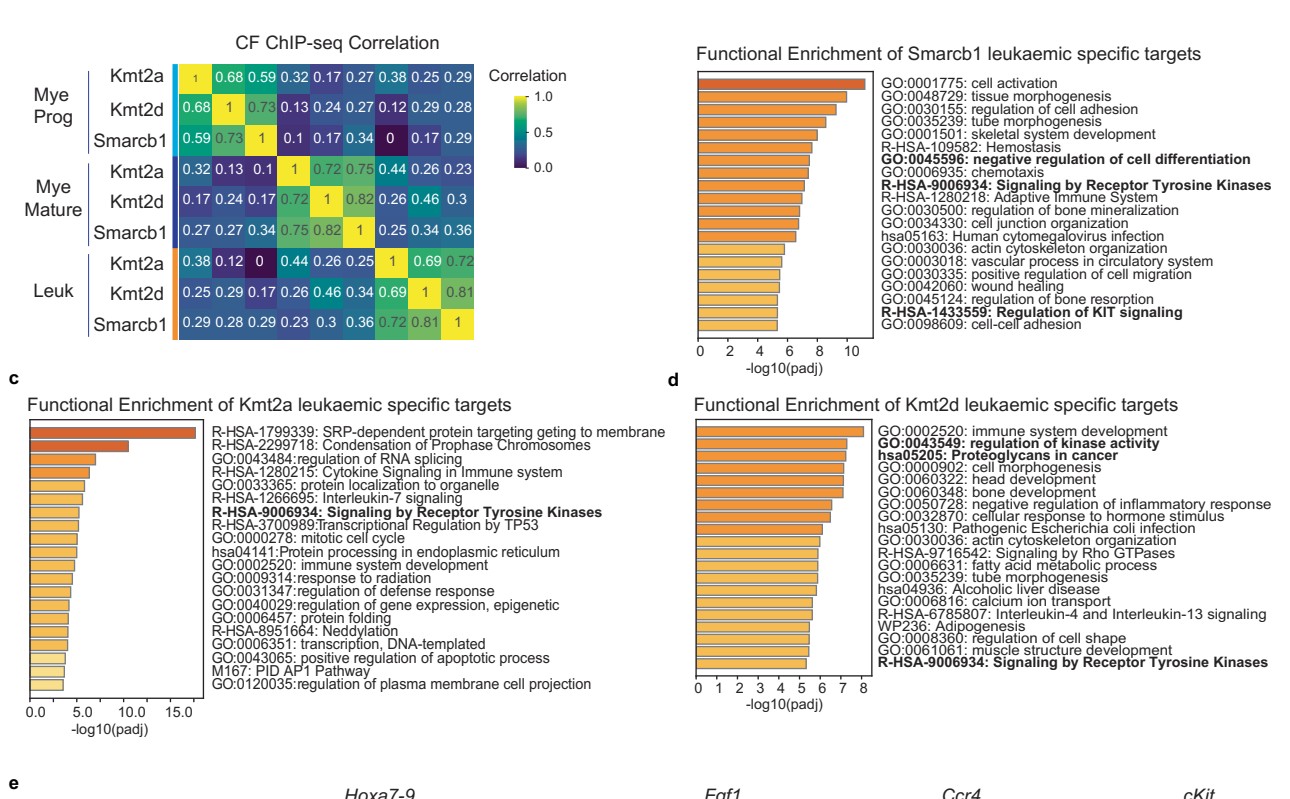

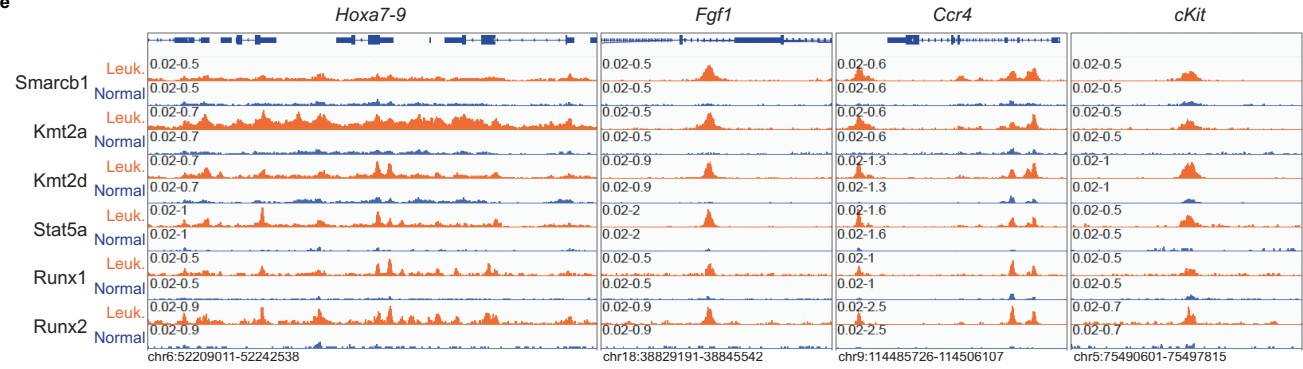

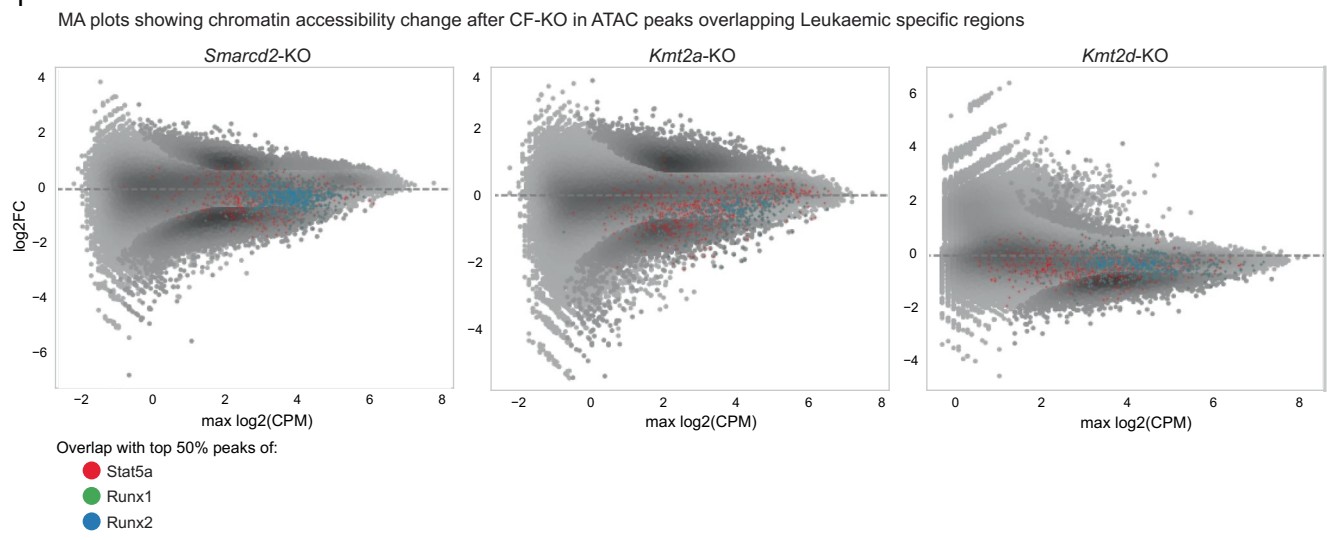

**Extended Data Fig. 10 | See next page for caption.**

**Extended Data Fig. 10 | Extended analysis of cBAF, MLL and MLL4 binding and TF-partnership in *Npm1c/Flt3-ITD* Leukemia. (a)** Correlation (spearman) analysis of CF binding patterns in Myeloid progenitors, Mature myeloid and leukemic populations. **(b-d)** Enrichment analysis of Smarcb1-, Kmt2a- and Kmt2d- bound loci specific of leukemia cells. Bar graphs show enriched terms across input gene lists, sorted by p-values. Targets connected to leukemia specific peaks (nearest TSS) were run in Metascape[87]. Functions with particularly high relevance for *Npm1c/Flt3-ITD* leukemia are highlighted in bold. **(e)** Genome-browser snapshots showing binding of cBAF/Smarcb1, MLL/Kmt2a and MLL4/Kmt2d on leukemic specific loci. **(f)** MA plots showing accessibility changes after acute depletion of *Smarcd2*-KO (cBAF), *Kmt2a*-KO (MLL) and *Kmt2d*-KO (MLL4) with respect to Control cells harbouring NTC sgRNAs. Color coding loci overlapping with Stat5a (red), Runx1(green) and Runx2 (blue).

Brian Huntly
Nikolaus Fortelny

# Reporting Summary

## Statistics

For all statistical analyses, confirm that the following items are present in the figure legend, table legend, main text, or Methods section.

| n/a | Confirmed | |
|---|---|---|
| ☐ | ☒ | The exact sample size (*n*) for each experimental group/condition, given as a discrete number and unit of measurement |
| ☐ | ☒ | A statement on whether measurements were taken from distinct samples or whether the same sample was measured repeatedly |
| ☐ | ☒ | The statistical test(s) used AND whether they are one- or two-sided <br> *Only common tests should be described solely by name; describe more complex techniques in the Methods section.* |
| ☐ | ☒ | A description of all covariates tested |
| ☐ | ☒ | A description of any assumptions or corrections, such as tests of normality and adjustment for multiple comparisons |
| ☐ | ☒ | A full description of the statistical parameters including central tendency (e.g. means) or other basic estimates (e.g. regression coefficient) AND variation (e.g. standard deviation) or associated estimates of uncertainty (e.g. confidence intervals) |
| ☐ | ☒ | For null hypothesis testing, the test statistic (e.g. *F*, *t*, *r*) with confidence intervals, effect sizes, degrees of freedom and *P* value noted <br> *Give P values as exact values whenever suitable.* |
| ☒ | ☐ | For Bayesian analysis, information on the choice of priors and Markov chain Monte Carlo settings |
| ☒ | ☐ | For hierarchical and complex designs, identification of the appropriate level for tests and full reporting of outcomes |
| ☒ | ☐ | Estimates of effect sizes (e.g. Cohen's *d*, Pearson's *r*), indicating how they were calculated |

*Our web collection on statistics for biologists contains articles on many of the points above.*

## Software and code

Policy information about availability of computer code

| Data collection | No software was used for data collection |
|---|---|
| Data analysis | bcl2fastq (version 2.20) <br> bowtie2 (version 2.3.4.2) <br> SAM tools (version 1.3.1) <br> R (version 4.0.2) <br> edgeR (version 3.32.1) <br> edgeR (version 3.34.1) <br> limma (version 3.46.0) <br> FlowJo (version 10.8.0) <br> FlowJo (version 10.8.1) <br> CellRanger (version 6.1.1) <br> Seurat (version 4.0.0) <br> Monocle 3 (version 0.2.3.0) <br> CytoTRACE (version 0.3.3) <br> SingleR (version 1.4.1) <br> ProjectTILs (version 2.0.2) <br> Scanpy (version 1.9.1) <br> Trim Galore (version 0.6.6) <br> Cutadapt (version 3.4) <br> ENCODE blacklist (version 2.0) |

bamCoverage (version 3.5.1)
BEDTools (version 2.27.1)
MACS (version 2.2.7.1)
HOMER (version 4.10)
featureCounts (version 2.0.1)
DESeq2 (version 1.32.0)
TOBIAS (version 0.13.2)
Prism (GraphPad software; version 9.1)
nebula (version 1.1.8)
Custom code can be found here https://github.com/csbg/tfcf

For manuscripts utilizing custom algorithms or software that are central to the research but not yet described in published literature, software must be made available to editors and reviewers. We strongly encourage code deposition in a community repository (e.g. GitHub). See the Nature Portfolio guidelines for submitting code & software for further information.

## Data

Policy information about availability of data

All manuscripts must include a data availability statement. This statement should provide the following information, where applicable:
- Accession codes, unique identifiers, or web links for publicly available datasets
- A description of any restrictions on data availability
- For clinical datasets or third party data, please ensure that the statement adheres to our policy

Data and Materials Availability:
Bulk Expression patterns of hematopoietic populations: Lara-Astiaso et al, Science 2012 (52), GEO accession (GSE60103)
Single-cell expression patterns of hematopoiesis: Izzo et al, Nat Genetics 2020 (45), GEO accession (GSE124822)
Perturb-seq datasets: in vivo, ex vivo and leukemic: GEO accession (GSE213511)
Chromatin accessibility of CF-KOs: GEO accession (GSE213506)
ChIP-seq datasets of CFs in vivo, ex vivo and leukemic: GEO accession (GSE213507)

Databases used in this study:
GRCm38/mm10 reference genome assembly (GENCODE vM23/Ensembl 98)

## Human research participants

Policy information about studies involving human research participants and Sex and Gender in Research.

| Reporting on sex and gender | No human samples were used in the study |
|---|---|
| Population characteristics | No human samples were used in the study |
| Recruitment | No human samples were used in the study |
| Ethics oversight | No human samples were used in the study |

Note that full information on the approval of the study protocol must also be provided in the manuscript.

# Field-specific reporting

Please select the one below that is the best fit for your research. If you are not sure, read the appropriate sections before making your selection.

☒ Life sciences ☐ Behavioural & social sciences ☐ Ecological, evolutionary & environmental sciences

For a reference copy of the document with all sections, see nature.com/documents/nr-reporting-summary-flat.pdf

# Life sciences study design

All studies must disclose on these points even when the disclosure is negative.

| Sample size | No sample size calculation was used. 2 biologically independent experimental replicates were performed for ChIP-seq and ATAC-seq profiling, this is a common standard in the field.<br>For the validation of CRISPR screen hits in normal and leukemia, a minimum of 3 biologically independent replicates were used, we believe this number provides strong robustness, especially when these results validate other orthogonal findings (CRISPR screens and functions derived from binding patterns).<br>In the Perturb-seq screens, we designed our experiments to assay on average 100 cells / CF-KO, which is an established standard in the field. |
|---|---|
| Data exclusions | Perturb-seq Analysis: |

| Data exclusions | To avoid spurious results arising from undersampling, we removed cell clusters with less than 31 cells and genes with less than 21 reads, which represent very small clusters and lowly sampled genes. These cutoffs were based on our exploratory analysis of the data and match the criteria used in seminal studies using perturbation screens (Replogle et al Cell,185,14, July 2022) |
|---|---|
| Replication | CRISPR screens and validation of candidates:<br>- Screens were performed in 2 replicates at a 500X CRISPR library coverage<br>- Validation of the effects of hits derived from the screens was performed in replicates of at least 3 independent experiments.<br>- Validation experiments were reproducible and confirmed the patterns derived from the bulk screens<br><br>In vivo Perturb-seq.<br>- Experiments were performed in different batches with all showing similar values for donor engraftment and transduction efficiency.<br>- We analysed the unperturbed patterns (cells with Non-Targeting Control guides) across batches. Batches where cells with Non-Targeting Control guides showed different trends were discarded.<br>- For the 40 top CF regulators including members of the COMPASS, BAF NurD and Repressors presented in the main figures we performed replicate experiments, which recapitulated the initial Perturb-seq patterns.<br><br>ATAC-seq of CF-KOs: All experiments were conducted in 2 replicates. All attempts of replication were successful<br><br>Leukaemia Perturb-seq.<br>- Experiments were performed in different batches with all showing similar values for transduction efficiency<br>- We analysed the unperturbed patterns (cells with Non-Targeting Control guides) across batches. Batches where cells with Non-Targeting Control guides showed different trends were discarded.<br>- For the 20 top CF regulators including members of the COMPASS, BAF NurD and Repressors presented in the main figures we performed replicate experiments, which recapitulated the initial Perturb-seq patterns.<br><br>ChIP-seq: All experiments were conducted in 2-3 replicates except for Kmt2d, Kmt2a ChIP-seq in early progenitors (GMPs and MEPs)<br>All attempts of replication were successful except for a failed ChIP-seq for Smarcb1 in MEPs that was discarded due to having low signal-to-noise.<br><br>Growth assays in CF-KOs: All experiments were conducted in 3-4 replicates. All attempts of replication were sucessful<br><br>FACS validation of chromatin factor perturbation were performed in at least 2 replicates. |
| Randomization | Allocation of animals for the bulk and Perturb-seq screens was randomized (using always even numbers of males and females in each experimental condition)<br>We have used bulk and single-cell (perturb-seq) to study the roles of chromatin factors in hematopoiesis and leukaemia. Our perturbations were performed in the same cell population (hematopoietic progenitors) producing a pool of different CF mutant cells. The generation of such pool of CF mutants is already a random process, thus we don't need further randomizaton |
| Blinding | CF-KOs and NTC controls were analyzed side by side and the experimental groups (specific CF-KOs) were experimentally determined based on the expression of gRNAs (and not assigned a priori). Thus, as groups are defined by the data, investigators could not be blinded. |

# Reporting for specific materials, systems and methods

We require information from authors about some types of materials, experimental systems and methods used in many studies. Here, indicate whether each material, system or method listed is relevant to your study. If you are not sure if a list item applies to your research, read the appropriate section before selecting a response.

## Materials & experimental systems

| n/a | Involved in the study |
|---|---|
| ☐ | ☒ Antibodies |
| ☐ | ☒ Eukaryotic cell lines |
| ☒ | ☐ Palaeontology and archaeology |
| ☐ | ☒ Animals and other organisms |
| ☒ | ☐ Clinical data |
| ☒ | ☐ Dual use research of concern |

## Methods

| n/a | Involved in the study |
|---|---|
| ☐ | ☒ ChIP-seq |
| ☐ | ☒ Flow cytometry |
| ☒ | ☐ MRI-based neuroimaging |

## Antibodies

| Antibodies used | SMARCB1/BAF47 (D8M1X) Rabbit mAb Cell Signalling 91735 Lot 2<br>Anti-BRD9 antibody Abcam ab137245 Lot GR3372527-6<br>Anti-KMT2D antibody produced in rabbit Sigma HPA035977<br>Anti-Kmt2a MLL1 Antibody Bethyl A300-086A Lot 6<br>IgG Rabbit IgG, polyclonal - Isotype Control (ChIP Grade) 100 ug Abcam ab171870<br>Anti-Stat5a Recombinant Anti-STAT5a antibody [E289] Abcam ab32043 Lot GR3238474-7<br>Anti-Cebpa Abcam ab40764 Lot GR4228581-2<br>Anti-Cebpe Sigma-Aldrich HPA002928<br>Anti-CD45R/B220 BV510 RA3-6B2 BioLegend ref 103247<br>Anti-CD3e BV510 145-2C11 BioLegend ref 100233 |
|---|---|

Anti-CD11b BV510 M1/70 BioLegend ref 101263
Anti-CD11b PECy7 M1/70 BioLegend ref 101215
Anti-Gr1 BV510 RB6-8C5 BioLegend ref 108437
Anti-Ter119 BV510 Ter-119 BioLegend ref 116237
Anti-CD16/32 (FcgR-III) PercPCy5.5 93 BioLegend ref 101323
Anti-CD34 FITC RAM34 Invitrogen ref 11-0341-82
Anti-CD41 APCCy7 MWReg30 BioLegend ref 133927
Anti-CD45.1 PECy7 A20 BioLegend ref 110730
Anti-CD45.2 APC/Fire750 104 BioLegend ref 109852
Anti-CD55 PE RIKO-3 BioLegend ref 131803
CD117 (c-kit) APC 2B8 BioLegend ref 105812
Anti-Sca1 PE D7 BioLegend ref 108107
Anti-CD11b Total-Seq B M1/70 Biolegend ref 101273
Anti-Ly6C Total-Seq B HK1.4,  Biolegend ref 128053
Anti-CD115 Total-Seq B AFS98,  Biolegend ref 135543
Anti-CD14 Total-Seq B Sa14-2,  Biolegend ref 123341
Anti-CD150 Total-Seq B TC15-12F12.2,  Biolegend ref 115951
Anti-CD48 Total-Seq B HM48-1,  Biolegend ref 103457
Anti-CD34 Total-Seq B SA376A4,  Biolegend ref 152213
Anti-CD117 Total-Seq B 2B8 Biolegend ref 105849
Anti-CD55 Total-Seq B RIKO-3 Biolegend ref 131817
Anti-CD41 Total-Seq B MWReg30 Biolegend ref 133941
Anti-CD326 Total-Seq B G8.8 Biolegend ref 118247
Anti-FceRI Total-Seq B Mar-01 Biolegend ref 134341

Validation

SMARCB1/BAF47 (D8M1X) Rabbit mAb  Cell Signalling 91735 Lot 2
Validated by the supplier (Cell Signalling):
Product Usage Information.
For optimal ChIP and ChIP-seq results, use 10 µl of antibody and 10 µg of chromatin (approximately 4 x 106 cells) per IP. This
antibody has been validated using SimpleChIP® Enzymatic Chromatin IP Kits.

Anti-BRD9 antibody Abcam ab137245 Lot GR3372527-6
Validation:
1) We compared the binding pattern obtained with this antibody to the Brd9 antibody provided by Active Motif (https://
www.activemotif.com/catalog/details/61537/brd9-antibody-pab currently discontinued) and found a strong correlation between
both patterns
2) Used for ChIP-seq in the following publication.
Inoue D et al. Spliceosomal disruption of the non-canonical BAF complex in cancer
Nature. 2019 October ; 574(7778): 432–436.

Anti-KMT2D antibody produced in rabbit  Sigma HPA035977
Validation:
- Used for ChIP-seq in:
Zhang J et at. Disruption of KMT2D perturbs germinal center B cell development and promotes lymphomagenesis.
Nature Medicine October 2015

Kmt2a MLL1 Antibody Bethyl A300-086A Lot 6
Validation:
- Used for ChIP-seq in 20 publications (see https://www.citeab.com/antibodies/654488-a300-086a-rabbit-anti-mll1-antibody-affinity-
purifie) amongst them:
Schwörer, S., et al. Epigenetic stress responses induce muscle stem-cell ageing by Hoxa9 developmental signals.
Nature on 15 December 2016

IgG Rabbit IgG, polyclonal - Isotype Control (ChIP Grade) 100 ug Abcam ab171870
- Validated by the provider (abcam)

Stat5a Recombinant Anti-STAT5a antibody [E289]  Abcam ab32043 Lot GR3238474-7
- Used for ChIP-seq in 2 publications
He, L.,  et al. Local blockage of self-sustainable erythropoietin signaling suppresses tumor progression in non-small cell lung cancer.
Oncotarget on 10 October 2017
Lee, K. M., et al. Inhibition of STAT5A promotes osteogenesis by DLX5 regulation.
Cell Death & Disease on 14 November 2018
https://www.citeab.com/antibodies/775960-ab32043-anti-stat5a-antibody-e289?des=3bc20e5fb5095dd9

Cebpa Abcam ab40764
- Used for ChIP-seq in 4 publications (below are the two most recent)
 Qin, Y., Grimm, S. A., et al. Alterations in promoter interaction landscape and transcriptional network underlying metabolic
adaptation to diet
Nature Communications on 19 February 2020
Yao, S., Wu, D., et al. Hypermethylation of the G protein-coupled receptor kinase 6 (GRK6) promoter inhibits binding of C/EBPα, and
GRK6 knockdown promotes cell migration and invasion in lung adenocarcinoma cells.
FEBS Open Bio on 1 April 2019

Anti-CD45R/B220 BV510 RA3-6B2 BioLegend ref 103247
Validation:

- Used in several publications (see https://www.biolegend.com/ja-jp/products/brilliant-violet-510-anti-mouse-human-cd45r-b220-antibody-7996) amongst them:
Hutter K et al. The miR-15a/16-1 and miR-15b/16-2 clusters regulate early B cell development by limiting IL-7 receptor expression. Front Immunol. 2022 Aug 25;13:967914.

Anti-CD3e BV510 145-2C11 BioLegend ref 100233
Validation:
- Used in several publications (see https://www.biolegend.com/ja-jp/products/brilliant-violet-510-anti-mouse-cd3-antibody-7990) amongst them:
Shen E et al. Control of Germinal Center Localization and Lineage Stability of Follicular Regulatory T Cells by the Blimp1 Transcription Factor.
Cell Rep. 2019 Nov 12;29(7):1848-1861.e6.

Anti-CD11b BV510 M1/70 BioLegend ref 101263
Validation:
- Used in several publications (see https://www.biolegend.com/ja-jp/products/brilliant-violet-510-anti-mouse-human-cd11b-antibody-7993) amongst them:
Ramakrishna C et al. Bacteroides fragilis polysaccharide A induces IL-10 secreting B and T cells that prevent viral encephalitis.
Nat Commun. 2019 May 14;10(1):2153.

Anti-CD11b PECy7 M1/70 BioLegend ref 101215
Validation:
- Used in several publications (see https://www.biolegend.com/ja-jp/products/pe-cyanine7-anti-mouse-human-cd11b-antibody-1921) amongst them:
Hatzi K et al. Histone demethylase LSD1 is required for germinal center formation and BCL6-driven lymphomagenesis.
Nat Immunol. 2019 Jan;20(1):86-96.

Anti-Gr1 BV510 RB6-8C5 BioLegend ref 108437
Validation:
- Used in several publications (see https://www.biolegend.com/en-us/products/brilliant-violet-510-anti-mouse-ly-6g-ly-6c-gr-1-antibody-8614) amongst them:
Liu Y et al. Rapid acceleration of KRAS-mutant pancreatic carcinogenesis via remodeling of tumor immune microenvironment by PPARδ.
Nat Commun. 2022 May 13;13(1):2665.

Anti-Ter119 BV510 Ter-119 BioLegend ref 116237
Validation:
- Used in several publications (see https://www.biolegend.com/en-us/products/brilliant-violet-510-anti-mouse-ter-119-erythroid-cells-antibody-8243) amongst them:
Yamaguchi A et al. Blockade of the interaction between BMP9 and endoglin on erythroid progenitors promotes erythropoiesis in mice.
Genes Cells. 2021 Oct;26(10):782-797.

Anti-CD16/32 (FcgR-III) PercPCy5.5 93 BioLegend ref 101323
Validation:
- Used in several publications (see https://www.biolegend.com/ja-jp/products/percp-cyanine5-5-anti-mouse-cd16-32-antibody-6165) amongst them:
Viny AD et al. Cohesin Members Stag1 and Stag2 Display Distinct Roles in Chromatin Accessibility and Topological Control of HSC Self-Renewal and Differentiation.
Cell Stem Cell. 2019 Nov 7;25(5):682-696.e8.

Anti-CD34 FITC RAM34 Invitrogen ref 11-0341-82
Validation:
- Used in several publications (see https://www.thermofisher.com/antibody/product/CD34-Antibody-clone-RAM34-Monoclonal/11-0341-82) amongst them:
Wilkinson AC et al. Long-term ex vivo haematopoietic-stem-cell expansion allows nonconditioned transplantation.
Nature. 2019 Jul;571(7763):117-121.

Anti-CD41 APCCy7 MWReg30 BioLegend ref 133927
Validation:
- Used in several publications (see https://www.biolegend.com/ja-jp/products/apc-cyanine7-anti-mouse-cd41-antibody-13014) amongst them:
Al-Rifai R et al. JAK2V617F mutation drives vascular resident macrophages toward a pathogenic phenotype and promotes dissecting aortic aneurysm.
Nat Commun. 2022 Nov 3;13(1):6592.

Anti-CD45.1 PECy7 A20 BioLegend ref 110730
Validation:
- Used in several publications (see https://www.biolegend.com/ja-jp/products/pe-cyanine7-anti-mouse-cd45-1-antibody-4917) amongst them:
Garo LP et al. MicroRNA-146a limits tumorigenic inflammation in colorectal cancer.
Nat Commun. 2021 Apr 23;12(1):2419.

Anti-CD45.2 APC/Fire750 104 BioLegend ref 109852
Validation:
- Used in several publications (see https://www.biolegend.com/ja-jp/products/apc-fire-750-anti-mouse-cd45-2-antibody-13589)

amongst them:
Formaglio P et al. Nitric oxide controls proliferation of Leishmania major by inhibiting the recruitment of permissive host cells.
Immunity. 2021 Dec 14;54(12):2724-2739.e10.

Anti-CD55 PE RIKO-3 BioLegend ref 131803
Validation:
- Used in several publications (see https://www.biolegend.com/ja-jp/products/pe-anti-mouse-cd55-daf-antibody-5514) amongst them:
Camps J et al. Interstitial Cell Remodeling Promotes Aberrant Adipogenesis in Dystrophic Muscles.
Cell Rep. 2020 May 5;31(5):107597.

CD117 (c-kit) APC 2B8 BioLegend ref 105812
Validation:
- Used in several publications (see https://www.biolegend.com/ja-jp/products/apc-anti-mouse-cd117-c-kit-antibody-72) amongst them:
Lawson H et al. CITED2 coordinates key hematopoietic regulatory pathways to maintain the HSC pool in both steady-state hematopoiesis and transplantation.
Stem Cell Reports. 2021 Nov 9;16(11):2784-2797.

Anti-Sca1 PE D7 BioLegend ref 108107
Validation:
- Used in several publications (see https://www.biolegend.com/ja-jp/products/pe-anti-mouse-ly-6a-e-sca-1-antibody-228) amongst them:
Tran NT et al. Efficient CRISPR/Cas9-Mediated Gene Knockin in Mouse Hematopoietic Stem and Progenitor Cells.
Cell Rep. 2019 Sep 24;28(13):3510-3522.e5.

# Eukaryotic cell lines

Policy information about cell lines and Sex and Gender in Research

| Cell line source(s) | HEK 293T (Sigma, 12022001-DNA-5UG) |
| --- | --- |
| Authentication | Purchased from the provider (Sigma) as an autheticated cell line. We did not performed any further authetication and used early passages (p< 8) were used for lentivirus production. |
| Mycoplasma contamination | Cell lines were tested negative for Mycoplasma |
| Commonly misidentified lines (See ICLAC register) | No commonly misidentified cells were used |

# Animals and other research organisms

Policy information about studies involving animals; ARRIVE guidelines recommended for reporting animal research, and Sex and Gender in Research

| Laboratory animals | C57BL/6J (Jackson Laboratory #JAX_000664)<br>Age: 12-14 weeks<br>Sex: Equal numbers of males and females<br><br>Gt(ROSA)26Sortm1.1(CAG-cas9*/EGFP)Rsky (Jackson Laboratory #JAX_026179)<br>Age: 12-15 weeks<br>Sex: Equal numbers of males and females<br><br>B6.SJL-Ptprca Pepcb/BoyJ (CD45.1)  (Jackson #002014)<br>Age: 12 weeks<br>Sex: Equal numbers of males and females<br><br>Npm1c/Flt3ITD/Cas9 (Huntly lab) - Primary leukemic cells obtained from bone-marrow tumours  were derived from this strain.<br>Age: 12 weeks<br>Sex: Female<br><br>Murine ethical compliance was fulfilled under the Guidelines of the Care and Use of Laboratory Animals and were approved by the Institutional Animal Care and Use Committees at University of Navarra, Spain, and the Animal Welfare Ethical Review Body at the University of Cambridge, UK. Research in the UK was conducted under Home Office license PP3042348. |
| --- | --- |
| Wild animals | The study does not involve wild animals |
| Reporting on sex | Equal numbers of female and males were used to:<br>- Obtain Haematopoietic progenitors for Bulk CRISPR screens and Perturb-seq experiments<br>- Isolate cell populations for ChIP-seq |
| Field-collected samples | The study does not involve field-collected samples |

Ethics oversight | All animal procedures were completed in accordance with the Guidelines of the Care and Use of Laboratory Animals and were approved by the Institutional Animal Care and Use Committees at University of Navarra, Spain, and the Animal Welfare Ethical Review Body at the University of Cambridge, UK.

Note that full information on the approval of the study protocol must also be provided in the manuscript.

## Methodology

| | |
|---|---|
| Replicates | Every ChIP seq analysis was performed with two replicate independent ChIP-seq experiments except for Kmt2a and Kmt2d ChIP-seq in myeloid and erythroid progenitors, where due to the difficulty of getting enough cells numbers the analysis of Brd9 and Smarcb1 ChIP-seq patterns was prioritised. |

| | |
|---|---|
| Sequencing depth | GEO name   Total Reads   Aligned reads after trimming, mapping, removing duplicates and blacklisted peeaks<br>Kmt2a-DM_rep1   71556716   52135927<br>Kmt2a-DM_rep2   20458692   16266441<br>Kmt2a-exvivoMonocytes_rep1   69985617   41947055<br>Kmt2a-exvivoMonocytes_rep2   12682223   9845731<br>Kmt2a-Ery   82954531   56071969<br>Kmt2a-GMP   73167956   53421804<br>Kmt2a-MEP   97640262   70833084<br>Kmt2a-Bcell   67491226   44742639<br>Kmt2a-Monocytes_rep1   32369273   25607927<br>Kmt2a-Monocytes_rep2   34539498   26022063<br>Kmt2d-exvivoMonocytes_rep1   41825984   26001792<br>Kmt2d-exvivoMonocytes_rep2   44949940   24358640<br>Kmt2d-exvivoMonocytes_rep3   72158751   48488577<br>Kmt2d-DM_rep1   25995989   16107654<br>Kmt2d-DM_rep2   90366626   59794611<br>Kmt2d-GMP   116393335   63951963<br>Kmt2d-MEP   43313559   65504507<br>Kmt2d-Ery_rep1   35064637   20413963<br>Kmt2d-Ery_rep2   37652595   22496310<br>Kmt2d-Bcell   38868015   23667460<br>Kmt2d-Monocytes_rep1   27979280   19401634<br>Kmt2d-Monocytes_rep2   23938072   16373041<br>Smarcb1-exvivoMonocytes_rep1   52835039   32356601<br>Smarcb1-exvivoMonocytes_rep2   41122100   29455975<br>Smarcb1-exvivoMonocytes_rep3   44557784   27003578<br>Smarcb1-DM_rep1   42890145   28328521<br>Smarcb1-DM_rep2   104296115   75890008<br>Smarcb1-GMP   92342249   58562130<br>Smarcb1-MEP   46733792   24150364<br>Smarcb1-Monocytes_rep1   20789867   14779534<br>Smarcb1-Monocytes_rep2   26354572   18880512<br>Smarcb1-Ery   18975965   13336679<br>Brd9-DM_rep1   65031476   43919123<br>Brd9-DM_rep2   95350112   67667153<br>Brd9-exvivoMonocytes   27240753   19317290<br>Brd9-GMP_rep1   52193534   36246200<br>Brd9-GMP_rep2   23607799   16429228<br>Brd9-MEP_rep1   62518967   46671329<br>Brd9-MEP_rep2   24865443   18494644<br>Brd9-MEP_rep3   40688029   28876910<br>Brd9-Ery_rep1   17103452   11946664<br>Brd9-Ery_rep2   24144295   16847147<br>Brd9-Monocytes_rep1   26536687   18426771<br>Brd9-Monocytes_rep2   26333906   19067019<br>Brd9-Bcell   14051972   9065470<br>Stat5a-DM   63783822   30893003<br>Stat5a-exvivoMonocytes   54541020   23981871 |

| | |
|---|---|
| Antibodies | SMARCB1/BAF47 (D8M1X) Rabbit mAb  Cell Signalling 91735 Lot 2<br>Anti-BRD9 antibody Abcam ab137245 Lot GR3372527-6<br>Anti-KMT2D antibody produced in rabbit  Sigma HPA035977<br>Kmt2a MLL1 Antibody Bethyl A300-086A Lot 6<br>IgG Rabbit IgG, polyclonal - Isotype Control (ChIP Grade) 100 ug Abcam ab171870<br>Stat5a Recombinant Anti-STAT5a antibody [E289]  Abcam ab32043 Lot GR3238474-7<br>Runx1 abcam ab23980  lot GR3213439-2<br>Runx2 abcam ab236639 lot GR3388032-15<br>Cebpa Abcam ab40764<br>CEBPE Sigma-Aldrich HPA002928 |

| | |
|---|---|
| Peak calling parameters | ChIP-seq reads were aligned to the GRCm38/mm10 reference genome assembly using Bowtie version 2.3.4.2 with parameters -X 1000 --no-discordant --no-mixed --very-sensitive. Peaks were called using MACS v2.2.7.1 with parameters -f BAMPE--keep-dup all and IgG as control. |

| | |
|---|---|
| Data quality | We followed the ChIP-seq nf-core pipeline. ChIP-seq reads were trimmed with default parameters using Trim Galore with Cutadapt. ChIP-seq reads were aligned to the GRCm38/mm10 reference genome assembly using Bowtie with parameters -X 1000 --no-discordant --no-mixed --very-sensitive. |

# ChIP-seq

## Data deposition

☒ Confirm that both raw and final processed data have been deposited in a public database such as GEO.

☒ Confirm that you have deposited or provided access to graph files (e.g. BED files) for the called peaks.

**Data access links**
*May remain private before publication.*

ChIP-seq datasets of CFs in vivo, ex vivo and leukaemic: GEO accession (GSE213507)
To review GEO accession GSE213513:
Go to https://www.ncbi.nlm.nih.gov/geo/query/acc.cgi?acc=GSE213513
Enter token whufyuosxjirrgx into the box

**Files in database submission**

GSM6588342 Smarcb1-GMP
GSM6588343 Smarcb1-MEP
GSM6588344 Smarcb1-Ery
GSM6588345 Smarcb1-Monocytes_rep1
GSM6588346 Smarcb1-Monocytes_rep2
GSM6588347 Smarcb1-exvivoMonocytes_rep1
GSM6588348 Smarcb1-exvivoMonocytes_rep2
GSM6588349 Smarcb1-exvivoMonocytes_rep3
GSM6588350 Smarcb1-DM_rep1
GSM6588351 Smarcb1-DM_rep2
GSM6588352 Brd9-GMP_rep1
GSM6588353 Brd9-GMP_rep2
GSM6588354 Brd9-MEP_rep1
GSM6588355 Brd9-MEP_rep2
GSM6588356 Brd9-MEP_rep3
GSM6588357 Brd9-Ery_rep1
GSM6588358 Brd9-Ery_rep2
GSM6588359 Brd9-Bcell
GSM6588360 Brd9-Monocytes_rep1
GSM6588361 Brd9-Monocytes_rep2
GSM6588362 Brd9-exvivoMonocytes
GSM6588363 Brd9-DM_rep1
GSM6588364 Brd9-DM_rep2
GSM6588365 Kmt2d-GMP
GSM6588366 Kmt2d-MEP
GSM6588367 Kmt2d-Ery_rep1
GSM6588368 Kmt2d-Ery_rep2
GSM6588369 Kmt2d-Bcell
GSM6588370 Kmt2d-Monocytes_rep1
GSM6588371 Kmt2d-Monocytes_rep2
GSM6588372 Kmt2d-exvivoMonocytes_rep1
GSM6588373 Kmt2d-exvivoMonocytes_rep2
GSM6588374 Kmt2d-exvivoMonocytes_rep3
GSM6588375 Kmt2d-DM_rep1
GSM6588376 Kmt2d-DM_rep2
GSM6588377 Kmt2a-GMP
GSM6588378 Kmt2a-MEP
GSM6588379 Kmt2a-Ery
GSM6588380 Kmt2a-Bcell
GSM6588381 Kmt2a-Monocytes_rep1
GSM6588382 Kmt2a-Monocytes_rep2
GSM6588383 Kmt2a-exvivoMonocytes_rep1
GSM6588384 Kmt2a-exvivoMonocytes_rep2
GSM6588385 Kmt2a-DM_rep1
GSM6588386 Kmt2a-DM_rep2
GSM6588387 Stat5a-exvivoMonocytes
GSM6588388 Stat5a-DM
GSM6588389 IgG-DM
GSM6588390 IgG-exvivoMonocytes_rep1
GSM6588391 IgG-exvivoMonocytes_rep2

**Genome browser session**
(e.g. UCSC)

https://genome.ucsc.edu/s/julenm/Lara-Astiaso_et_al

We removed duplicated regions with Picard Tools and filtered out ENCODE blacklist regions and non-interesting chromosomes.
We pooled replicates.
Peaks were called using MACS with parameters -f BAMPE--keep-dup all.
Peaks at FDR 5% and above 5-fold enrichment vs IgG:

GMP_Smarcb1_ChIP11_peaks.narrowPeak 17325
MEP_Smarcb1_ChIP12_peaks.narrowPeak 5453
Leukaemia_Smarcb1_peaks.narrowPeak 15777

Exvivo-Mono_Smarcb1_peaks.narrowPeak 16333
Mono_Smarcb1_peaks.narrowPeak 11194
Ery_Smarcb1_peaks.narrowPeak 30

GMP_Brd9_peaks.narrowPeak 5124
MEP_Brd9_peaks.narrowPeak 8371
Exvivo-Mono_Brd9_peaks.narrowPeak 11648
Mono_Brd9_peaks.narrowPeak 11719
Bcell_Brd9_peaks.narrowPeak 3118
Ery_Brd9_peaks.narrowPeak 2168

GMP_Kmt2d_peaks.narrowPeak 22114
MEP_Kmt2d_peaks.narrowPeak 3111
Leukaemia_Kmt2d_peaks.narrowPeak 29038
Exvivo-Mono_Kmt2d_peaks.narrowPeak 18123
Mono_Kmt2d-merged_peaks.narrowPeak 7567
Bcell_Kmt2d_peaks.narrowPeak 8285
Ery_Kmt2d_peaks.narrowPeak 1266

GMP_Kmt2a_peaks.narrowPeak 33869
MEP_Kmt2a_peaks.narrowPeak 28581
Leukaemia_Kmt2a_peaks.narrowPeak 17532
Exvivo-Mono_Kmt2a_peaks.narrowPeak 12585
Mono_Kmt2a_peaks.narrowPeak 1346
Ery_Kmt2a_peaks.narrowPeak 5413
Bcell_Kmt2a_peaks.narrowPeak 13741

Leukaemia_Stat5a_peaks.narrowPeak 26878
Exvivo-Mono_Stat5a_peaks.narrowPeak 16857

| Software | Trim Galore  v0.6.6<br>Cutadapt v3.4<br>Bowtie v2.3.4.2<br>Picard v2.25.4<br>ENCODE blacklist regions v2.0<br>MACS v2.2.7.1<br>Code is available at https://github.com/csbg/tfcf/tree/main/ATAC_ChIP/ChIP |
|---|---|

# Flow Cytometry

## Plots

Confirm that:

☒ The axis labels state the marker and fluorochrome used (e.g. CD4-FITC).

☒ The axis scales are clearly visible. Include numbers along axes only for bottom left plot of group (a 'group' is an analysis of identical markers).

☒ All plots are contour plots with outliers or pseudocolor plots.

☒ A numerical value for number of cells or percentage (with statistics) is provided.

## Methodology

| Sample preparation | Sorting of haematopoietic progenitors for Exvivo CRISPR Screens<br>Femora, tibiae, ilia, humerus, sternum and scapula were harvested from 12-14 week old C57BL/6J and ROSAxCas9 mice (equal ratio of males and females), crushed with a pestle and mortar using cold (4 ºC) autoMACS Running Buffer and filtered through a 70 µM strainer. Red Blood Cells were lysed using RBC Lysis Buffer and c-Kit+ cells were enriched using mouse CD117 magnetic beads (Miltenyi) , following the manufacturer's protocol. The c-Kit enriched fraction was stained with anti-Lineage (B220, CD3, CD11b, Gr1, Ter-119), anti-CD117 (cKit) and PE anti-Sca1. Lin-/cKit+/Sca1+ hematopoietic progenitors cells (LSKs) were FACS-sorted in 1 mL of DMEM/F12 + 1X Pen/Strep.<br><br>Ex vivo CRISPR FACS Readouts.<br>Cultures were harvested by centrifugation at 300 g for 5 minutes and washed twice with 1X cold PBS. Then the cell pellets were stained with the Readout specific cocktails (see below) plus a viability marker (TOPRO or Propidium Iodide). Viable BFP+ cells (containing CRISPR guides) were gated from Cas9 (GFP+) and Non-Cas9 (GFP-) fractions and, from each fraction the readout populations (see below) were sorted in 1.5 mL tubes containing PBS + 0.1% BSA.<br><br>In vivo CRISPR Screens<br>Femora, tibiae, ilia, humerus, sternum and scapula were harvested from bone marrow transplanted B6.SJL-Ptprca Pepcb/BoyJ (CD45.1), crushed with a pestle and mortar using cold (4 ºC) autoMACS Running Buffer and filtered through a 70 µM |
|---|---|

strainer. Red Blood Cells were lysed using RBC Lysis Buffer and c-Kit+ cells were enriched using mouse CD117 magnetic beads (Miltenyi) , following the manufacturer's protocol. The purified cKit+ fraction was stained with TOPRO (viability), anti-Lineage (CD3, CD19, Ter119, CD11b, Gr1) and anti-CD117 (cKit) antibodies. For single-cell RNAseq we FACS-sorted 200,000 viable (TOPRO-), GFP+ (Cas9), BFP+ (sgRNA) cells from Lineage- and Lineage+/cKit+ fractions and processed each of them in a 10X single-cell RNA-seq partition aiming at a final coverage of 500 single-cells per sgRNA.

Isolation of in vivo hematopoietic cells for ChIP-seq.
Murine haematopoietic cells were harvested from 12-14 week old C57BL6 mice (balanced numbers of males and females) as described above and stained for the isolation of:
GMP: Lineage (CD3, CD19, CD11b, Gr1, Ter119, B220)-, cKit+, Sca-1-, FcgRIII+, CD34+
MEP: Lineage (CD3, CD19, CD11b, Gr1, Ter119, B220)-, cKit+, Sca-1-, FcgRIII-, CD34-
Monocytes: CD3-, CD19-, Ter119-, CD11b+
B-cell: CD3-, CD19+, Ter119-, CD11b-
Erythroid cells were FACS-sorted from spleens of 12 week-old C57BL6 mice as: CD3-, CD19-, Ter119+, CD11b-, Gr1-.
Cells were sorted in PBS + 0.1% BSA and crosslinked immediately after sorting

Analysis of leukaemic populations exvivo.
Npm1c/Flt3-ITD/Cas9 double mutant (DM) cells were generated from lineage-depleted, bone marrow cells of primary transgenic mice post-leukemic onset. Cells were maintained in XVIVO-20 medium (Lonza) supplemented with 5% Fetal Bovine Serum (FBS) (ThermoFisherScientific), 1% PSG (Gibco), murine SCF 50 ng/mL (PeproTech), murine IL-3 10 ng/mL (PeproTech) and murine IL-6 10 ng/mL (R&D Systems), in a 37oC and 5% CO2 atmospheric environment. Npm1c/Flt3-ITD/Cas9 double mutant (DM) cells were passaged every 2 days and cultured for short time (passage 3-5) to maintain the original leukaemic properties.
For FACS analysis, cells were washed twce with ice-cold PBS and stained with:
 CD11b (PE-Cy7 conjugated; clone M1/70; BD Biosciences),
Gr-1 (Ly6G/Ly6C; APC-Cy7 conjugated; RB6-8C5 clone; BD Biosciences),
CD55 (PE-conjugated; clone RIKO-3; Biolegend),
CD41 (APC-conjugated; clone MWReg30; Biolegend) and
CD34 (FITC-conjugated; clone RAM34; BD Bioscience).
Gran-like (CD11b-high/Gr-1+); Ery/Baso-like (CD55-high/CD41-) and CD34+ fractions were subsequently FACS-sorted (BD Influx; BD Bioscience)

| Instrument | BD LSR Fortessa II; BD Biosciences<br>BD Influx; BD Bioscience |
| --- | --- |
| Software | FlowJo (version 10.8.0) |
| Cell population abundance | The purity of the post sorting fractions was further characterized with scRNA-seq |
| Gating strategy | LSK purificaton<br>1- Exclude doublets<br>2- Remove debris<br>3- Gate Lineage-negative cells<br>4- Gate ckit-positive, Sca1-positive<br><br>FACS Readouts in CRISPR screens (Progenitor vs Differentated):<br>1- Exclude doublets<br>2- Remove debris<br>3- Gate GFP-positive (Cas9), BFP-ckit-positive, Sca1-positive<br>4- Gate Lineage-negative cells<br>5- Gate:<br>a) ckit-positive, Sca1-positive = Multipotent Progenitors<br>b) ckit-positive, Sca1-negative = Differentiated<br><br>FACS Readouts in CRISPR screens (Myeloid vs Mega-erythroid)<br>1- Exclude doublets<br>2- Remove debris<br>3- Gate GFP-positive (Cas9), BFP-ckit-positive, Sca1-positive<br>4- Gate Lineage-negative cells<br>5- Gate ckit-positive, Sca1-negative<br>6- Gate:<br>a) FcgR-III positive = Myeloid progenitors<br>b) FcgR-III negative = Mega-erythroid progenitors<br><br>FACS Readouts in CRISPR screens (Myeloid vs non-myeloid)<br>1- Exclude doublets<br>2- Remove debris<br>3- Gate GFP-positive (Cas9), BFP-positive (sgRNA)<br>4- Gate:<br>a) FcgR-III positive, CD11b-postive = Myeloid fraction<br>b) FcgR-III negative, CD11b-negative = Non-myeloid<br><br>FACS Readouts in CRISPR screens (Terminal myeloid differentiation)<br>1- Exclude doublets |

2- Remove debris
3- Gate GFP-positive (Cas9), BFP-positive (sgRNA)
4- Gate:
a) Gr1-positive, CD11b-postive = Mature Myeloid cells
b) Gr1-negative, CD11b-negative = Myeloid progenitors

FACS-sorting for in vivo Perturb-seq
1- Exclude doublets
2- Gate viable cells
3- Gate BFP-positive (CRISPRed cells)
4- Gate:
a) Lineage-negative= Lin-negative fraction
b) Lineage-positive, ckt-positive= Lin-neg/ckit-pos fraction

FACS-sorting for isolation of leukemic cells
1- Exclude doublets
2- Gate viable cells
3- Gate Lineage-negative cells

FACS-sorting for Perturb-seq in Leukemia
1- Exclude doublets
2- Gate viable cells
3- Gate BFP-positive (CRISPRed cells)

Exemplar FACS plots can be found in the Supplementary Materials

☒ Tick this box to confirm that a figure exemplifying the gating strategy is provided in the Supplementary Information.

