## [Peer Review File · Nature Genetics]

Peer Review Information

Manuscript Title: In vivo screening characterizes chromatin factor functions during normal and malignant hematopoiesis

Corresponding author name(s): Professor Brian Huntly, Dr Felipe Prosper, Professor Nikolaus Fortelny

Reviewer Comments & Decisions:

Decision Letter, initial version:

7th Nov 2022

Dear Brian,

Your Article, "Systematic study of chromatin factor function uncovers strong lineage determining roles and divergent behaviours between normal and malignant haematopoiesis" has now been seen by 3 referees. You will see from their comments below that while they find your work of interest, some important points are raised. We are interested in the possibility of publishing your study in Nature Genetics, but would like to consider your response to these concerns in the form of a revised manuscript before we make a final decision on publication.

Briefly, the reviewers all sound appreciative of your manuscript and - at this stage - supportive of an eventual publication.

Reviewer #1 provides a strikingly positive report. They suggest that the manuscript is suitable for publication in its current state.

Reviewer #2 is, conversely, more circumspect. They make a number of requests for technical clarification that, to our reading, sound thoughtful and would improve the study in terms of understanding the robustness of your screens. They have also, pleasingly, suggested multiple options in case the best one is too onerous!

Reviewer #3 strikes the same positive tone as Referee #1. That said, they do also have some requests for improvement.

To guide the scope of the revisions, the editors discuss the referee reports in detail within the team, including with the chief editor, with a view to identifying key priorities that should be addressed in revision and sometimes overruling referee requests that are deemed beyond the scope of the current study. We hope that you will find the prioritized set of referee points to be useful when revising your study. Please do not hesitate to get in touch if you would like to discuss these issues further.

We therefore invite you to revise your manuscript taking into account all reviewer and editor comments. Please highlight all changes in the manuscript text file. At this stage we will need you to upload a copy of the manuscript in MS Word .docx or similar editable format.

*2) If you have not done so already please begin to revise your manuscript so that it conforms to our Article format instructions, available [here](http://www.nature.com/ng/authors/article_types/index.html). Refer also to any guidelines provided in this letter.

[redacted]

Sincerely,

Michael Fletcher, PhD
Senior Editor, Nature Genetics

ORCID: 0000-0003-1589-7087

Referee expertise:

Referee #1: CRISPR screens, chromatin factor complexes.

Referees #2 and #3: haematopoiesis, leukaemia, single-cell analyses, genomics.

Reviewers' Comments:

Reviewer #1:

Remarks to the Author:

The manuscript by Lara-Astiaso describes a high-throughput functional-genetic screening study that seeks to comprehensively define requirements for chromatin regulators during normal and malignant hematopoiesis. This study is performed on a heroic scale using a multitude of -omics methods in a manner I have never seen attempted previously. While ambitious at every level, I am particularly impressed with the rigorous quality control performed throughout to ensure that each finding reported in the main figure is well-supported by the orthogonal assays. In other words, I would rate the technical merits of this work very highly. While the value of this work in the field will ultimately be as a resource article, I believe there are a number of non-obvious findings specific in nature that elevates the work. For example, the contextual requirements of Kmt2d and Brd9 in myeloid versus lymphoid lineages, as well as in cancer versus normal is provocative and compelling based on the data presented. In addition, the mechanistic depth was suitable for this work – linking CFs to TFs via motif analysis is highly appropriate and will lay a strong foundation for future biochemical investigation. Overall, I am very impressed with this study. So much so, that I will not ask for any additional experiments to improve this work. The authors have done enough to justify publication, and for the reasons listed above I support publication without delay.

Reviewer #2:

Remarks to the Author:

The manuscript "Functional study of chromatin factors uncovers strong lineage determining roles and divergent behaviours between normal and malignant haematopoiesis" by Lara-Astiaso, Goñi-Salaverri and colleagues aims to identify chromatin factors (CFs) involved in HSC differentiation. Using CRISPR screens in ex vivo and single-cell in vivo approaches (Perturb-seq), the authors could identify CFs with lineage-specific dependencies. The authors report different lineage specificities for factors from related epigenetic complexes as well as important interactions between CFs and key transcription factors identified by analyzing chromatin accessibility at TF-binding sites. Lastly, the study characterizes the aberrant roles of certain CFs in leukemia, which were found to block myeloid differentiation through interactions with alternative transcription factors.

This study provides a comprehensive analysis of the role of CFs in murine hematopoietic differentiation ex vivo and in vivo. The study fits the journal scope and is of interest to the field, in particular the Perturb-seq dataset could improve the understanding of the exact role of certain epigenetic factors in HSC differentiation. However, some of the approaches and technologies used are not always well explained or justified as well as some results would benefit from an improved interpretation/

contextualization. Thus, there are some limitations to the study that could be further discussed as well as some important points that should be addressed:

Limitations:

- Due to the *ex vivo* culture conditions on which the pre-screen is based, and the time point chosen for transplantation, the focus of this study is on CFs mediating myeloid or erythroid differentiation. This could be discussed more explicitly in the discussion section.
- The *in vivo* setting is based on transplantations. Similar to some of the discrepancies that have been observed between *ex vivo* and *in vivo* CRISPR screens, there might be differences between the transplantation setting that the authors have used and an unperturbed *in vivo* situation. This could be discussed more explicitly in the discussion sections.

Conceptual point:

Some of the interpretation of the results would benefit from a more in-depth contextualization with the already established literature on epigenetics. Although, the authors classify the CFs based on known epigenetic complexes, often throughout the manuscript they do not take into account already well-characterized functional differences between them. As illustrative examples:

o In line 110: "SET1 components were required to initiate differentiation but displayed antagonistic behaviors at lineage branching points, where *Wdr82* and *Setd1b* acted as a pro-myeloid factor, but *Cxxcl1* and *Setd1a* demonstrated mega-erythroid dependency; suggesting the existence of functionally diverse SET1 complexes". This sentence suggests that the existence of functionally diverse SET1 complexes is something new or not known, while *Setd1a* and *Setd1b* complexes have been well-characterized in the past.

o In line 176: "these results demonstrate a lack of redundancy between the three writers of H3K4 methylation, which suggest that cell-type specific H3K4 methylation is mediated by particular epigenetic writers in different cellular lineages". And line 364: "This shows that, despite depositing similar epigenetic marks, the different H3K4-methyltransferases are not redundant and suggests that they individually regulate specific H3K4me patterns, or that their distinct roles are mediated via catalytic-independent activities".

The distinct H3K4 methyltransferases are known to deposit H3K4me1 (MLL4) or H3K4me3 (MLL1 and SET1). These marks are enriched in either enhancers or active promoters, respectively. Thus, disruption of these distinct methyltransferases would most likely result in aberrant expression of distinct sets of genes, which could explain the distinct lineage dependencies.

Specific points:

1. The inclusion of more guide controls for known TFs would have been highly valuable to assess effect size. While we do not expect to re-run the screen, maybe smaller control assays with guides for some targets can be re-run jointly with known TFs regulating self-renewal vs differentiation or myeloid vs erythroid priming.
2. Replicates appear to be missing in several scenarios making the evaluation of the robustness of some of the effects difficult. How robust are the results derived from the CRISPR screen? Have they been performed in replicates? If so please provide information on robustness, if not it would be appreciated if this can be done, at least, for selected candidates.
3. While *in vivo* CRISPR knock out screens are not possible in humans, it would be of interest to what degree the findings obtained in this study are conserved in humans. Can published CHIP or ATAC-seq datasets be employed to at least partly address this.
4. Can the authors elaborate on the inclusion of two read-outs to assess the CF-KO lineage-specific dependencies in the Perturb-seq data? Why not only use CF-KO enrichment/reduction in particular clusters instead of entire lineages? This would allow the identification of the exact cell states that are altered without the need of pseudotime. Alternatively, the authors could also use differential abundance testing in a continuous manner rather than using discrete clusters using, for instance, the miloR package. If the authors come to the conclusion that their approach is superior to our

suggestions, feel free to keep it that way.

5. In Figure 3A, can the authors please clarify whether the gene expression was analyzed by cell cluster or using all cells from the specific CF-KO? If the latter, the results could be reflecting the aberrant cell abundancies rather than the molecular mechanisms underlying the CF depletion as suggested by the authors.

6. In the ATAC-seq experiment shown in Fig.3, did the authors only analyze the differentiated cell types? Only analyzing the end-points of differentiation might preclude the identification of dynamic changes occurring during differentiation. The authors could think about performing scATAC-seq instead, for instance with Spear-ATAC. This is not an absolute must, maybe alternative approaches could be used to provide insights into dynamics.

(<https://www.nature.com/articles/s41467-021-23213-w>).

7. Section 246 following. Are there concrete indications that disruptions of ncBAF is indeed involved in leukemogenesis? Can the authors demonstrate that disruptions of ncBAF functionally predisposes to leukemias? If not, the term "pre-leukemic" should be avoided.

8. Are the cellular fitness measurements shown in Fig. 5B a specific effect of depleting the CFs in leukemia cells or is cell viability also affected in the CF-KO in normal progenitors? Can the authors show the viability for the CFs used in the Perturb-seq?

9. Loss of Smarcd2 or Kmt2d in leukemic cells was shown to lead to reduced accessibility in particular Stat5a loci, but this was only shown for two particular examples. Could the authors show the loss of accessibility at Stat5a-bound loci in a more systematic way?

Minor points:

- In general, methods and samples used in the different experiments are not always clear to the reader (i.e. what cell types are analyzed in the ATAC-seq in Fig.3?; what was the reasoning for using CITE-Seq and why is it only mentioned in the discussion but not the main text?).
- Figures are not always called or are mistakenly called in the main text (i.e. Fig. 4C is not called and in line 281 Fig. 4J is called when it should be Fig. 4I). Please check throughout the text.
- Significance is missing in Fig. 6H. Please provide statistics.

Reviewer #3:

Remarks to the Author:

In this paper, the authors perform a very elegant CRISPR screen to determine which chromatin proteins, out of a list of 550, contribute to specific lineage decisions as well as maintenance of multipotency. They validated a few key factors and then performed an in vivo Perturb-seq experiment with 40 key factors. One of the more interesting things to come out of this is the high level of specificity observed between highly related complexes – for example, expansion of erythroid cells with Kmt2d KO vs expansion of B cells with a Kmt2a KO. They then went on to explore TF expression profiles and in particular used digital footprinting (with ATAC-seq data) to determine which TF motifs were associated with chromatin protein function. They explore the differential role of cBAF and ncBAF complexes in more detail, and interrogate the function of several chromatin factors in an NPM1/Flt3 AML model. Interestingly, despite the differential impact on normal development, the different chromatin proteins all seemed to display a uniform impact on decreased growth of leukaemia cells, except for Brd9. Finally, they look at changes in the distribution of key chromatin proteins in normal versus leukemic cells.

Overall, this is a very interesting screen and analysis that will provide a fantastic ongoing resource for the field, and could potentially provide a useful platform for drug discovery.

I have a few questions and points that need to be clarified that will make the paper more understandable, but I want to make it clear that I find this paper quite exciting and will enjoy

returning to it again in the future:

1. In Figure 3E, the digital footprinting analysis, is the loss of a TF footprint due to the loss of expression of the TFs, or are the TFs still expressed in many cases but have simply lost accessibility to their binding sites?
2. Related to (1), digital footprinting using ATAC-seq can be prone to bias, and the presence/absence of a motif is not the same as the presence/absence of a TF. Can ChIP-seq for some key TFs be used to validate this TF binding profile change in the different chromatin protein KOs?
3. The ChIP-seq heat maps indicate that Smarcb1, Brd9 and Kmt2a/2d have quite different binding profiles, but it is a little unclear what the example tracks in Extended Figure 6A are meant to display. The quality of the ChIP-seq looks excellent, but the example tracks look quite similar to me, and the panel seems to display genes where the binding profile for these 4 proteins is quite similar (although it changes with differentiation). This is fine as it is, but it would also be useful to see some example tracks for differential gene target binding e.g. an important gene that binds Smarcb1 but not Brd9 for instance.
4. Line 267-269: "This suggests that the myeloid maturation defect observed in Brd9-KO cells may relate to impaired recruitment and activity of Cebp and Ap1 TFs." I'm not sure I understand this statement. If we take the motif enrichment at face value for a moment, does this statement imply that the chromatin proteins are responsible for recruiting TFs? This is quite a different view from the way the field thinks, and I'm not sure I follow the authors logic as to how their data supports this interpretation. A bit more explanation would be helpful.
5. Again, it would be useful to know if the digital footprinting analysis for Figure 4H and 4I can be validated with at least some TF ChIP-seq in NTC vs Brd9 KO cells.
6. Menin inhibitors are currently in clinical trials for NPM1 mutant leukemias. Do Menin inhibitors produce a similar profile as Kmt2a and Men1 in normal cells and in the Npm1/Flt3 model? I am thinking of the in vitro differentiation experiments. This may seem to be a side issue, but I am thinking that one use of the system(s) outlined here (especially the in vitro differentiation) would be for future drug discovery, and it would be useful to see how a highly specific drug that is currently going through clinical trials performs relative to the gene specific and complex KOs.
7. As already mentioned, I overall like this paper quite a bit and think it is interesting and provides important results. Something I already mentioned is that I am still not quite sure what the authors are trying to say about the relationship between TF binding and chromatin protein activity. There is an over reliance on ATAC-seq and motif analysis to represent TF binding (except for the specific case of Stat5a), which is not a good substitute for TF specific ChIP-seq. In addition, reduced ATAC signal in the presence of a KO (e.g. Figure 6G) does not necessarily indicate that the chromatin factor is directly responsible for maintaining open chromatin, as the effect could be indirect (e.g. loss of expression of a remodeling factor in the KO). If the authors are trying to say that chromatin protein activity is responsible for TF recruitment, I think they need to clarify how their data supports this view. I think the data is still consistent with the view that TFs are responsible for recruiting specific chromatin protein complexes, and the specificity of these different complexes comes from both differential TF-CF dependencies (for example, like that seen in this recent paper Hendy et al 2022, PMID: 36113480) and/or possibly from different chromatin protein-promoter or enhancer specificities (e.g. Neumayr et al 2022 PMID: 35650434; Haberle et al 2019 PMID: 31092928).

Minor points

1. The legend for Figure 4 skips D.

2. Figure 4 panel, the Day 28 and Day 14 labels are reversed, I think. If they are not, then the authors statement that Brd9 KO cells are expanded is not correct.

Author Rebuttal to Initial comments

Response to the reviewers and editors for revised manuscript:

Functional study of chromatin factors uncovers strong lineage determining roles and divergent behaviours between normal and malignant hematopoiesis

We are delighted to have the opportunity to resubmit our revised manuscript for reconsideration. In doing so we have very carefully considered the views and excellent suggestions of the three reviewers. We have addressed these as below, in a point-by-point manner, with our responses in blue and specific references to the manuscript in red.

Reviewer #1:

Remarks to the Author:

The manuscript by Lara-Astiaso describes a high-throughput functional-genetic screening study that seeks to comprehensively define requirements for chromatin regulators during normal and malignant hematopoiesis. This study is performed on a heroic scale using a multitude of -omics methods in a manner I have never seen attempted previously. While ambitious at every level, I am particularly impressed with the rigorous quality control performed throughout to ensure that each finding reported in the main figure is well-supported by the orthogonal assays. In other words, I would rate the technical merits of this work very highly. While the value of this work in the field will ultimately be as a resource article, I believe there are a number of non-obvious findings specific in nature that elevates the work. For example, the contextual requirements of Kmt2d and Brd9 in myeloid versus lymphoid lineages, as well as in cancer versus normal is provocative and compelling based on the data presented. In addition, the mechanistic depth was suitable for this work – linking CFs to TFs via motif analysis is highly appropriate and will lay a strong foundation for future biochemical investigation. Overall, I am very impressed with this study. So much so, that I will not ask for any additional experiments to improve this work. The authors have done enough to justify publication, and for the reasons listed above I support publication without delay.

We thank the reviewer for their exceptionally kind words and are very pleased that they feel that our work will both be important as a resource but also in relation to individual mechanistic insights such as the contextual requirements of Kmt2d and Brd9 in myeloid versus lymphoid lineages, as well as in cancer versus normal regulation. We are particularly pleased that they have noted the rigour that we have tried to conduct the study with across every layer. Due to the novelty and scale of our screens, we have held a very high bar to first convincing ourselves of the veracity of the data, prior to validating it in orthogonal studies and appreciate that this has been picked up and commented upon by this reviewer.

Reviewer #2:

Remarks to the Author:

The manuscript “Functional study of chromatin factors uncovers strong lineage determining roles and divergent behaviours between normal and malignant haematopoiesis” by Lara-Astiaso, Goñi- Salaverri and colleagues aims to identify chromatin factors (CFs) involved in HSC differentiation. Using CRIPSR screens in *ex vivo* and single-cell *in vivo* approaches (Perturb-seq), the authors could identify CFs with lineage-specific dependencies. The authors report different lineage specificities for factors from related epigenetic complexes as well as important interactions between CFs and key transcription factors identified by analyzing chromatin accessibility at TF- binding sites. Lastly, the study characterizes the aberrant roles of certain CFs in leukemia, which were found to block myeloid differentiation through interactions with alternative transcription factors.

This study provides a comprehensive analysis of the role of CFs in murine hematopoietic differentiation *ex vivo* and *in vivo*. The study fits the journal scope and is of interest to the field, in particular the Perturb-seq dataset could improve the understanding of the exact role of certain epigenetic factors in HSC differentiation. However, some of the approaches and technologies used are not always well explained or justified as well as some results would benefit from an improved interpretation/contextualization. Thus, there are some limitations to the study that could be further discussed as well as some important points that should be addressed:

We thank this reviewer for their generally very supportive comments and for their constructive criticism. We have addressed their concerns below:

Limitations:

- Due to the *ex vivo* culture conditions on which the pre-screen is based, and the time point chosen for transplantation, the focus of this study is on CFs mediating myeloid or erythroid differentiation. This could be discussed more explicitly in the discussion section.

We thank the reviewer for this insightful observation. We completely agree that the limitations of the *ex vivo* culture system for the initial and larger screen focus on the twin trajectories of myeloid and erythroid differentiation. We have highlighted the limitations of the study in a specific paragraph in the discussion section, where we mention the myeloid and erythroid bias of our *ex vivo* screen system and propose alternative culture conditions that could be substituted to interrogate B-cell and T-cell lineages.

In the same paragraph, we also discuss that the *in vivo* Perturb-seq has been performed in the setting of emergency haematopoiesis where, upon reconstitution myeloid and erythroid lineages are prioritised. However, we would also like to respectfully point out that, despite the augmented erythroid and myeloid trajectories, the detailed *in vivo* screen performed for the prioritized CF permitted interrogation of almost all hematopoietic lineages, except for the T-cell lineage.

For example:

- This screen allowed us to detect a strong dependency of B-cells on ncBAF complex-member genes (*Brd9* and *Smarca1*), that phenocopy *Pax5* and *Ebf1* (known B-cell master regulator TFs)-KOs and that are consistent with the *Ebf1* motif-enrichment at B

- cell-specific Brd9- bound loci.
- The screen further allowed the detection of CF-KOs that upregulate differentiation towards lineages difficult to interrogate *ex vivo*. These include Megakaryocytes (SET1 members) or Basophils (upregulation of basophil markers upon *Ehmt1*-KO, *Yeats4*-KO as well as ncBAF members, *Brd9*- and *Gltsr1*-KO).

We therefore believe that our complementary strategy, bulk *ex vivo* functional screens and *in vivo* Perturb-seq, together form a powerful toolkit to interrogate regulatory roles of CF. The bulk screen allows the screening and prioritisation of candidates at scale and the selection of efficient sgRNAs. The *in vivo* Perturb-seq serves to dissect phenotypic effects for the prioritised CF under near physiological conditions. However, we do agree with the reviewer that a more diverse *ex vivo* system could permit a more thorough interrogation and we expect that this article sets the stage for such studies in the future.

- The *in vivo* setting is based on transplantations. Similar to some of the discrepancies that have been observed between *ex vivo* and *in vivo* CRISPR screens, there might be differences between the transplantation setting that the authors have used and an unperturbed *in vivo* situation. This

could be discussed more explicitly in the discussion sections.

We again thank the reviewer for this very valid point. We entirely agree that our screens are performed under the selective pressures of strong cytokine induction and regenerative stress. As explained above, we have discussed this further in the discussion section, and suggested methods to assess CF function in homeostatic haematopoiesis. These include transplantation of progenitors harbouring inducible Cas9 and transduced with CF libraries cloned in low immunogenic vectors, with subsequent induction of Cas9 expression and LOF once the haematopoietic system has fully regenerated post-transplantation.

Conceptual point:

Some of the interpretation of the results would benefit from a more in-depth contextualization with the already established literature on epigenetics. Although, the authors classify the CFs based on known epigenetic complexes, often throughout the manuscript they do not take into account already well-characterized functional differences between them.

We thank the reviewer for this comment and apologise that we were perhaps over-simplistic in our categorisation of epigenetic complexes and their composite members. In our revised manuscript we have been more thorough in the text, providing further references to the literature describing previous knowledge/literature on CF complex composition and roles.

As illustrative examples:

- o In line 110: "SET1 components were required to initiate differentiation but displayed antagonistic behaviors at lineage branching points, where *Wdr82* and *Setd1b* acted as a pro-myeloid factor, but *Cxxcl1* and *Setd1a* demonstrated mega-erythroid dependency; suggesting the existence of functionally diverse SET1 complexes". This sentence suggests that the existence of functionally diverse SET1 complexes is something new or not known, while *Setd1a* and *Setd1b* complexes have been well-characterized in the past.

We thank the reviewer for this comment. We agree and have now removed it from the discussion.

o In line 176: “these results demonstrate a lack of redundancy between the three writers of H3K4 methylation, which suggest that cell-type specific H3K4 methylation is mediated by particular epigenetic writers in different cellular lineages”. And line 364: “This shows that, despite depositing similar epigenetic marks, the different H3K4-methyltransferases are not redundant and suggests that they individually regulate specific H3K4me patterns, or that their distinct roles are mediated via catalytic-independent activities”.

The distinct H3K4 methyltransferases are known to deposit H3K4me1 (MLL4) or H3K4me3 (MLL1 and SET1). These marks are enriched in either enhancers or active promoters, respectively. Thus, disruption of these distinct methyltransferases would most likely result in aberrant expression of distinct sets of genes, which could explain the distinct lineage dependencies.

The reviewer is correct in suggesting that the catalytic differences between MLL1 and MLL4 could explain their distinct lineage dependencies. We have instead highlighted the differences between two H3K4me3 writers (MLL1 and SET1), which we have further characterised with expanded Perturb-seq analysis. We are aware that such functional diversity for SET1 HK4me3 writers is not novel, as a similar observation has been described for Kmt2b/MLL2 and Set1 in ES cell differentiation (PMID: 32637595, PMID: 28723559, PMID: 30355503). In view of this, we have mentioned this in the discussion, where we present our results as an additional exemplar of non- redundant roles of COMPASS complexes.

With respect to the H4K4me1 writers (MLL4/Kmt2d and MLL3/Kmt2c), we think it interesting to point out that, contrary to studies in ES cells (PMID: 28483418), we detect a clear lack of redundancy between these two factors. Here the sole loss of function of *MLL4/Kmt2d* is enough to cause major defects in lineage differentiation, both *ex vivo* and *in vivo*. It also leads to leukaemic exhaustion as reported previously (PMID: 25079327). Therefore, we believe that, even though this is not the major ambition of the paper, our study offers novel insights on the redundant and specific roles of the different H3K4me1/3 writers.

In addition, we have used the discussion to suggest the potential regulatory mechanisms by which COMPASS complexes mediate lineage specific trajectories. Of note, we detect major chromatin accessibility loss in CF-KOs targeting members of SET1, MLL1 and MLL4 complexes both in the normal and leukaemic settings. While this may simply reflect an indirect effect of the loss of the methylation marks, it may also argue for catalytically-independent regulation of chromatin and gene expression by these complexes, as highlighted in seminal studies by the Wysocka & Shilatifard labs. Another line of evidence for that is the extensive MLL1/ Kmt2a binding at distal regulatory elements that have no or low H3K4me3 – these observations potentially argue for additional regulatory roles exerted by MLL1 such as scaffolding other CF and/or in regulating DNA looping.

1. The inclusion of more guide controls for known TFs would have been highly valuable to assess effect size. While we do not expect to re-run the screen, maybe smaller control assays with guides for some targets can be re-run jointly with known TFs regulating self-renewal vs differentiation or myeloid vs erythroid priming.

We thank the reviewer for this excellent suggestion. In general, we agree with them, however, we believe that it would be more valuable to conduct such perturbations *in vivo*, since, as we have shown, this permits a more detailed characterization of lineage roles and generates a valuable resource that can be used by ourselves and others, for instance, to perform phenocopy analysis that would find further TF-CF interactions. We have therefore performed *in*

in vivo Perturb-seq for the following TF, chosen as master regulators of specific lineages, allowing us to functionally cover our differentiation space.

Ceba
Cebb
Cebe
Irf8
Gat1
Klf1
Hox7
Hox9
Mei1
Run1
Atf4
Tal1
Cite2
Cite4
Pax5
Ebf1
Elk3
Elk4
Erg
Gaba

As shown in the updated Figure 2 and Extended Figure 4, disruptions of *Hoxa7*, *Meis1*, *Cebpa*, and *Klf1* as well as both *Ebf1* and *Pax5* generate a strong depletion in their cognate lineages. In addition, these additional experiments confirm the described role of *Irf8* as a key monocyte

regulator. Beyond their role as excellent controls, we believe that these *in vivo* TF dependency maps at single-cell resolution will themselves provide a valuable resource, providing further insights that have as yet not been explored, in particular their effects on gene expression signatures, as described in Extended Figure 6.

In addition to these TF perturbations we have generated data for another 20 CFs, so our *in vivo* Perturb-seq dataset now comprises 80 factors in total (60 CFs + 20 TFs) described in Figure 2 and Extended Figures 4 and 6. This extended analysis further shows the diversity of CF roles in haematopoiesis, for instance we uncover another dichotomy for BAF complexes, where cBAF and pBAF members act as, respectively, Myeloid and Erythroid regulators.

2. Replicates appear to be missing in several scenarios making the evaluation of the robustness of some of the effects difficult. How robust are the results derived from the CRISPR screen? Have they been performed in replicates? If so please provide information on robustness, if not it would be appreciated if this can be done, at least, for selected candidates.

We thank the reviewer for this comment. We apologise for not having initially explained the screens in sufficient detail. First, we would like to note the relative scarcity of primary LSKs, making it difficult to perform intermediate-sized screens with LSKs; achieving robust coverage (>500X) is obviously a cumbersome and cell/mouse-dependent task. Therefore, we opted for a “middle-way approach”, where we performed a single-replicate screen for 680 CFs plus validation screens including replication of ~200 CFs (that included the strongest hits) for the lineage priming and self-renewal systems (which were the most informative screens). We now highlight this in the text and include specific details of this approach in the materials and

methods. As can be seen below, the reproducibility for these 200 CFs is very high. To further clarify this strategy and experimental basis, we have also added an additional panel showing reproducibility for these 200 CFs (Extended Figure 1d).

In addition, we would like to point out that we have extensively validated the CRISPR screen results using 12 single-CF KO in *ex vivo* analysis (Extended Figure 2) and smaller libraries of sgRNAs in *in vivo* analysis (*Brd9*, *c/nxBAF*).

3. While in vivo CRISPR knock out screens are not possible in humans, it would be of interest to what degree the findings obtained in this study are conserved in humans. Can published CHIP or ATAC-seq datasets be employed to at least partly address this.

We fully agree with the reviewer that validation of our results in a human system would be ideal. However, such analyses are difficult in primary human cells and datasets do not as yet exist for them. We have also searched for CF ChIP-seq or Cut&Run binding patterns in primary human cells, but again we have not found any relevant resource to address if the murine roles (as

reported here) are conserved in humans. However, we instead considered screens in human cell lines, predominantly cancer cell lines, to test the generalisability of our findings across organisms. We thus obtained results from genome-wide CRISPR screen from a publication of Sabatini et al. (Wang T et al Cell, 2017), as well as those from the DepMap consortium, where various leukaemic cell lines were tested for viability following genome-wide gene knock-out screens (“genetic dependency”). As both datasets show very similar results, we continued only with the DepMap data. We next obtained viability scores from our CRISPR screens by comparing TPM-normalized reads for each guide RNA between Cas9 and Non-Cas9, resulting in a log fold change that measures the effect of a KO on viability, akin to the dependency values provided by DepMap, and then compared the results of our study to DepMap. Although our analyses differ from those of the DepMap consortium, resulting in different scales, we observe a clear correlation in the results, as well as some distinct outliers. These results corroborate our findings and demonstrate the relevance also in the human setting.

4. Can the authors elaborate on the inclusion of two read-outs to assess the CF-KO lineage-specific dependencies in the Perturb-seq data? Why not only use CF-KO enrichment/reduction in particular clusters instead of entire lineages? This would allow the identification of the exact cell states that are altered without the need of pseudotime. Alternatively, the authors could also use differential abundance testing in a continuous manner rather than using discrete clusters using, for instance, the miloR package. If the authors come to the conclusion that their approach is superior to our suggestions, feel free to keep it that way.

We thank the reviewer for this considered and nuanced comment. Indeed, in our initial analysis, we started with many small clusters, but found that this approach lacked statistical power due to the sparsity of individual CF-KOs. Furthermore, it was difficult to visually present 50+ clusters. We therefore focused on entire lineages. Since we have now significantly expanded the dataset, we decided to test a more fine-grained approach that permits the interrogation of relevant differentiation stages. This approach now comprises the majority of **Figure 2** and we now believe that it improves the paper significantly.

In addition, to more faithfully representing the pleiotropic roles of CFs, we have now performed additional analyses to better dissect CF contributions to lineage progression. We thus generated 3 dedicated analyses that score the contribution of factors to specific branching points (i) early Myeloid vs Erythroid, ii) overall Myeloid vs Erythroid, and iii) Monocyte vs Granulocyte

differentiation. Finally, we kept the trajectory analysis, which (we believe) provides the continuous measure of differentiation that was suggested by the reviewer. Trajectory analyses thus enable us to characterise stalled trajectories that are not always evident when using a segmented, cluster-based approach, particularly with somewhat arbitrary boundaries.

In summary, we believe that our study represents a significant methodological and conceptual improvement versus previous *in vivo* perturbation studies with single-cell resolution. For instance, Giladi et al (2018 PMID: 29915358) simply measured the effect of specific KOs on the expression of differentiation markers as proxies of effects on cellular lineages (considering neither cluster IDs nor trajectories). Moreover, the Giladi study reports major heterogeneity on the distribution of NTC cells that prevents a cluster base analysis.

Finally, we followed the reviewer's suggestion and performed MiloR-based analysis. As MiloR is developed for comparisons between a limited number of conditions, we re-wrote some of the underlying functions, to be able to test a large number of coefficients (CF-KOs). For some KOs (such as *Brd9*, *Rcor1* and *Smarcd2*) MiloR analyses produced patterns very similar to our analysis. However, MiloR failed to capture other effects, even known and gross effects such as the disruption of Myeloid differentiation after *Cebpa*-KO, or the increased accumulation of myeloid cells in *Hdac3*-, *Rbbp4*- and *Chd4*-KOs (see figure below). We believe that, because MiloR divides the single cell space into an extreme number of very small subclusters (neighbourhoods), it misses larger patterns that are not confined to very small clusters.

5. In Figure 3A, can the authors please clarify whether the gene expression was analyzed by cell cluster or using all cells from the specific CF-KO? If the latter, the results could be reflecting the aberrant cell abundancies rather than the molecular mechanisms underlying the CF depletion as suggested by the authors.

We thank the reviewer for raising this point, which we have internally debated and also tested. In the paper, we purposefully compare cells with CF-KOs to NTCs independently of their cluster IDs based on a rationale described in detail further below. Therefore, the expression patterns may in part reflect patterns of cellular abundancies.

Nevertheless, to test the effect of cluster membership, we performed DE analysis but added the cluster membership as a covariate to the model, such that the model “knows” about cluster differences and therefore differences of cell abundance between clusters are accounted for. We next compared these results of this “cluster covariate” approach to our initial “cluster-agnostic” approach and find that the log fold changes calculated are clearly correlated (figure below).

Therefore, we are satisfied that the consideration of cluster membership does not majorly affect our DE results.

This figure shows the log fold changes obtained for three exemplary CF KOs between the two approaches (“cluster covariate” and “cluster-agnostic”). The same CF tested with the same approach produces identical log fold changes. The same CF tested with two different approaches generates highly similar (correlated) log fold changes, demonstrating that the two approaches produce similar results. This is not an artefact of our analyses, as log fold changes obtained from different CF KOs are not correlated.

This figure summarises the correlations of log fold changes shown for examples above but across all CFs. Log fold changes are calculated by comparing KO to NTCs, and then correlated. On the right, we show correlations obtained for the same CF but with different approaches (“cluster covariate” and “cluster-agnostic”). As observed in the examples above, the two approaches produce results that are highly correlated across all CFs. This is compared to the correlation between different CFs tested with the same approach on the left, which produces non-correlated log fold changes.

In addition to the above, we decided to retain our initial, cluster-agnostic analyses based on the consideration that any transcriptional changes observed when comparing KOs to NTCs are ultimately consequences of CF-KOs, including shifts in cell populations. Ideally, we would be able to distinguish direct and indirect effects, i.e. genes that are directly regulated by a CF versus genes that are changed due to indirect effects, such as downstream TFs. We felt that neither approach (using or not using the cluster covariates) properly addresses this problem, in particular considering that transcriptomics was performed after 14 days. Therefore, instead of trying to solve this problem computationally, we thus chose to focus on experimental analyses with ChIP-seq and ATAC-seq (presented in Figures 3-4) in order to specifically study the direct CF targets.

In summary, we first found that considering cluster membership only marginally effects results. We further concluded that DE results are potentially indirect effect, which we follow up with dedicated analyses more suitable to identify direct CF targets.

6. In the ATAC-seq experiment shown in Fig.3, did the authors only analyze the differentiated cell types? Only analyzing the end-points of differentiation might preclude the identification of dynamic changes occurring during differentiation. The authors could think about performing scATAC-seq instead, for instance with Spear-ATAC. This is not an absolute must, maybe alternative approaches could be used to provide insights into dynamics (<https://www.nature.com/articles/s41467-021-23213-w>).

We have analysed the effect of CF-KOs in bulk by sorting all the cells with a specific KO (BFP+ cells) after 7 days of culture under either lineage priming or Myeloid differentiation conditions. We therefore have a continuum of cell types at this time point and agree that these bulk profiles are somewhat confounded by cellular heterogeneity (as we have shown, each CF-KO alters the lineage proportions differently) and lack temporal resolution. However, despite these obvious limitations, we would like to note that our system has found substantial differences in chromatin accessibility that have allowed us to connect different TF, via their motifs, to three CFs that show similar myeloid dependencies: Smarcd2, Kmt2d and Wdr82. Therefore, we believe that our approach is generally sufficient to capture strong differences in TF-CF dependencies.

Others like the Schübeler lab have used similar bulk approaches (in ES cells) to interrogate the roles of SWI/SNF (BAF) and ISWI remodelers in mediating TF accessibility and similarly found major differences in the TF dependencies between Smarca4 SWI/SNF and Smarca5 ISWI. Inspired by such an approach, we decided to interrogate a larger cohort of CF-KOs in two differentiation systems to explore whether specific TF-CF connections help to explain the observed CF lineage dependencies. We believe this point is well supported in our current

Figure 3.

That said, we believe that this reviewer has raised a very valid and interesting point regarding the dynamic nature of such accessibility changes and their likely temporal relationship to direct and indirect effects of CF-KO. We have therefore assessed the accessibility dynamics following KO of two exemplar CFs: *Kmt2d* and *Wdr82*. We have profiled chromatin accessibility at days 3, 5 and 7 days after these CF-KOs to distinguish early changes in TF motif accessibility (more likely direct CF targets) from late TF motif perturbation potentially reflecting indirect targets.

This demonstrates early alteration of myeloid TF motif accessibility in both cases that become more pronounced as the culture time progresses. We therefore believe that these represent direct effects of the CF on specific motifs, that later “spread” to other genomic locations. In addition, we see that increased motif accessibility after *Kmt2d* and *Wdr82*-KOs appears later in time (especially Gata factors after *Wdr82*-KO), suggesting a likely indirect (secondary) effect derived from the disruption of “upstream” myeloid TF programs. We have included this analysis in **Extended Data Figure 7** and reported these findings in the main text.

Finally, we would like to thank the reviewer for the excellent suggestion of trying Spear-ATAC. We are very keen on using such an approach to study the chromatin regulatory roles of the different CFs. However, we consider that developing and optimising this technique will be lengthy and cumbersome (both experimentally and analytically). In addition, we feel that it falls beyond the scope and remit of the current paper and that it would unduly slow us down in publishing our current story. Moreover, we feel that the results produced by Spear-ATAC, especially *in vivo*, would merit a separate article in itself and would be wasted simply as a single panel within a figure from a paper with an already complicated story. In the future, we will endeavour to apply this technique to further questions related to the function of gene regulatory complexes in healthy and malignant haematopoiesis.

7. Section 246 following. Are there concrete indications that disruptions of ncBAF is indeed involved in leukemogenesis? Can the authors demonstrate that disruptions of ncBAF functionally predisposes to leukemias? If not, the term “pre-leukemic” should be avoided.

Whilst there is some genetic and mechanistic evidence linking the aberrant splicing of *BRD9*, which leads to non-sense mediated decay and protein downregulation in CLL, MDS and uveal melanoma (Inoue et al, PMID31597964 and https://ashpublications.org/blood/article/134/Supplement_1/637/426534/Spliceosomal-Disruption-of-the-Non-Canonical-SWI-human-cells), we do not have formal evidence that disruptions of ncBAF are definitely involved in leukemogenesis. Therefore, we agree with the circumspection of this reviewer and will remove the term pre-leukemic and substitute it with “pre-leukemic like”, a softer and more descriptive term implying the phenotypic similarities of pre-leukaemia to those of the *Brd9*-KO.

8. Are the cellular fitness measurements shown in Fig. 5B a specific effect of depleting the CFs in leukemia cells or is cell viability also affected in the CF-KO in normal progenitors? Can the authors show the viability for the CFs used in the Perturb-seq?

We thank the reviewer for this insightful comment. To address this point, we have compared the representation of guides RNAs in the plasmid pools to the cell counts in the Perturb-seq data (Extended Data Figure 4e). This shows characteristic drop-out scores for each of the TFs and CFs assessed *in vivo* at 14 day post-transplant indicating the growth advantage generated by disrupting certain CFs like ncBAF members or TFs like *Irf8*-KO – previously reported as a preleukaemic mutation (PMID 8861914).

Therapeutically, we observed that BAF disruption is deleterious in both the normal and leukemic scenario, indicating that a safer therapy would involve targeting leukemic specific cBAF-TF interactions (as we report and suggest in the paper). Similarly, disruption of *Kmt2d* *in vivo* doesn't seem toxic to normal haematopoiesis, although it will likely cause depletion of progenitor and Monocyte subsets. However, this would be therapeutically tolerated in the short-term and would be expected to resolve when an agent specifically targeting KMT2D was stopped.

9. Loss of *Smarcd2* or *Kmt2d* in leukemic cells was shown to lead to reduced accessibility in particular *Stat5a* loci, but this was only shown for two particular examples. Could the authors show the loss of accessibility at *Stat5a*-bound loci in a more systematic way?

We apologise for our being highly selective in our choice of loci to show for differential accessibility upon CF-KO in leukaemia. It was simply our attempt to keep the narrative as simple as possible. In our revised manuscript we show global accessibility changes at *Stat5a* sites specific for the leukaemia in *Smarcd2*-KO, *Kmt2a*-KO and *Kmt2d*-KO and for two other leukemic specific TFs: *Runx1* and *Runx2* (Figure 6e, Extended Data Figure 10f). The three CF-KOs show a clear decrease in the accessibility of the loci bound by these CFs. Again, we thank the reviewer for his suggestion as it improves the message of the paper.

e

Accessibility Change at TF bound sites after CF-KOs:

Minor points:

- In general, methods and samples used in the different experiments are not always clear to the reader (i.e. what cell types are analyzed in the ATAC-seq in Fig.3?; what was the reasoning for using CITE-Seq and why is it only mentioned in the discussion but not the main text?).

We apologise for this lack of clarity and have rectified this in our revised manuscript. For example, we now mention that we used CITE-seq to characterise leukaemic differentiation trajectories in the main text.

- Figures are not always called or are mistakenly called in the main text (i.e. Fig. 4C is not called and in line 281 Fig. 4J is called when it should be Fig. 4I). Please check throughout the text.

We apologise for this mistake and have corrected this in the revised manuscript

- Significance is missing in Fig. 6H. Please provide statistics.

We apologise for this mistake and have corrected this in the revised manuscript

Reviewer #3:

Remarks to the Author:

In this paper, the authors perform a very elegant CRISPR screen to determine which chromatin proteins, out of a list of 550, contribute to specific lineage decisions as well as maintenance of multipotency. They validated a few key factors and then performed an in vivo Perturb-seq experiment with 40 key factors. One of the more interesting things to come out of this is the high level of specificity observed between highly related complexes – for example, expansion of erythroid cells with Kmt2d KO vs expansion of B cells with a Kmt2a KO. They then went on to explore TF expression profiles and in particular used digital footprinting (with ATAC-seq data) to determine which TF motifs were associated with chromatin protein function. They explore the differential role of cBAF and ncBAF complexes in more detail, and interrogate the function of several chromatin factors in an NPM1/Flt3 AML model. Interestingly, despite the differential impact on normal development, the different chromatin proteins all seemed to display a uniform impact on decreased growth of leukaemia cells, except for Brd9. Finally, they look at changes in the distribution of key chromatin proteins in normal versus leukemic cells.

Overall, this is a very interesting screen and analysis that will provide a fantastic ongoing resource for the field, and could potentially provide a useful platform for drug discovery.

We thank this reviewer for their generally very positive and constructive comments and are glad that they find our manuscript both interesting and a fantastic resource for the field. We also appreciate that this reviewer recognises the clinical implications of our work and the potential of our screen as a platform for drug discovery.

I have a few questions and points that need to be clarified that will make the paper more understandable, but I want to make it clear that I find this paper quite exciting and will enjoy returning to it again in the future:

Again, we thank the reviewer and are happy that they will enjoy further reviewing our resubmission.

1. In Figure 3E, the digital footprinting analysis, is the loss of a TF footprint due to the loss of expression of the TFs, or are the TFs still expressed in many cases but have simply lost accessibility to their binding sites?

As can be seen in the bottom panel, below, we have assessed the expression of specific TFs after CF-KOs in the same conditions where the TF motif accessibility was measured. We detect changes in TF expression upon CF depletion but, in general, these don't mirror the TF motif accessibility changes elicited upon the same CF-KO. Therefore, we believe that the loss of a TF footprint, in general, relates to a loss of activity and binding of that TF rather than its decreased expression. We have made this apparent, demonstrating the fold change in expression of individual TF in Supplementary Figure 6.

2. Related to (1), digital footprinting using ATAC-seq can be prone to bias, and the presence/absence of a motif is not the same as the presence/absence of a TF. Can ChIP-seq for some key TFs be used to validate this TF binding profile change in the different chromatin protein KOs?

We appreciate this comment from the reviewer and agree that validation of TF motif analysis would greatly strengthen our story. We have therefore performed ChIP-Seq for *Cebpa* and *Cebpe* in primary GMPs to validate the TF motif results found upon *Brd9*-KO. We observe a strong correlation between chromatin accessibility changes at *Cebp* motifs derived from TOBIAS and the accessibility changes at the specific *Cebpa/e* binding loci measured by ChIP-seq. This information is now included in Figure 4 (see below).

3. The ChIP-seq heat maps indicate that *Smarchb1*, *Brd9* and *Kmt2a/2d* have quite different binding profiles, but it is a little unclear what the example tracks in Extended Figure 6A are meant to display. The quality of the ChIP-seq looks excellent, but the example tracks look quite similar to me, and the panel seems to display genes where the binding profile for these 4 proteins is quite similar (although it changes with differentiation). This is fine as it is, but it would also be useful to see some example tracks for differential gene target binding e.g. an important gene that binds *Smarchb1* but not *Brd9* for instance.

We thank the reviewer for this point and apologise that our highly specific choice of loci did not allow the genomewide picture to be fully elicited. We have addressed this request in two ways; (1) we have chosen a few other snapshots of differential gene target binding, and (2) we have generated heatmaps of global differential binding, for *Brd9* and *Smarchb1* ChIP-seq in Monocytes and GMP that further confirm the cBAF (*Smarchb1*) to ncBAF (*Brd9*) switch in mature Myeloid cells. These novel analyses are shown in Extended Fig. 8c-d.

4. Line 267-269: "This suggests that the myeloid maturation defect observed in *Brd9*-KO cells may relate to impaired recruitment and activity of *Cebp* and *Ap1* TFs." I'm not sure I understand this statement. If we take the motif enrichment at face value for a moment, does this statement

imply that the chromatin proteins are responsible for recruiting TFs? This is quite a different view from the way the field thinks, and I'm not sure I follow the authors logic as to how their data supports this interpretation. A bit more explanation would be helpful.

We are sorry for the misunderstanding. Our prevailing view is that the TF recruits the CF, particularly if it has pioneering activity. This interaction likely initiates chromatin changes generating and maintaining a stable nucleosome free region (detected by ATAC-seq) and this

presumably recruits the rest of the transcriptional machinery activating transcription and subsequent cell differentiation. Our data strongly suggests that in the absence of the partner CF (i.e. ncBAF complex), the TF fails to produce open chromatin and induce gene expression. Others have shown similar effects of specific TF motif accessibility following acute depletion of the BAF ATPase module (Kubicek - PMID: 33558760)

We do not know if, in the absence of the CF (Brd9), the TF (Cebpa/e) remains bound to its motif within closed chromatin via its pioneering ability. This is an interesting question, but is beyond the remit of our study. However, what is clear is that loss of the CF leads to a loss of chromatin accessibility. We currently posit that in the absence of the CF's inherent or recruited remodelling activity, the affinity of the TF for its motif is diminished leading to reduced residence time on chromatin. In this scenario, perhaps the ability to recruit CFs with specific/strong effects on accessibility is another defining property of pioneer factors - this is an interesting speculation, and one that we intend to explore in the future, but again it falls beyond the scope of this paper.

5. Again, it would be useful to know if the digital footprinting analysis for Figure 4H and 4I can be validated with at least some TF ChIP-seq in NTC vs Brd9 KO cells.

As discussed above, we have validated this motif analysis by ChIP-Seq of Cebpa and Cebpe in primary GMPs. When integrating this dataset with the accessibility profiles in Control (NTC) GMPs and Brd9-KO GMPs we observe a strong decay in accessibility at Cebpa and Cebpe binding loci validating our hypothesis. We have included this analysis in Fig. 4g-i

6. Menin inhibitors are currently in clinical trials for NPM1 mutant leukemias. Do Menin inhibitors produce a similar profile as Kmt2a and Men1 in normal cells and in the Npm1/Flt3 model? I am thinking of the in vitro differentiation experiments. This may seem to be a side issue, but I am thinking that one use of the system(s) outlined here (especially the in vitro differentiation) would be for future drug discovery, and it would be useful to see how a highly specific drug that is currently going through clinical trials performs relative to the gene specific and complex KOs.

We thank this reviewer for this excellent suggestion. We have indeed applied the Menin inhibitor to the *Npm1c/Flt3-ITD* model and can demonstrate that it very nicely phenocopies the effects of the Kmt2a-KO and Men1-KO to decrease the growth kinetics of the AML and also to strongly mandate myeloid differentiation. This data is now shown in Figure 5i and Supplementary Figure 7 and mentioned in the results section. Beyond the scope of our first report we also feel that this demonstrates the utility of our perturbation system to further shed light on potential mechanisms of action of epigenetic and other targeted therapies.

As already mentioned, I overall like this paper quite a bit and think it is interesting and provides important results. Something I already mentioned is that I am still not quite sure what the authors are trying to say about the relationship between TF binding and chromatin protein activity. There is an over reliance on ATAC-seq and motif analysis to represent TF binding (except for the specific case of Stat5a), which is not a good substitute for TF specific ChIP-seq. In addition, reduced ATAC signal in the presence of a KO (e.g. Figure 6G) does not necessarily indicate that the chromatin factor is directly responsible for maintaining open chromatin, as the effect could be indirect (e.g. loss of expression of a remodeling factor in the KO). If the authors are trying to say that chromatin protein activity is responsible for TF recruitment, I think they need to clarify how their data supports this view. I think the data is still consistent with the view that TFs are responsible for recruiting specific chromatin protein complexes, and the specificity of these different complexes comes from both differential TF-CF dependencies (for example, like that seen in this recent paper Hendy et al 2022, PMID: 36113480) and/or possibly from different chromatin protein-promoter or enhancer specificities (e.g. Neumayr et al 2022 PMID: 35650434; Haberle et al 2019 PMID: 31092928).

We thank the reviewer for this comment and we apologise for our lack of initial clarity. In general, we agree that our data are most consistent with the sequence of events being that TF binding likely precedes CF recruitment, particularly for TF with pioneering activity, and that the TF-CF complex thereafter works together to maintain the chromatin accessibility state. However, our CF have multiple functions (ATP-dependent remodelers, epigenetic -readers, -writers, and -erasers, and co-repressors and co-activators), and it is likely that no simple “one size fits all” mechanism governs CF-TF interactions. Therefore, we also suggest that an alternative and not mutually exclusive explanation could be a sequential model, where specific CFs, already deployed through multivalent interactions with epigenetic modifications, regulate subsequent TF activity by modulating the chromatin state at their binding sites. This may be particularly apposite for dynamic differentiation trajectories and we propose this as a possible mechanism whereby the Chromatin Repressors attenuate myeloid pro-inflammatory TF responses. Regardless of the specific detail, central to the CF-TF collaboration appears to be the CF-mediated regulation and maintenance of TF accessibility and binding. As evidenced in our dynamic ATAC studies, early alterations in accessibility are observed even following KO of two methyltransferase complex components (*Kmt2d* and *Wdr82*). These data also inform CF-CF interactions with regulatory potential; the changes in accessibility suggest that the MLL4 and SET1 complexes recruit chromatin remodelers, likely BAF members, an observation reinforced by the strong KO phenocopy observed between *Kmt2d* and several cBAF subunits. However, whether this recruitment involves direct protein-protein interactions, or indirectly and

dependent upon catalysis, through configuring a local pattern of histone methylation that recruit remodelers via their own reader modules, will require further investigation.

We have outlined these thoughts in our revised discussion.

Minor points

1. The legend for Figure 4 skips D.

We have corrected this in our revised manuscript.

2. Figure 4 panel, the Day 28 and Day 14 labels are reversed, I think. If they are not, then the authors statement that Brd9 KO cells are expanded is not correct.

We have corrected this in our revised manuscript.

Decision Letter, first revision:

18th Apr 2023

Dear Brian,

Thank you for submitting your revised manuscript "Systematic study of chromatin factor function uncovers strong lineage determining roles and divergent behaviours between normal and malignant haematopoiesis" (NG-A61046R). It has now been seen by the original referees and their comments are below. The reviewers find that the paper has improved in revision, and therefore we'll be happy in principle to publish it in Nature Genetics, pending minor revisions to satisfy the referees' final requests and to comply with our editorial and formatting guidelines.

Sincerely,

Michael Fletcher, PhD
Senior Editor, Nature Genetics

ORCID: 0000-0003-1589-7087

Reviewer #2 (Remarks to the Author):

The authors have responded satisfactorily to my concerns and I would recommend its publication in this journal.

Reviewer #3 (Remarks to the Author):

The authors have done an extensive amount of work to address the reviewer comments and I think have greatly improved the paper. For my specific comments, they have clarified their view on the relationship between TF binding and chromatin protein binding. They have also provided some very exciting data on the function of Menin inhibitors in their system, thus providing a potential way to use this as a platform for drug discovery. I also appreciate the new TF ChIP-seq data. Overall, I remain excited about the results in this study and the reviewers have answered my questions to my complete satisfaction.

Final Decision Letter:

11th Jul 2023

Dear Brian,

I am delighted to say that your manuscript "In vivo screening characterizes chromatin factor functions during normal and malignant hematopoiesis" has been accepted for publication in an upcoming issue of Nature Genetics.

Your paper will be published online after we receive your corrections and will appear in print in the next available issue. You can find out your date of online publication by contacting the Nature Press Office (press@nature.com) after sending your e-proof corrections. Now is the time to inform your Public Relations or Press Office about your paper, as they might be interested in promoting its publication. This will allow them time to prepare an accurate and satisfactory press release. Include your manuscript tracking number (NG-A61046R1) and the name of the journal, which they will need when they contact

our Press Office.

Please note that *Nature Genetics* is a Transformative Journal (TJ). Authors may publish their research with us through the traditional subscription access route or make their paper immediately open access through payment of an article-processing charge (APC). Authors will not be required to make a final decision about access to their article until it has been accepted. [Find out more about Transformative Journals](https://www.springernature.com/gp/open-research/transformative-journals)

Authors may need to take specific actions to achieve [compliance](https://www.springernature.com/gp/open-research/funding/policy-compliance-faqs) with funder and institutional open access mandates. If your research is supported by a funder that requires immediate open access (e.g. according to [Plan S principles](https://www.springernature.com/gp/open-research/plan-s-compliance)) then you should select the gold OA route, and we will direct you to the compliant route where possible. For authors selecting the subscription publication route, the journal's standard licensing terms will need to be accepted, including [self-archiving-and-license-to-publish](https://www.nature.com/nature-portfolio/editorial-policies/self-archiving-and-license-to-publish). Those licensing terms will supersede any other terms that the author or any third party may assert apply to any version of the manuscript.

Please note that Nature Portfolio offers an immediate open access option only for papers that were first submitted after 1 January, 2021.

You can now use a single sign-on for all your accounts, view the status of all your manuscript submissions

and reviews, access usage statistics for your published articles and download a record of your refereeing activity for the Nature journals.

If you have not already done so, we invite you to upload the step-by-step protocols used in this manuscript to the Protocols Exchange, part of our on-line web resource, [natureprotocols.com](https://www.nature.com/natureprotocols.com). If you complete the upload by the time you receive your manuscript proofs, we can insert links in your article that lead directly to the protocol details. Your protocol will be made freely available upon publication of your paper. By participating in [natureprotocols.com](https://www.nature.com/natureprotocols.com), you are enabling researchers to more readily reproduce or adapt the methodology you use. [Natureprotocols.com](https://www.nature.com/natureprotocols.com) is fully searchable, providing your protocols and paper with increased utility and visibility. Please submit your protocol to <https://protocolexchange.researchsquare.com/>. After entering your [nature.com](https://www.nature.com) username and password you will need to enter your manuscript number (NG-A61046R1). Further information can be found at <https://www.nature.com/nature-portfolio/editorial-policies/reporting-standards#protocols>

Sincerely,

Michael Fletcher, PhD
Senior Editor, Nature Genetics
ORCID: 0000-0003-1589-7087